# A small molecule HIF-1α stabilizer that accelerates diabetic wound healing

Guodong Li [1,7], Chung-Nga Ko[2,7], Dan Li[1,7], Chao Yang [1,7], Wanhe Wang [2], Guan-Jun Yang[1], Carmelo Di Primo [3,4], Vincent Kam Wai Wong[5], Yaozu Xiang[6], Ligen Lin[1✉], Dik-Lung Ma[2✉] & Chung-Hang Leung [1✉]

Impaired wound healing and ulcer complications are a leading cause of death in diabetic patients. In this study, we report the design and synthesis of a cyclometalated iridium(III) metal complex **1a** as a stabilizer of hypoxia-inducible factor-1α (HIF-1α). In vitro biophysical and cellular analyses demonstrate that this compound binds to Von Hippel-Lindau (VHL) and inhibits the VHL–HIF-1α interaction. Furthermore, the compound accumulates HIF-1α levels *in cellulo* and activates HIF-1α mediated gene expression, including *VEGF*, *GLUT1*, and *EPO*. In in vivo mouse models, the compound significantly accelerates wound closure in both normal and diabetic mice, with a greater effect being observed in the diabetic group. We also demonstrate that HIF-1α driven genes related to wound healing (i.e. *HSP-90*, *VEGFR-1*, *SDF-1*, *SCF*, and *Tie-2*) are increased in the wound tissue of **1a**-treated diabetic mice (including, *db/db*, HFD/STZ and STZ models). Our study demonstrates a small molecule stabilizer of HIF-1α as a promising therapeutic agent for wound healing, and, more importantly, validates the feasibility of treating diabetic wounds by blocking the VHL and HIF-1α interaction.

[1] State Key Laboratory of Quality Research in Chinese Medicine, Institute of Chinese Medical Sciences, University of Macau, Macao, China. [2] Department of Chemistry, Hong Kong Baptist University, Kowloon Tong, Hong Kong, China. [3] Laboratoire ARNA, University of Bordeaux, Bordeaux, France. [4] INSERM U1212, CNRS UMR 5320, IECB, Pessac, France. [5] State Key Laboratory of Quality Research in Chinese Medicine, Macau University of Science and Technology, Macao, China. [6] Shanghai East Hospital of Tongji University, School of Life Sciences and Technology, Tongji University, Shanghai, China. [7] These authors contributed equally: Guodong Li, Chung-Nga Ko, Dan Li, Chao Yang. ✉email: ligenl@um.edu.mo; edmondma@hkbu.edu.hk; duncanleung@um.edu.mo

Poor wound healing, especially in diabetic patients, leads to substantial morbidity and mortality and can have a far-reaching socioeconomic impact[1]. The annual worldwide market for advanced wound healing products for hard-to-heal wounds and scars exceeds US$5 billion[2]. Wound healing in diabetic patients is compromised due to impaired wound contraction, reduced blood supply, and infection[3–5]. To date, several treatment strategies for diabetic wounds are available, including regular debridement, surgical revascularization, infection therapy, pressure-offloading, and bioengineered alternative tissue products[6–8]. However, most treatments are effective only for mild to moderate wounds and are not 100% effective in preventing the risk of amputation and recovering full skin functionalities[9,10]. Therefore. the development of novel drugs or therapies for the treatment of diabetic wound healing in clinical practice is desperately needed.

Wound healing is a systematic and dynamic process involving epithelialization, angiogenesis, granulation tissue formation, and wound contraction, all of which are regulated by hypoxia-inducible factor 1-α (HIF-1α)[11,12]. In normal tissues, HIF-1α steady-state levels are low due to oxygen-dependent hydroxylation of HIF-1α by prolyl hydroxylase domain proteins (PHDs)[13]. This allows HIF-1α to combine with Von Hippel-Lindau (VHL) tumor suppressor protein to generate a complex that is readily recognized by the proteasome and rapidly degraded[14,15]. Notably, hyperglycemia can decrease the stability of HIF-1α, leading to the inhibition of HIF-1α target gene expression, which could account for the poor healing and ulcer complications in diabetic patients[16]. This suggests that strategies to improve the stability of HIF-1α, such as by blocking the interaction between VHL and HIF-1α, could be a promising strategy for the treatment of diabetes wound complications. Currently, several reported inhibitors of PHDs as stabilizers for HIF-1α have entered clinical trials for the treatment of chronic anemia[17]. However, PHDs inhibitors are associated with side effects, such as fatal liver necrosis for FG-2216, raising concerns about their safety and tolerability for widespread use[18].

Protein–protein interactions (PPIs) are attractive targets in drug discovery[19]. For example, venetoclax, a B-cell lymphoma 2 inhibitor used for the treatment of chronic lymphocytic leukemia, was the first small molecule inhibitor of PPIs approved by the FDA[20]. Ciulli and co-workers were the first to report a bona-fide VHL–HIF-1α protein–protein interaction inhibitor VH298 as a high-quality chemical probe of the HIF-signaling cascade in the hypoxia signaling pathway[21]. This inhibitor represented an attractive starting point to the development of potential new therapeutics targeting hypoxia signaling[21–23].

Over the last decade, metal-based compounds possessing several promising advantages have been explored for targeting PPIs[24]. Metal complexes, with their high structural diversity and readily tunable steric and electronic properties, can adopt a wide range of geometrical shapes based on the oxidation state of their metal center, and the nature of their co-ligands[25–30]. The sophisticated three-dimensional geometries available to transition metal complexes might allow them to be more effective in generating compounds with suitable shapes and functional groups that are complementary to the binding regions of PPI surfaces[31,32]. In this study, we report the identification of an iridium(III) complex as a potent inhibitor of the VHL–HIF-1α PPI, which can effectively induce the accumulation of HIF-1α in cellulo and in vivo. Moreover, we also evaluated the potential of this complex on wound healing in three diabetic mouse models, including leptin-receptor-deficient (db/db), streptozotocin-induced (STZ) and high-fat diet/streptozotocin-treated (HFD/STZ) diabetic mice models, demonstrating that the blocking of the VHL and HIF-1α interaction is a viable strategy for treating diabetic ulcers.

## Results

**Screening and structure-based optimization of small molecules as potential HIF-1α stabilizers by monitoring HRE-driven luciferase activity.** A library of cyclometalated Ir(III)/Rh(III) metal complexes **1–14** (as racemates) with diverse structures were selected for initial screening, in order to identify favorable substructures for the design of the next round of complexes (Fig. 1a). In cells, HIF-1α moves into the nucleus and binds to the hypoxia response element (HRE) in the promoters of transactivated genes. In this study, the dual-luciferase reporter (DLR) assay was performed to monitor the variations of HRE-driven luciferase activity induced by the complexes (Fig. 1f). Human embryonic kidney HEK293 cells were treated with complexes (10 μM) for 8 h, and the HRE-driven luciferase activity of the cell lysates was determined. Complexes that inhibit the interaction between VHL and HIF-1α would be expected to increase the level of HRE-driven luciferase activity in the cell lysates. The Rh(III) complex Rh(brpy)$_2$(dmeophen) **1** (where brpy = 2-(4-bromophenyl)pyridine and dmeophen = 4,7-dimethoxy-1,10-phenanthroline) emerged as the top candidate in the first round of screening (Fig. 1b), with slightly higher activity compared to the positive control compound, **P1** ((2S)-4-hydroxy-1-(2-(((Z)-2-(3-methoxybenzylidene)-3-oxo-2,3-dihydro benzofuran-6-yl)oxy)acetyl)pyrrolidine-2-carboxylic acid) (Supplementary Fig. 1), a previously reported inhibitor of the VHL–HIF-1α PPI discovered by our group[33].

Based on the structure of complex **1**, a focused library of 14 cyclometalated Rh(III) and Ir(III) complexes containing different C^N or N^N donor ligands were designed and synthesized (**1a–1n**) (Fig. 1c). This library was enriched in the brpy C^N and dmeophen N^N ligands that were identified in the first round of screening to be favorable substructures for potency. In the second round of screening, the Ir(III) complex **1a**, containing two 2-phenyl pyridines (ppy) C^N ligands and the dmeophen N^N ligand, showed the highest activation of HRE-drive luciferase activity and was about twice as potent as both the parent complex **1** as well as the positive control compound **P1** (Fig. 1d). Notably, complex **1a** demonstrated remarkable stability in a [$d_6$]DMSO/D$_2$O (9:1) solution for at least 7 days at 298 K as verified by $^1$H NMR spectroscopy (Supplementary Fig. 2) and in acetonitrile/H$_2$O (9:1) solution for at least 7 days at 298 K as determined by UV/Vis spectroscopy (Supplementary Fig. 3).

To further investigate the potency of complex **1a** at modulating HIF-1α transcriptional activity, a dose–response assay was carried out. The results showed that complex **1a** exhibited dose-dependent activation of HRE-driven luciferase activity in HEK293 cells (Fig. 1e). Moreover, the effects of the isolated ligands of complex **1a** on HRE-drive luciferase activity were also studied. The results showed that neither brpy nor dmeophen had any significant effect on HRE-drive luciferase activity (Supplementary Fig. 4). This demonstrates the role of the Ir(III) center in coordinating the structure of the entire complex in order to confer HIF-1α transcriptional activity.

**Complex 1a binds to VHL and antagonizes the interaction of VHL–HIF-1α in cellulo.** We performed co-immunoprecipitation (co-IP) experiments to further understand the mechanism of action of complex **1a**. HEK293 cells were incubated with complex **1a** for 2 h, and cell lysates were subjected to co-IP using VHL antibodies. The results showed that complex **1a** dose-dependently reduced the amount of HIF-1α-OH co-precipitating with VHL, suggesting that it was able to inhibit the VHL–HIF-1α interaction in the treated cells (Fig. 2a and Supplementary Fig. 5).

It is generally known that the efficacy of therapeutics is dependent on the extent of binding of the drug to the target

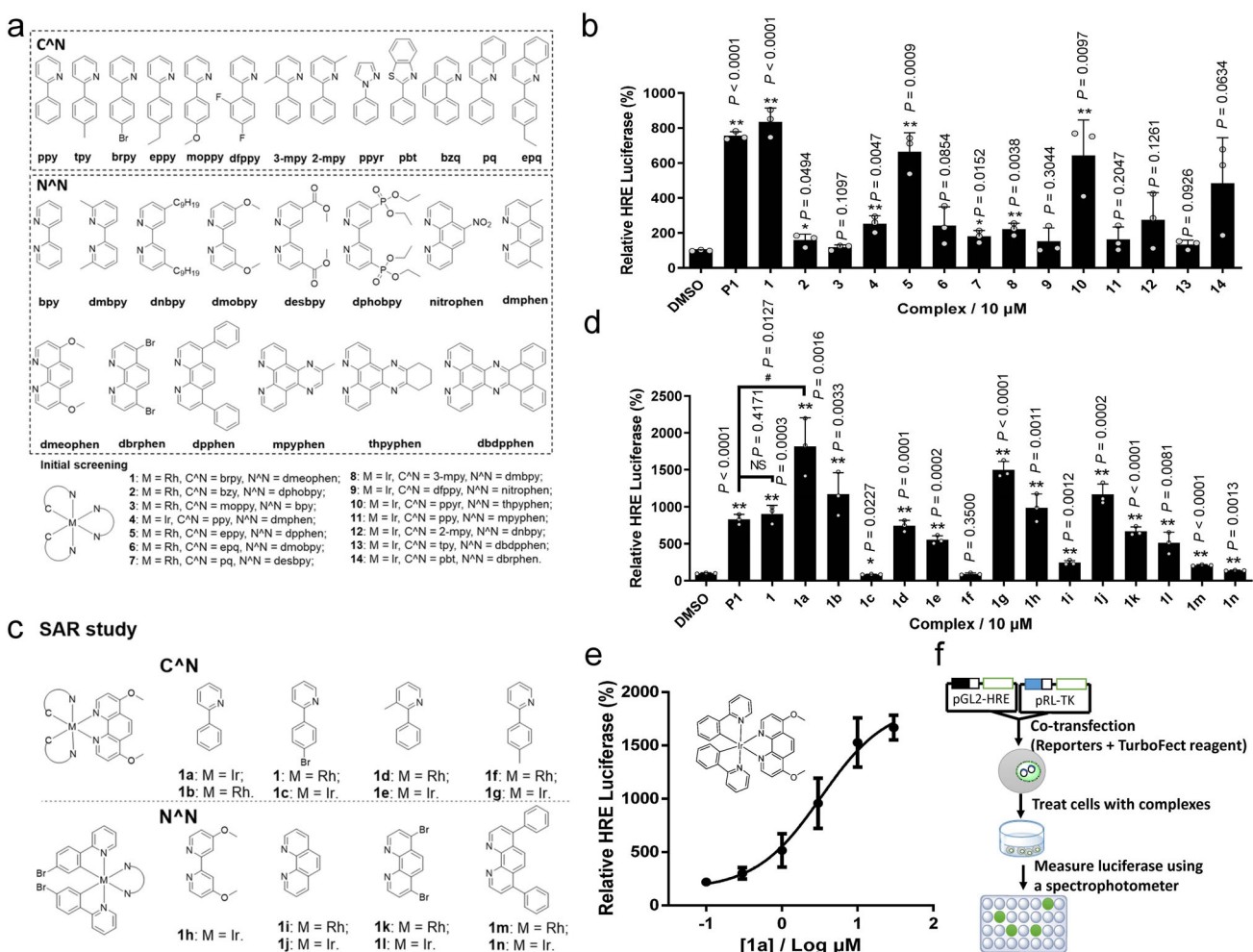

**Fig. 1 Effect of complexes on HIF-1α activation of the HRE-driven reporter as determined via a dual-luciferase reporter assay. a** Chemical structures of the small molecules used for preliminary screening. All complexes are as racemic $PF_6^-$ salts. **b** Effect of complexes **1–14** on HIF-1α activation of the HRE-driven reporter. HEK293 cells were treated with complexes **1–14** or **P1** for 8 h. Error bars represent the standard deviations (SD) of the results from three independent experiments. *P* values were calculated using a two-sided *t*-test. *$P < 0.05$, **$P < 0.01$ vs. DMSO group. **c** Chemical structures of the focused library of cyclometallated Ir(III) and Rh(III) complexes. **d** Effect of complexes **1a–1n** on HIF-1α activation of the HRE-driven reporter. HEK293 cells were treated with complexes **1a–1n** or **P1** for 8 h. **e** Dose-dependent effect of complex **1a** on HIF-1α-driven activity. HEK293 cells were treated with the indicated concentrations of complex **1a** for 8 h. Data are expressed as means ± SD ($n = 3$ independent experiments), *P* values were calculated using a two-sided *t*-test. #$P < 0.05$ 1a vs. P1, NS (not significant, $P > 0.05$) 1 vs. P1. *$P < 0.05$, **$P < 0.01$ vs. DMSO group. **f** Schematic diagram of the DLR showing co-transfected HRE-luciferase and pRL-TK plasmids in HEK293 cells. Data are expressed as means ± SD ($n = 3$ independent experiments). Source data are provided as a Source Data file.

protein. Therefore, the cellular thermal shift assay (CETSA) was performed to verify the engagement between complex **1a** and VHL and PHD2. After incubating HEK293 cell lysates with complex **1a** followed by heating to set temperatures, VHL and PHD2 levels in the soluble fraction were quantitated by Western blotting. In the presence of complex **1a** (3 μM), an obvious shift of about 6 °C in the VHL melting curve was observed (Fig. 2b). This result indicates that complex **1a** could stabilize VHL in cell lysates. In contrast, complex **1a** had no appreciable effect on the thermal stabilization of PHD2 (Fig. 2b). We performed further biophysical experiments to demonstrate the direct binding of complex **1a** to VHL in vitro. We first prepared plasmids to express and purify human recombinant VHL:ElonginB:ElonginC (VBC) complex (Supplementary Fig. 6a), which was verified using a pull-down assay (Supplementary Fig. 6b). Circular dichroism (CD) measurements showed a distinct shift of the signal when VBC complex was incubated with 3 μM of **1a** in 1% DMSO, while no significant changes were observed with 1% DMSO alone, indicating that **1a** could regulate VBC secondary structure via directly binding to

VBC complex (Supplementary Fig. 6c). We also verified VBC's high-affinity interaction with the HIF-1α peptide DEALAHyp-YIPD[34–36], which is involved with mediating the VHL–HIF-1α interface (Supplementary Fig. 6d–f). The stabilization of complex **1a** towards VBC was also corroborated using a fluorescence-based protein thermal shift assay (FTS), which revealed by a marked shift of the melting curve (ca. 3.0 °C) of purified VBC in the presence of complex **1a** with VH298 (ca. 5.0 °C) as a positive control (Fig. 2c). Meanwhile, isothermal titration calorimetry (ITC) revealed a $K_d$ value of $1.08 \pm 0.20$ μM (Fig. 2d) between complex **1a** and VBC complex, similar to the $K_d$ value of 2.06 μM determined using a competitive fluorescence polarization assay (Fig. 2e). The ITC data in Fig. 2d also indicates that **1a** binds to VBC complex with a 1:1 stoichiometry, and that the binding between **1a** and VBC is strongly enthalpy-driven ($\Delta G = -8.14$ kcal/mol, $-T\Delta S = -0.29$ kcal/mol, $\Delta H = -7.85$ kcal/mol), suggesting that hydrogen bonds and electrostatic interactions may play a key role in this binding. Moreover, the binding kinetics of **1a** from VBC was characterized by biolayer interferometry (BLI),

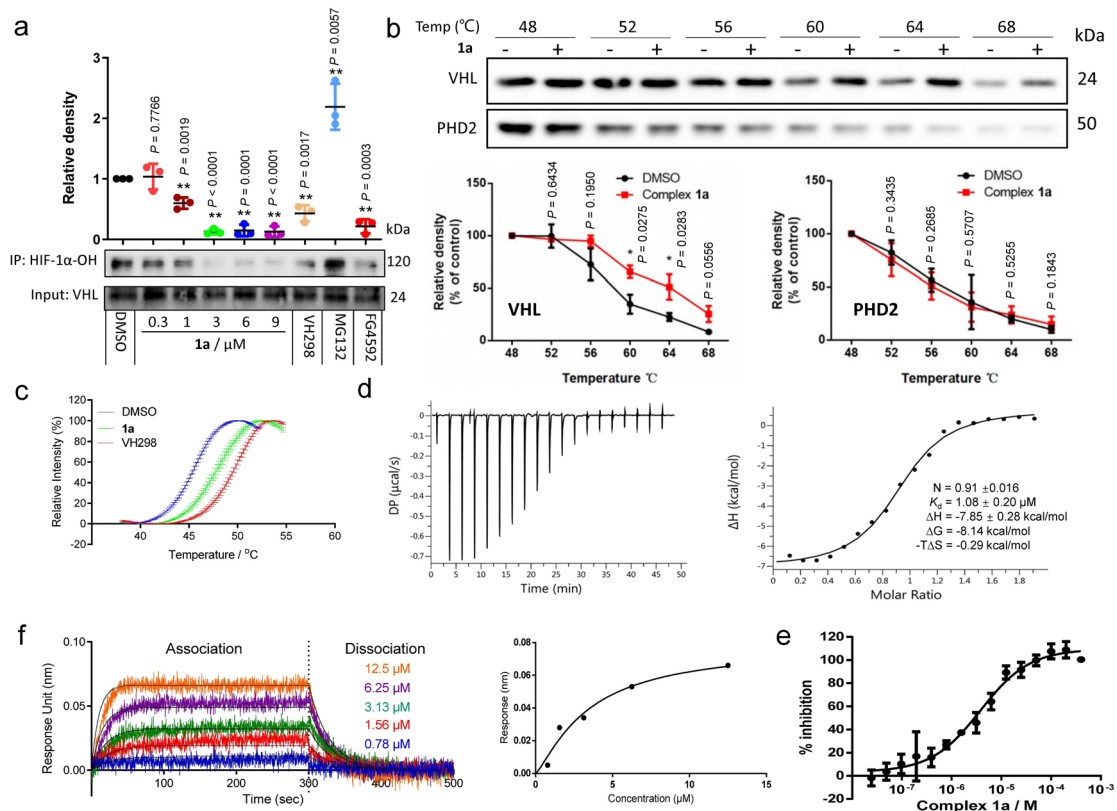

**Fig. 2 Complex 1a disrupts the VHL–HIF-1α interaction by selectively engaging VHL. a** Complex **1a** inhibits the interaction of VHL–HIF-1α *in cellulo*. HEK293 cells were treated with vehicle DMSO (1% for 2 h), **1a** (0.3, 1, 3, 6 and 9 μM for 2 h), VH298 (100 μM for 2 h), MG132 (20 μM for 3 h), or FG-4592 (100 μM for 2 h) before lysis. Cell lysates were collected and incubated with VHL antibody overnight at 4 °C. The proteins were immunoprecipitated using agarose beads. The levels of hydroxy-HIF-1α co-precipitated with VHL were detected using a hydroxy-HIF-1α antibody. **b** Cellular thermal shift assays to monitor cellular target engagement of VHL and HIF-1α. HEK293 cell lysates were treated with **1a** (3 μM) at room temperature for 30 min and then heated at different temperatures ranging from 48 to 68 °C for 5 min. The protein samples were collected and detected by Western blotting using either VHL or PHD2 antibodies. **c** Fluorescence-based protein thermal shift assay of VBC in the presence or absence of VH298 (100 μM) and complex **1a** (100 μM). **d** ITC titration of complex **1a** (300 μM) into recombinant VHL protein (30 μM). ITC experiments were carried out in a MicroCal PEAQ-ITC isothermal titration calorimeter (Malvern Panalytical). **e** A competitive fluorescence polarization binding assay was performed to evaluate displacement of a fluorescent peptide (FAM-DEALAHyp-YIPD) binding to VBC ($K_d = 421.50 \pm 65.23$ nM) by complex **1a**. The $IC_{50}$ of complex **1a** ($K_d = 2.06$ μM) was determined to be 3.79 μM in the presence of 125 nM the fluorescent peptide and 450 nM VBC using a four-parameter logistic equation. **f** BLI kinetic analysis of the interaction between complex **1a** and VBC. VBC was surface-immobilized to Ni-NTA biosensors. BLI sensorgrams showing the binding of complex **1a** to surface-immobilized VBC. The $K_d$ values for a 1:1 interaction were calculated from the kinetic fit ($K_d = 6.74 \pm 0.19$ μM) and steady-state fit ($K_d = 5.80 \pm 0.64$ μM), respectively. The Ni-NTA biosensor tips coated with His-tagged VBC were dipped in increasing concentrations of **1a** (0.78, 1.56, 3.13, 6.25, and 12.5 μM) to measure the binding affinity of **1a** to VBC ($k_{on} = 5.22 \times 10^2$ M$^{-1}$ s$^{-1}$) and subsequently moved to wells containing buffer to measure dissociation rates ($k_{off} = 3.52 \times 10^{-2}$ s$^{-1}$). Data are expressed as means ± SD ($n = 3$ independent experiments for figures **a** and **b**, samples for figures **c** and **e**), P values were calculated using a two-sided *t*-test. *$P < 0.05$, **$P < 0.01$ vs. DMSO group. Source data are provided as a Source Data file.

which revealed that **1a** bound to the immobilized recombinant His-tagged VBC protein complex with $K_d$ values of $6.74 \pm 0.19$ μM (kinetic fit) and $5.80 \pm 0.64$ μM (steady-state fit). These biophysical experiments are in good agreement for the $K_d$ and stoichiometry for **1a** with VBC. Taken together, these data demonstrate that **1a** is able to bind to the VBC complex and displace a high-affinity HIF-1α peptide from VBC.

Finally, we showed that complex **1a** was unlikely to activate HIF signaling via oxygen depletion, PHD inhibition, and/or proteasomal inhibition (Supplementary Fig. 7). Similarly, complex **1a** had no effect on the level and activity of HIF-2α, which is essential in activating the COX-2 signaling axis in cancer cells (Supplementary Fig. 8)[37,38]. These results suggest that complex **1a** selectively targets VHL and inhibits the VHL–HIF-1α interaction.

**Complex 1a up-regulates HIF-1α target gene products *in cellulo*.** Previous studies have demonstrated that inhibition of the

interaction between VHL and HIF-1α results in an accumulation of HIF-1α and the up-regulation of HIF-1α target gene products, including VEGF[39], GLUT1[40], and EPO[41]. Therefore, we investigated the effect of complex **1a** on HIF-1α and HIF-1α target protein levels *in cellulo*. After incubation of HEK293 cells with complex **1a** for 2 h, the expression levels of HIF-1α, VEGF, GLUT1 and EPO were increased in a dose-dependent manner (Fig. 3a). This can be attributed at least in part to the disruption of the VHL–HIF-1α interaction by complex **1a**.

To confirm that the increase of HIF-1α was due to the inhibition of VHL activity, clear cell renal cell carcinoma A498 cells, which lack functional VHL, were treated with **1a** or VH298, a potent inhibitor of the VHL–HIF-1α PPI[21]. The PHD inhibitor FG-4592 was also used as a reference compound. As expected, no increase in HIF-1α levels was observed in A498 cells treated with **1a** or VH298, whereas HEK293 cells showed a clear accumulation of HIF-1α in the presence of VHL inhibitors (Fig. 3b). These data provide evidence for the stabilization of HIF-1α by **1a** via binding

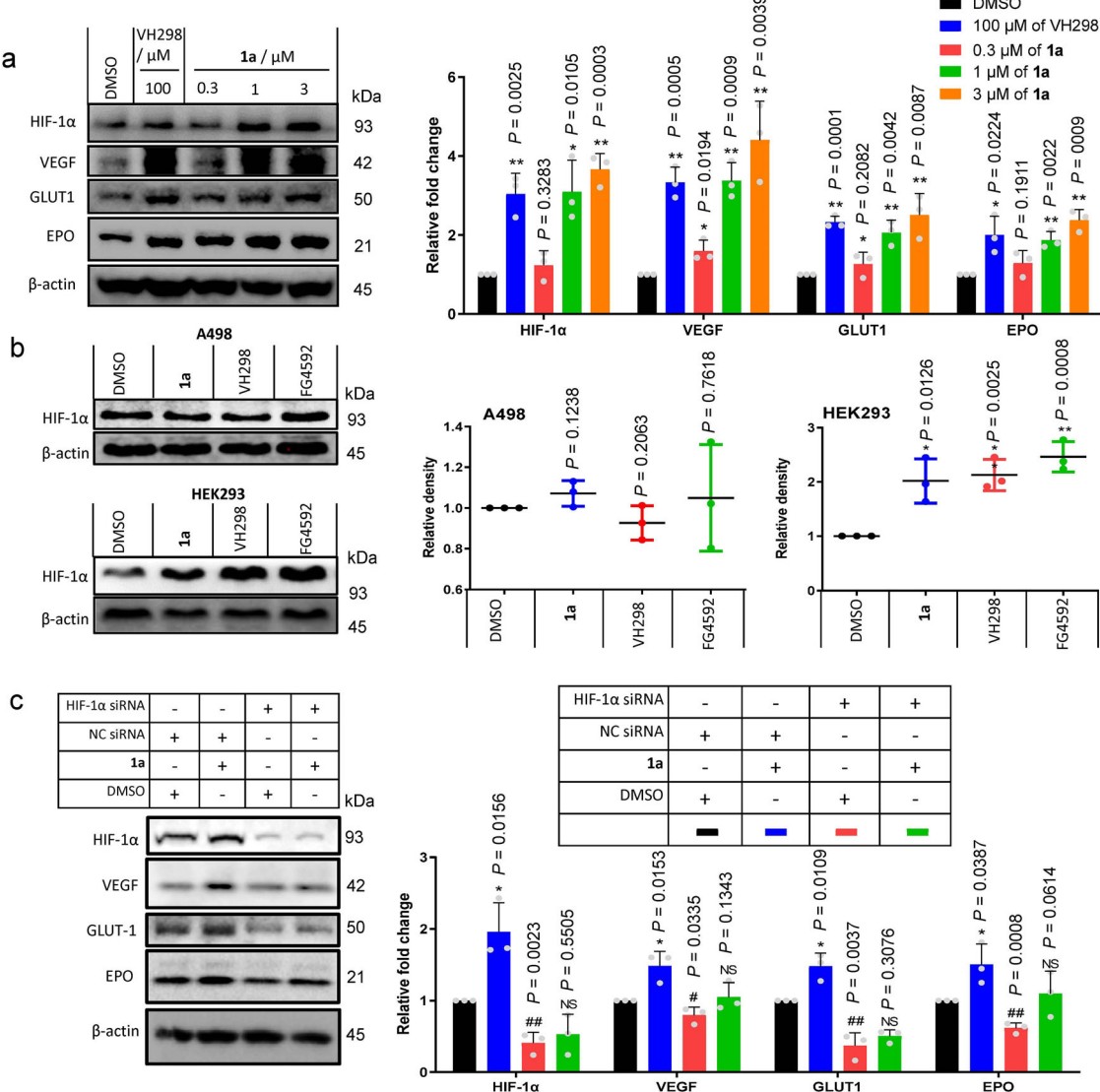

**Fig. 3 Effect of complex 1a on HIF-1α target gene products *in cellulo*. a** Effect of complex **1a** on specific proteins (GLUT1, VEGF, and EPO) mediated by HIF-1α in HEK293 cells. HEK293 cells were treated with **1a** for 2 h. Cell lysates were collected and analyzed by Western blotting. **b** Complex **1a** has no effect on HIF-1α accumulation in a VHL null cancer cell line. A498 (VHL-null), or HEK293 cells were treated with 1% DMSO, **1a** (3 μM), VH298 (100 μM), and FG-4592 (100 μM) for 2 h before lysis. **c** Effect of complex **1a** on EPO, GLUT1, and VEGF levels in HEK293 cells with or without HIF-1α knockdown. Left: The expression levels of EPO, GLUT1, and VEGF in HEK293 cells with or without knockdown HIF-1α in the presence or absence of **1a**. Right: Densitometry analysis of EPO, GLUT1, and VEGF levels on the Western blot. HIF-1α siRNA (sense, 5′-CUGAUGACCAGCAACUUGA-3′, antisense, 5′-UCAAGUUG CUGGUCAUCAG-3′). Negative control (NC) siRNA (sense, 5′-UAGCGACUAAACACAUCAA-3′, antisense, 5′-UUGAUGUGUUUAGUCGCUA-3′). Data are expressed as means ± SD ($n = 3$ independent experiments), *P* values were calculated using a two-sided *t*-test. #*P* < 0.05, ##*P* < 0.01 HIF-1α siRNA vs. NC siRNA, \**P* < 0.05, \*\**P* < 0.01 **1a** vs. vehicle/DMSO group, respectively. Source data are provided as a Source Data file.

to VHL and antagonizing the interaction of VHL–HIF-1α *in cellulo*.

A knockdown assay was performed to further verify the on-target effect of complex **1a**. EPO, GLUT1, and VEGF are target genes regulated by HIF-1α. As expected, knockdown of HIF-1α in HEK293 cells using siHIF-1α reduced the accumulation of EPO, GLUT1, and VEGF compared with control cells (Fig. 3c). Importantly, when compared its effect in normal cells, **1a** was less able to increase the accumulation of EPO, GLUT1, and VEGF in HIF-1α knockdown cells. This provides evidence that **1a** acts via a HIF-1α-dependent pathway in order to exert its on-target effects in HEK293 cells. However, as only partial effects of HIF-1α knockdown are observed, we do not rule out the possibility that other pathways could be involved in mediating the on-target

effects of complex **1a**. Taken together, the above results suggest that complex **1a** can be potentially developed as a drug candidate for the treatment of human diseases related to impaired angiogenesis or wound healing, resulting from effective and specific blockade of the VHL–HIF-1α interaction.

**Complex 1a accelerates wound healing in diabetic mice.** Inspired by the in vitro results, the effect of complex **1a** on wound healing in vivo was investigated in *db/db*, STZ, and HFD/ STZ diabetic mice. The *db/db* and age-matched wild-type (WT) mice were locally administered with vehicle (0.8% w/v Carbopol 974P NF in distilled water, pH 7.0) and 0.25 mg/mL complex **1a** (mixed in 0.8% w/v Carbopol 974P NF in distilled water) every other day for 8 days, respectively (Fig. 4a). The local application

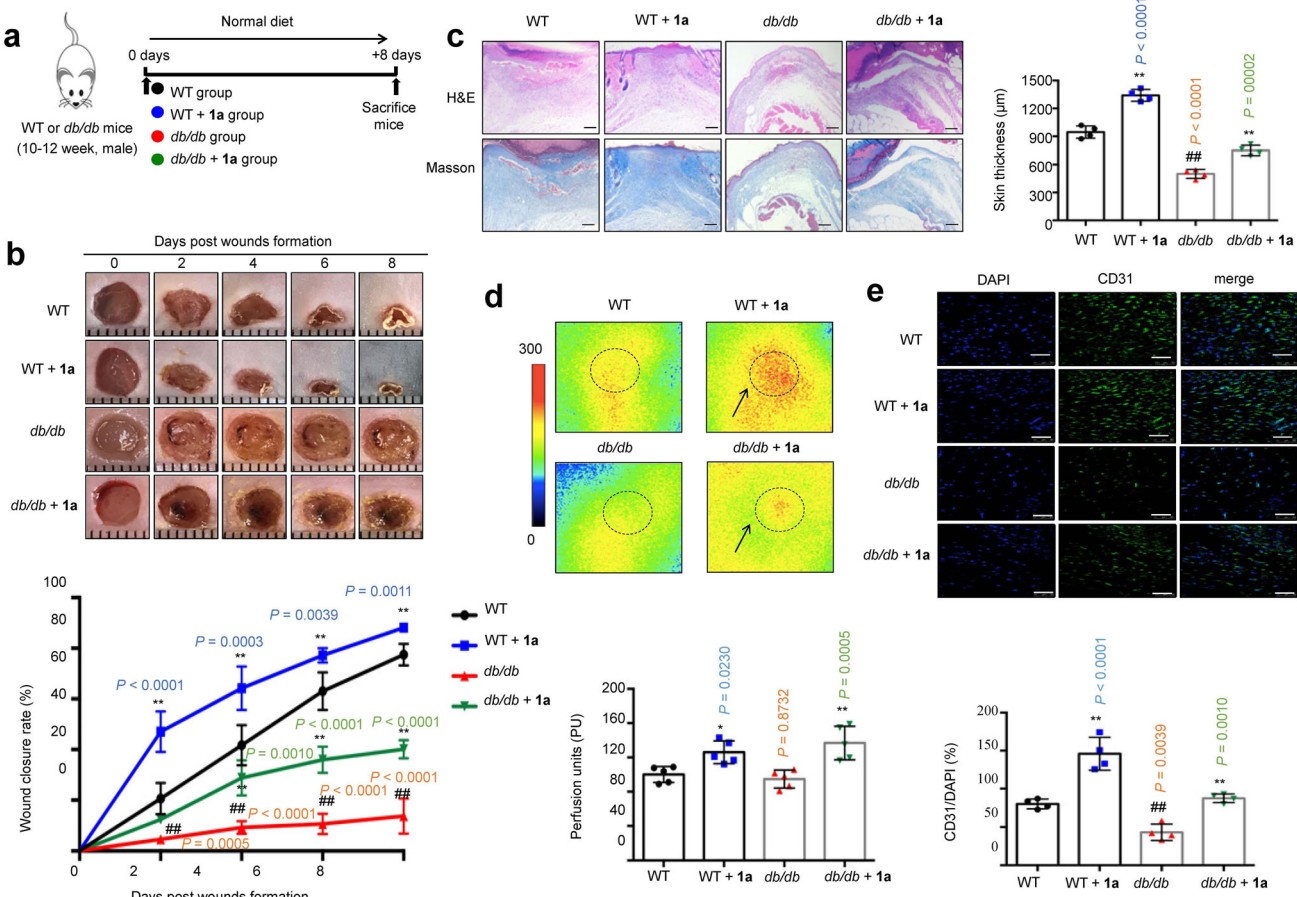

**Fig. 4 Complex 1a (0.25 mg/mL) accelerates wound closure in *db/db* mice. a** Timeline for in vivo experiments. **b** Image of representative wound (left) and wound closure rate (right) (*n* = 5 mice). **c** H&E and Masson's trichrome staining of dorsal skin section and skin thickness from the top of the epidermis to the bottom of the dermis in mice after 8 days post-injury (*n* = 4 mice). Scale bar = 200 μm. **d** Laser doppler imager in dorsal skin: representative images were shown for each group (left) and baseline perfusion on back skin of mice (right) after 2 days post-injury (dotted line circle represents the wound bed in mice of each group and arrow represents perfusion intensity (*n* = 5 mice). **e** CD31 and DAPI double staining in wound bed of mice after 8 days post-injury (*n* = 4 mice). Scale bar = 50 μm. Data are expressed as means ± SD, *P* values were calculated using a one-way ANOVA with Tukey's multiple comparison test. #*P* < 0.05, ##*P* < 0.01 WT vs. *db/db*, \**P* < 0.05, \*\**P* < 0.01 **1a** vs. vehicle.

of complex **1a** did not affect body weight during the experimental period, for both WT and *db/db* mice (Supplementary Fig. 9). STZ mice were obtained by a single injection of high-dose streptozotocin (150 mg/kg) (Supplementary Fig. 10a)[42]. HFD/STZ mice were generated by 8 weeks HFD feeding, followed by low-dose streptozotocin injection for 7 days (40 mg/kg/day) (Supplementary Fig. 11a)[43]. 3 days after streptozotocin injection, STZ or HFD/STZ mice with fasting blood glucose levels between 15 and 28 mmol/L were considered as diabetic mice and used in wound healing experiments. STZ and HFD/STZ mice were intraperitoneally injected with either vehicle (PEG 400:distilled water = 6:4, v/v) or 1.25 mg/kg complex **1a** every other day for 8 days, respectively. Inductively coupled plasma mass spectrometry analysis confirmed the presence of iridium in skin samples of dosed mice from STZ and HFD/STZ mice, demonstrating that complex **1a** could reach the target area (Supplementary Fig. 12). In both HFD/STZ and STZ models, neither diabetic nor normal control mice (NC) showed obvious changes in blood glucose levels (Supplementary Fig. 13a and b) or body weight (Supplementary Fig. 13c and d) after exposure to complex **1a**.

In the wound healing experiment, two full-thickness skin lesions were excised in the interscapular area of each mouse, and the wound area was monitored every other day. In the WT/NC

groups, the rates of wound closure were approximately 21%/22%, 42%/41%, 63%/57%, and 77%/74% after 2 days, 4 days, 6 days, and 8 days post-injury, respectively (Fig. 4b, Supplementary Figs. 10b and 11b). Treatment of complex **1a** in WT/NC mice accelerated wound closure compared to the vehicle group, approaching about 64%/65% of closure after 4 days and almost complete wound closure by 8 days (Fig. 4b, Supplementary Figs. 10b and 11b). The wound closure rates of the diabetic mice group were lower than those of the WT/NC group on corresponding days; only about 14%, 62% and 44% of the wound was healed after 8 days post-injury for *db/db*, HFD/STZ and STZ mice, respectively, compared to 77%/74% for WT/NC mice (Fig. 4b, Supplementary Figs. 10b and 11b). As expected, complex **1a** accelerated wound closure in all three diabetic mouse models. The rates of wound closure in **1a**-treated *db/db* mice were about 28% after 4 days and 40% after 8 days post-injury (cf. 9% and 14% in untreated *db/db* mice, respectively) (Fig. 4b); the rates of wound closure in **1a**-treated HFD/STZ mice were 62% after 4 days and 82% after 8 days post-injury (cf. 30% and 62% in untreated HFD/STZ mice, respectively) (Supplementary Fig. 11b); and the rates of wound closure in **1a** treated STZ mice were 50% after 4 days and 76% after 8 days post-injury (cf. 24% and 44% in untreated STZ mice, respectively) (Supplementary Fig. 10b).

Taken together, these results indicate that complex **1a** could accelerate wound healing in both normal and diabetic mice, with a greater effect being observed in the diabetic group.

The epithelial thickness of the regenerated skin in each group was compared using H&E staining and Masson's trichrome staining. Encouragingly, in both normal and diabetic mice groups, complex **1a** increased skin thickness after 8 days post-injury (Fig. 4c, Supplementary Figs. 10c and 11c), and also enhanced collagen deposition in wound areas (Fig. 4c, Supplementary Figs. 10c and 11c). One of the key processes related to wound healing is tissue angiogenesis. Skin perfusion pressure tests indicated that complex **1a** remarkably increased skin blood flow rate after 2 days post-injury in both normal and diabetic mice (Fig. 4d, Supplementary Figs. 10d and 11d). Moreover, CD31 immunostaining images showed that complex **1a** significantly enhanced microvessel density in the wound areas in both normal and diabetic groups (Fig. 4e, Supplementary Figs. 10e and 11e). Taken together, these results indicate that complex **1a** is effective at both increasing wound healing and angiogenesis in vivo, in both normal and diabetic mice.

The expression of HIF-1α, VEGF, GLUT1, and EPO was significantly increased in the wound tissue of **1a**-treated WT/NC, *db/db*, HFD/STZ and STZ groups at 8 days post-wounding, compared with the vehicle mice as shown by Western blotting (Fig. 5a, Supplementary Figs. 14a and 15a). The expression levels of HIF-1α and its target genes were significantly lower in the diabetic group compared to the WT/NC group, demonstrating that HIF-1α signal transduction was impaired (Fig. 5a, Supplementary Figs. 14a and 15a)[44]. Moreover, HIF-1α target genes essential for wound healing cell motility (i.e. *HSP-90*), angiogenesis (i.e. *VEGFR-1*), and recruitment of CAG (i.e. *SDF-1*, *SCF*, and *Tie-2*) were also increased in the wound tissue of **1a**-treated diabetic mice as revealed by qRT-PCR, with weaker effects being observed in the WT/NC mice (Fig. 5b, Supplementary Figs. 14b and 15b, Supplementary Table S2). Taken together, these results provide a mechanistic basis for the observed enhancement of wound healing and angiogenesis induced by complex **1a** in diabetic mice.

## Discussion

World diabetes patients are expected to reach 400 million by 2030[10]. Wound healing complications, leading to foot ulcers or even amputation, are a major factor contributing to diabetes-induced mortality[45]. Therefore, novel drugs are needed for promoting wound healing, particularly in diabetic patients. HIF-1α is critical in wound healing because it plays a key role in regulating vital processes involved in tissue repair[16,46]. PHD inhibitors have been previously developed as HIF-1α stabilizers[17]. Several PHDs inhibitors have been introduced into clinical trials for anemia or ischemia, including BAY-853934, FG-4592, FG-2216 and GSK1278863. Of these, FG-4592 has entered phase 3 trials to treat anemia in patients with chronic kidney disease[17]. The effectiveness of PHDs inhibitors in clinical trials demonstrates that HIF-1α signaling can serve as a drug target for angiogenesis-related diseases[16]. However, there are a few disadvantages to PHDs inhibitors, such as poor target selectivity and adverse side effects[47]. For example, during a phase 2 trial of FG-2216, many patients exhibited abnormal liver enzyme test results and one patient developed fatal hepatic necrosis[18]. Previous studies have shown that hyperglycemia destabilizes HIF-1α and impairs its function in diabetic mice through a VHL-dependent mechanism[16]. Thus, stabilizing HIF-1α via blocking the downstream interaction of HIF-1α and VHL is a potentially superior strategy to inhibiting upstream PHDs, in order to avoid HIF-independent off-target effects as has been observed with PHD

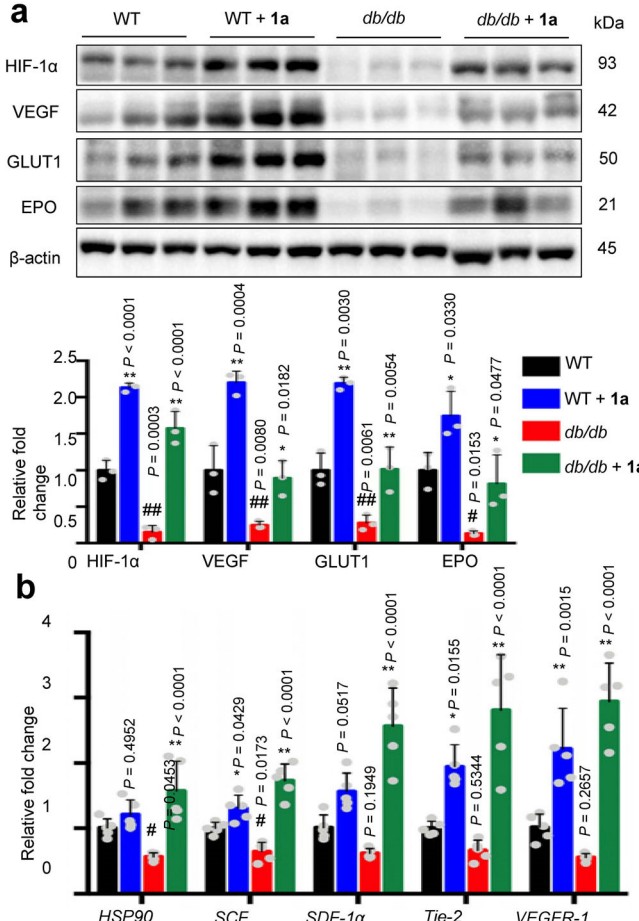

**Fig. 5 Complex 1a (0.25 mg/mL) activates gene expression regulated by HIF-1α in *db/db* mice at 8 days post-injury. a** Western blot analyses and quantitation of HIF-1α, VEGF, GLUT1, and EPO in wound tissue ($n = 3$ mice). All proteins were normalized by β-actin. **b** The mRNA levels of HIF-1α target genes involved in wound healing were analyzed by qRT-PCR in wound tissues ($n = 5$ mice). Data are expressed as means ± SD, $P$ values were calculated using a one-way ANOVA with Tukey's multiple comparison test. #$P < 0.05$, ##$P < 0.01$ WT vs. *db/db*, *$P < 0.05$, **$P < 0.01$ **1a** vs. vehicle. Source data are provided as a Source Data file.

inhibitors in clinical development. In this context, interrupting the VHL–HIF-1α PPI may be a highly effective strategy for the treatment of diabetic wounds[21,35,48]. Until now, few inhibitors of the interaction between VHL and HIF-1α have been discovered[21,22,33,35,49,50], and only VH298 has been reported to promote enthesis healing and wound healing in vivo[51,52].

Recently, Group 9 metal complexes have emerged as versatile scaffolds for drug discovery or bioanalysis due to their easily adjustable reactivity (from liable to inert), high aqueous solubility, inertness to air and synthetic accessibility[31,53]. The unique but adaptable properties of metal compounds, including their electronic properties, chemical reactivities, and molecular architectures, enable them to adopt three-dimensional geometries that can probe hitherto unexplored hotspots of the binding site of proteins. In this study, we report the Ir(III) complex **1a** as the first metal-based inhibitor of the VHL–HIF-1α interaction. Complex **1a** was identified after two rounds of screening, allowing preliminary structure–activity relationship (SAR) trends to be drawn for this series of compounds. In the first round of screening, the Rh(III) complex **1**, containing two brpy C^N ligands and a 4,7-dmeophen N^N ligand, displayed the highest activation of HIF-1α transcriptional activity (Fig. 1a and b). As no other complexes

in the first round of screening contain bromine groups, this could suggest the importance of the bromine group is activating HIF-1α activity. Complex **9**, containing fluorine groups, showed low activity, indicating that fluorine is less preferred than bromine for potency. Moreover, complexes **6** and **7** bearing C^N ligands derived from 2-phenylquinoline (phq) also showed low activity, suggesting that C^N ligands based on 2-phenyl pyridine (ppy) are more favored. After complex **1**, the Rh(III) complex **5** containing the 4,7-diphenyl-1,10-phenanthroline (dpphen) N^N ligand was the next most potent candidate. This suggests that conjugating phenyl groups to the 1,10-phenanthroline (phen) N^N ligand could also be somewhat tolerated, although the methoxy groups of 4,7-dmophen (as in complex **1**) were still preferred for activating HIF-1α transcriptional activity.

A focused library of 14 cyclometalated Rh(III) and Ir(III) complexes (**1a–1n**) containing different C^N or N^N donor ligands was subsequently synthesized based on the structure of the lead complex **1** (Fig. 1c). Analysis of the data from the second round of screening (Fig. 1d) revealed additional SAR trends. The most potent complex **1a**, bearing containing two 2-phenylpyridine (ppy) C^N ligands and the dmeophen N^N ligand, is substantially more active than the Rh(III) congener **1b**, indicating that the Ir(III) metal center is very important for the biological potency of **1a**. In contrast, the Ir(III) congener (**1c**) of the original parent Rh(III) complex **1** showed drastically reduced activity, indicating that for this combination of C^N and N^N ligands, the Rh(III) center is greatly preferred. Upon comparing the Ir(III) complexes **1c** and **1a**, it can be seen that changing the C^N ligand from brpy (as in **1c**) to ppy (as in **1a**) vastly improves the activity of the complex. After **1a**, the second-most potent compound was the Ir(III) complex **1g**, which differs from **1a** only by the presence of an additional methyl group at the 8-position of the ppy C^N ligand. However, adding a methyl group to the 6-position of the ppy ligand (as in **1e**) substantially weakens the activity of the complex, indicating that steric factors are highly important for the interaction of the complexes with the target. Upon comparing the N^N ligands, it can be seen that the dmeophen ligand of the parent complex still remains the preferred scaffold. For example, changing from **1a** to **1h**, which has the 4,4′-dimethoxy-2,2′-bipyridine (dmobpy) ligand rather than dmeophen, led to a decrease in activity, indicating that the additional fused phenyl ring in **1a** is important for biological potency. Replacing the two methoxy groups of **1a** with hydrogen groups (as in **1i** or **1j**), bromine groups (as in **1k** or **1l**) or phenyl groups (as in **1m** or **1n**) also led to drastically reduced activity for both the Rh(III) and Ir(III) congeners, respectively. This indicates that the biological activity of the complexes is highly sensitive to the nature of the substituents on the N^N ligand.

Complex **1a** selectively targeted VHL in vitro as revealed by CETSA and biophysical experiments (including ITC, FP, and BLI), leading to the disruption of the VHL–HIF-1α interaction *in cellulo* as shown by co-IP, and the stimulation of HIF-1α-directed signaling as revealed using the DLR assay. Moreover, complex **1a** effectively up-regulated HIF-1α target gene products *in cellulo*. HIF-1α levels are decreased under hyperglycemia[16]. In murine diabetes models, complex **1a** induced the accumulation of HIF-1α and the activation of HIF-1α-driven genes that are important for angiogenesis, including VEGFR, SDF-1α and SCF. Importantly, complex **1a** significantly improved wound healing and angiogenesis in diabetic mice. This study, therefore, demonstrates the validity of inhibiting the VHL–HIF-1α interaction as a therapeutic avenue for diabetic wound healing via both topical and intraperitoneal delivery routes.

VH298 is the most effective VHL–HIF-1α PPI inhibitor described to date, and has ca. 100-fold stronger binding affinity to VHL than complex **1a**[21–23]. VH298 engages with high affinity

and specificity with VHL as its only major cellular target, leading to selective on-target accumulation of hydroxylated HIF-α in a concentration-dependent and time-dependent fashion in different cell lines[21]. As an alternative scaffold class, the metal complex **1a** has shown promising wound healing results in in vivo models of diabetes. In this study, HFD/STZ mice were prepared through a HFD with subsequent multiple injections of a low dose of STZ. The key advantage of this model is to mimic the slow pathogenesis of type 2 diabetes that occurs in humans, encompassing the slow development from adult-onset diet-induced obesity to glucose intolerance, insulin resistance, the resulting compensatory insulin release and finally STZ-induced partial β-cell death. In this type 2 diabetes model, complex **1a** showed promising wound healing activity when administered via intraperitoneal injection. Additionally, the topical application of complex **1a** was performed on another widely used diabetic model, *db/db* mice, to confirm that our compound could also promote wound closure when applied externally.

While the binding affinity of complex **1a** is around 100-fold lower than that of the existing VHL–HIF-1α inhibitor VH298, we have demonstrated here that administering complex **1a** at higher dosages (over 30-fold higher compared to the dosage of VH298 used in a previous study[52]) and through various routes can lead to a significant accumulation of complex **1a** at injured skin tissue and promising wound healing effects in animal models of diabetes (including *db/db*, HFD/STZ and STZ) without significant toxicity. Hence, this report provides an additional scaffold for the development of wound healing therapeutics and also validates the feasibility of VHL–HIF-1α inhibitors on treating diabetic wounds through different routes of drug administrations, including intraperitoneal injection and topical application. However, it should be noted that the *db/db*, HFD/STZ and STZ models used in this study each have their own limitations and cannot fully mimic human diabetes[54,55]. Thus, further research is still needed in order to validate VHL–HIF-1α inhibition as a target for wound healing in clinical practice.

## Methods

**General synthesis of [M₂(C^N)₄Cl₂] complexes**. Cyclometalated dichloro-bridged dimers of the general formula $[M_2(C^N)_4Cl_2]$, where M = Ir(III)/Rh(III), were synthesized according to a literature method[56]. In brief, $MCl_3 \cdot 3H_2O$ was heated to 130 °C with 2.2 equivalents of cyclometallated C^N ligands in 3:1 methoxymethanol and deionized water under a nitrogen atmosphere for 12 h. The reaction was cooled to room temperature, and the product was filtered and washed with three portions of deionized water and then three portions of ether (3 × 50 mL) to yield the corresponding dimer.

**General synthesis of [M(C^N)₂(N^N)]PF₆ complexes**. These complexes were synthesized using a modified literature method[56–58]. Briefly, a suspension of $[M_2(C^N)_4Cl_2]$ (0.2 mM) and corresponding N^N (0.44 mM) ligands in a mixture of dichloromethane:methanol (1:1, 20 mL) was refluxed overnight under a nitrogen atmosphere. The resulting solution was allowed to cool to room temperature and was filtered to remove the unreacted cyclometalated dimer. To the filtrate, an aqueous solution of ammonium hexafluorophosphate (excess) was added and the filtrate was reduced in volume by rotary evaporation until precipitation of the crude product occurred. The precipitate was then filtered and washed with several portions of water (2 × 50 mL) followed by diethyl ether (2 × 50 mL). The product was recrystallized by acetonitrile:diethyl ether vapor diffusion to yield the titled compound. Complexes **1–14** and **1a–1m** were characterized by $^1$H-NMR, $^{13}$C-NMR, high-resolution mass spectrometry (HRMS) and elemental analysis.

**Transient transfection[59]**. HEK293 cells were seeded in six-well plates 24 h before transfection. HRE-luciferase plasmid (4 μg) and pRL-TK plasmid (4 μg) and TurboFect reagent (6 μL) were mixed together in a serum-free DMEM medium and the resulting solution was incubated for 20 min at room temperature. The mixture was added dropwise to the HEK293 cells in the wells. The cells were incubated for 32 h at 37 °C in a $CO_2$ incubator before use.

**Dual-luciferase reporter assay[59]**. The inhibition of HIF-1α activity was assayed by a reporter assay using a DLR assay system (Promega, Madison, WI, USA). Transiently transfected cells were treated with complexes or **P1** for 8 h before

measurement. Luciferase activity was integrated over a 10 s period and measured using a spectrophotometer (Spectra-max M5, Molecular Devices, USA). The results were standardized with the activity of Renilla luciferase. All data are expressed as means ± SD.

**Co-immunoprecipitation**[60]. The inhibition of VHL–HIF-1α interactions was investigated using a co-IP assay following the manufacturer's instructions. Briefly, HEK293 cells ($1 \times 10^6$ cells/well) were treated with indicated concentrations of complex **1a**, **P1** (30 μM) or DMSO for 2 h. After cell lysis and protein lysate separation, 100 μg of total protein was incubated with VHL antibody (1:1000; GTX101087, GeneTex) at 4 °C overnight. The proteins were immunoprecipitated using agarose beads. The levels of co-precipitated HIF-1α-OH were visualized using ECL Western Blotting Detection Reagent (GE Healthcare).

**Western blotting**[59]. After electrophoresis of protein samples (30 μg of total protein) on SDS–PAGE gels, the samples were transferred to a PVDF membrane and incubated at room temperature with a blocking solution for 1 h. The membrane was treated with primary antibodies and incubated overnight at 4 °C. After 1 h incubation with the secondary antibodies (Supplementary Table S1), protein bands were visualized using ECL Western Blotting Detection Reagent (GE Healthcare).

**Cellular thermal shift assay**[61]. CETSA was performed to monitor the target engagement of **1a** in HEK293 cell lysates. Briefly, $1 \times 10^6$ HEK293 cells were lysed and lysates were collected, diluted in PBS and separated into aliquots. Each aliquot was treated with **1a** (10 μM) or DMSO. 30 min after incubation at room temperature, the complex-treated lysates were divided into 50 μL in each of PCR tubes and heated individually at different temperatures (Veriti thermal cycler, Applied Biosystems/Life Technologies). The heated lysates were centrifuged and the supernatants were analyzed on SDS–PAGE followed by immunoblotting analysis by probing with antibodies.

**Isothermal titration calorimetry**. ITC experiments were carried in a MicroCal PEAQ-ITC isothermal titration calorimeter (Malvern Panalytical)[22]. Briefly, complex **1a** and recombinant VBC complex were dialyzed into the ITC buffer (20 mM Bis–Tris, 150 mM NaCl, 2 mM DTT, 1% DMSO) overnight. Complex **1a** (300 μM) was titrated against 30 μM VBC complex, consisting of 19 injections of 2 μL complex **1a** solution at a rate of 2 s/μL at 150 s time intervals. An initial injection of ligand (0.4 μL) was made and discarded during data analysis. The experiment was carried out at 25 °C while stirring at 750 rpm. The generated data were fitted to a single binding site model using the Setup MicroCal PEAQ-ITC Analysis Software provided by the manufacturer. Three control titrations, in which (1) **1a** is titrated into VBC buffer; (2) VBC buffer is titrated into VBC complex; (3) VBC buffer is titrated into VBC buffer, were also analyzed by using a composite model.

**Biolayer interferometry**. The binding affinities of inhibitors to recombinant VBC were measured by BLI on an OctetRed 96 (Fortebio). Ni-NTA biosensors were loaded with 25 μg/mL His-tagged VBC in BLI kinetics buffer (PBS buffer containing 0.02% Tween 20 and 0.1% BSA) with a binding value of ca. 5.5 nm, washed in the same buffer and transferred to wells containing complex **1a** or HIF-1α peptide at indicated concentrations in the same buffer. The Ni-NTA biosensor tips coated with His-tagged protein complex were dipped in increasing concentrations of **1a** or HIF-1α for 300 s and subsequently dissociated in the wells containing buffer for another 300 s. Negative control performed with BLI kinetics buffer against Ni-NTA biosensors was subtracted from the sample response against VBC-loaded Ni-NTA biosensors. The equilibrium dissociation constant ($K_d$) value for a 1:1 interaction was calculated from the kinetic fit and steady-state fit, respectively. The $K_d$ and associated standard errors were calculated using Octet analysis software.

**Knockdown assay**. HEK293 cells were seeded in the six-well plate at 80% confluence in DMEM medium for 24 h. Lipofectamine 3000 reagent and siRNA were gently mixed and incubated for 15 min at room temperature. Remove growth medium from cells and replace with 0.5 mL of fresh medium. Then the mixture of 500 μL was added to each well. Cells were incubated at 37 °C in a $CO_2$ incubator for 72 h post-transfection before further research.

**Animal experimental**[42,62]. Male *db/db* mice were purchased from the Model Animal Research Center of Nanjing University (Nanjing, China). Male C57BL/6J mice were purchased from the animal facility of the Faculty of Health Sciences, University of Macau. All mice were housed in the animal facility of the University of Macau, maintained at 23 ± 1 °C (50 ± 5% relative humidity) with 12 h light/dark cycles with free access to water and a regular chow diet. For the *db/db* model, 10–12 weeks old WT and *db/db* mice were randomly divided into two groups, respectively. The vehicle and **1a** groups were locally treated with vehicle (0.8% w/v Carbopol 974P NF in distilled water, pH 7.0, Chineway, Shanghai, China) or 0.25 mg/mL complex **1a** (mixed in 0.8% w/v Carbopol 974P NF in distilled water)

every other day for 8 days. For the STZ model, 10–12 weeks old mice were randomly divided into two groups. One group of mice were intraperitoneally injected with streptozocin (150 mg/kg body weight, 0.1 M citrate buffer, pH 4.5, Sigma-Aldrich, St. Louis, MO, USA) to induce diabetes, and the other group of mice was intraperitoneally injected with the same volume of citrate buffer (NC). The mice were maintained for 3 days. After fasted for 6 h, the blood glucose levels were measured by a One-Touch Ultra glucometer (Lifescan, Milpitas, CA, USA). For the HFD/STZ model, 6-8 weeks old mice were randomly divided into two groups, fed a regular chow diet (NC) or HFD (60% calories from fat, Trophic Animal Feed High-Tech Co., Nantong, Jiangsu, China) for 8 weeks. Then, the HFD-fed mice were received a daily intraperitoneal injection of streptozocin (40 mg/kg) for 7 days. For STZ and HFD/STZ models, the mice with blood glucose between 15.0 and 28.0 mmol/L, accompanied by manifestations of polydipsia, polyuria and polyphagia, were considered to be diabetic mice for the following experiments. The NC, STZ and HFD/STZ mice were randomly allocated into two groups, respectively. The vehicle and **1a** groups were intraperitoneally injected with the vehicle (PEG 400:distilled water = 6:4, v/v) and 1.25 mg/kg complex **1a** every other day for 8 days. All animal experiments were approved by the Animal Ethical and Welfare Committee of the University of Macau (No. ICMS-AEC-2014-06). All experiments complied with all relevant ethical regulations.

**Skin wound model**[16]. Mice were anesthetized by inhalation of 3% isoflurane. Prior to excision for wounds, dorsal hair was shaved with an electric clipper followed by a depilatory cream. The skin was rinsed with alcohol and two full-thickness wounds extending through the panniculus carnosus were created on the dorsum on each side of midline, using a 6-mm biopsy punch. Digital photographs were recorded on the day of surgery and every other day post-injury. A circular reference was placed alongside to permit correction for the distance between the camera and the animals. Wound area was quantitated using ImageJ (National Institutes of Health); wound closure rates were calculated as the following formulation: (wound area on day 0−wound area on day X)/wound area on day 0 × 100%.

**Histological analysis and microvessel density assay**[43]. After fixation in 4% paraformaldehyde, the skin samples were embedded in paraffin and sectioned (5 μm). For histological evaluation, sections were deparaffinized and rehydrated followed by hematoxylin and eosin (H&E) and Masson's trichrome staining. For immunohistochemical staining of CD31, the wound tissue sections were deparaffinized and stained with CD31 antibody (1:1000; ABclonal, Cambridge, MA, USA). The slides were examined under ×400 magnification to identify the area with the highest vascular density, and five randomly high-power field areas of the highest microvessel density were selected for each section. The average was calculated as the microvessel density of this sample.

**Statistical analysis**. Data were analyzed using GraphPad Prism 6.0 software. All experimental data were presented as mean ± SD (standard deviation), and each experiment has performed a minimum of three times. Significant differences between groups were determined using a one-way analysis of variance (ANOVA) unless otherwise noted. $P < 0.05$ was considered statistically significant throughout the study.

**Reporting summary**. Further information on research design is available in the Nature Research Reporting Summary linked to this article.

## Data availability
The data that support the findings of this study are available from the corresponding author upon reasonable request. Source data are provided with this paper.

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

## Acknowledgements

We would like to thank Prof. Simon Ming-Yuen Lee, Dr. Rui-Bing Wang, Dr. Hiu-Yee Kwan, Dr. Li Wang and Mr. Rui-Hong Chen for technical assistance. We also acknowledged the animal facility of the Faculty of Health Sciences, University of Macau. This work is supported by the Health and Medical Research Fund (HMRF/14150561), the National Natural Science Foundation of China (22077109, 21775131 and 81872754), the Hong Kong Baptist University Century Club Sponsorship Scheme 2020, the Interdisciplinary Research Matching Scheme (RC-IRMS/16-17/03), the Inter-disciplinary Research Clusters Matching Scheme (RC-IRCs/17-18/03), SKLEBA and HKBU Strategic Development Fund (SKLP_1920_P02), the Science and Technology Development Fund, Macao SAR (0072/2018/A2 and 0031/2019/A1), the University of Macau (MYRG2019-00002-ICMS, MYRG2018-00037-ICMS and MYRG2018-00187-ICMS).

## Author contributions

G.L., C.Y., and G.-J.Y. carried out the in vitro and *in cellulo* experiments. C.-N.K. and W.W. carried out synthesis and optimization of metal complexes. D.L. performed the in vivo experiments. G.L., C.-N.K., D.L. and C.Y. performed the data analysis and wrote the manuscript. C.D.P., V.K.W.W. and Y.X. analyzed the results. L.L., D.-L.M. and C.-H.L. designed the experiments and analyzed the results.

## Competing interests

The authors declare no competing interests.
