## [Peer Review File · Nature Communications]

Reviewers' Comments:

Reviewer #1:

Remarks to the Author:

Review of Yang et al. manuscript for Nat. Comm.

This manuscript reports an Iridium complex (compound 1a) as inducer of HIF-1alpha activity. The main claim of the paper is that the compound works by binding to VHL and competitively disrupts the well-known interaction between VHL and HIF-1alpha – which would lead to stabilization of HIF-1alpha and upregulation of HIF-dependent transcriptional activity. Indeed, a VHL inhibitor (VH298) active in cells via this exact mechanism was previously reported and characterized in a recent article published in the same journal (Frost et al. Nat. Comm. 2016, 7: 13312).

The authors' claim is based on experimental evidence provided on Figure 5, from co-IP and CETSA experiments. The evidence gathered to support such a claim is insufficient, inappropriate and wholly unconvincing, for the reasons explained below, thus raising severe reservations and doubts about the validity of the claim. There are several key experiments that should absolutely be performed to provide evidence to support or refute the claim of whether the compound binds to VHL and block the VHL-HIF interaction. I strongly suspect that the compound is merely a HIF stabilizer, via an unknown mechanism that has got nothing to do with VHL. There are many examples of compounds that stabilize HIF in cells, indeed via a wide variety of mechanisms (iron chelation, oxygen depletion, PHD inhibition, proteasomal inhibition and so on), and the reported compound 1a could act via any of those mechanisms. There are also metal complexes, such as CoCl₂, that act as HIF stabilizers and thus hypoxia mimetic in cells – via mechanism that remain unclear. Until this issue is resolved, the claim that 1a is a VHL-HIF inhibitor will remain unsubstantiated, and studying the effect of compound 1a in vivo seems far-fetched and unwarranted, and could lead to no more information than assessing the effect of CoCl₂ in mice!

In light of these serious and substantive concerns, my strong recommendation is that the manuscript in this current form should be precluded from publication - not only in this journal but in any reputable peer-reviewed scientific journal.

Provided below are constructive criticism and suggestions of crucial experiments that will help the authors in their future work.

1) The cellular HRE luciferase assays reports on HIF transcriptional activity. It does not report on blockade of the VHL-HIF interaction. Increase in cellular HIF activity can be the result of any other mechanism, as noted above.

2) I have serious issues with the co-IP data and CETSA data reported in Fig. 5. Specificity of bands detected using antibodies on western blots should be validated for any target using si-RNA or other means of target knockout/knockdown. It is very surprising that the authors detect a band for HIF-1alpha under normoxic conditions for the control DMSO treatments (Fig. 5a left band for input HIF; Fig. 5b bands corresponding to the "minus 1a" treatments). HIF-1alpha levels are barely detectable under normal conditions in cancer cells such as the used HEK-293. For this reason, it does not make sense that the author to monitor HIF-1alpha levels in CETSA! For the same reason, immunoprecipitating HIF-1alpha makes no sense at all, as HIF-1alpha levels will be highly variable between DMSO and compound treatments. The authors should instead perform the IP in the reversed fashion – i.e. IP'ing with VHL antibody, and monitoring levels of co-IP'ed HIF-1alpha (hydroxylated form) as a result of different treatments (e.g. proteasome inhibitor and VH298 inhibitor as positive controls; PHD inhibitor as negative control). It is surprising and concerning that the authors have not used any solid positive and negative controls in their cellular experiments. Strongly recommended for future work will be to use MG132/bortezomid as proteasome inhibitor, VH298 as VHL inhibitor and FG-4592 as PHD inhibitor. The authors should also use an antibody specific to the hydroxylated form of HIF-1alpha to distinguish whether the

stabilized HIF is hydroxylated or not.

In any cases, aside of all due reservations outlined above, the effects reported in both the co-IP and CETSA blots presented are less than ideal (unreliable?) and difficult to interpret (barely detectable) - and no quantification is included.

3) If the compound were really acting as inhibitor of the VHL-HIF interaction, their binding to VHL and ability to block the VHL-HIF interaction should be observed in vitro against recombinant proteins, using biophysical assays widely reported in the literature (see Frost et al. Nat Comm 2016, PMID: 27811928; van Molle et al. Chem Biol 2012, PMID: 23102223). Useful controls in cells to resolve whether the compound act on-target against VHL (or not) would be to use a VHL null cancer cell line such as a renal cancer cell lines and compare the cellular effect of the compounds in this cellular background with that in cells containing functional VHL.

4) The authors barely acknowledge the existence of a well-characterized and validated VHL inhibitor: compound VH298. The compound is available under no restriction of use from commercial vendors (Tocris) and should have been included as positive control (instead of much less characterized compound P1). The paper describing VH298 is briefly mentioned (cited as Ref. 12) and incorrectly (and questionably!) referred to swiftly when describing PHD inhibitors. VH298 and other analogues are now used world-wide by academic institutes and pharma industry, both in their own right as VHL inhibitors as well as for incorporation into conjugates. There is no question that VH298 should have been front and centre and a direct comparison included. If I were cynical, I would say they were trying to brush it under the carpet...

5) Another question that the authors may wish to address in future work is the effect of their compounds on the expression and activity of HIF-2alpha.

Reviewer #2:

Remarks to the Author:

This is an interesting and highly relevant manuscript. The authors use a systematic approach to identify new compounds that could improve wound healing in diabetic ulcers, specifically focusing on antagonizing the interaction between HIF-1a and VHL. They identified a complex (1a) which dose-dependently reduced the amount of VHL co-precipitating with HIF-1a and increased the expression of HIF-1a, VEGF, GLUT1 and EPO. These proteins were also shown to be increased in wound tissue by complex 1a 8 days after wounding of mice. The strengths of the paper are that it is methodologically sound, presents both cellular and in vivo data and touches on a highly relevant area with potential clinical applications, although the road to translation is still long. The weaknesses of the paper are that it is to a large extent observational and lacks full support for the molecular mechanism of action and that the connection with the human diabetic situation is fairly weak.

Specific comments:

1. Why were Ir(III)/Rh(III) complexes chosen? Please motivate this starting point better in the Results part.
2. Please report statistical comparisons in figure 4 between compound 1a and compound 1 and P1.
3. It is not clear whether Fig. 5C is based on only one experiment. If so, it needs to be repeated so that n=3 at least. The DMSO curve is rather different between the left and right graph, which is unexpected, and may by chance exaggerate the differences if this is only one experiment each.
4. In Figure 6, what happens if 1a is added in the absence of VRI? It would be interesting to see the effect of 1a on control cells alone.
5. The in vivo model is a short-term model of diabetes. The lack of good complication models is well known, but it should be acknowledged in the text that it is very difficult to directly draw parallels of the findings from the STZ mouse, with a brief period of diabetes, with the human situation of years/decades with the disease. This model is a reflection of the acute detrimental

effects of hyperglycemia on wound healing rather than diabetic complications.

6. The authors overstate the case when asserting that they provide a mechanistic explanation. The observations that VEGF, EPO and GLUT1 as well as other HIF1a target genes increase is interesting but only observational. The authors should, at least in cell studies, investigate the effect of the compound after knock-down of HIF1a to exclude that the compound has effects that are independent of HIF1a. It would also be of interest to assess the relative contribution of the different pathways of HIF1a-induced genes to the effects by e.g. knocking down VEGF, EPO and GLUT1 and investigate the resultant effect of the compound in zebra fish.

7. I lack a rational motivation for why antagonists of the interaction between HIF-1 α and VHL would be more harmless than PHDs which have been clinically tested. Is this just an optimistic anticipation, or can the authors support their strategy with literature evidence that this route is likely to avoid the side effects associated with PHDs?

Reviewer #3:

Remarks to the Author:

Yang et al-reported a new substance that inhibits the VHL-HIF1 α interaction with potential positive effect on angiogenesis and on wound healing in diabetes. The work is interesting but it needs further work to strengthen the conclusions of the manuscript.

Comments

The logic behind the choice of the library should be clarified.

What are the thoughts of the authors about the mechanisms of actions of complex 1a for angiogenesis in zebrafish since they still observe it after blocking the VEGFR2 receptor?

The local application of complex 1a is needed for evaluate the local effect of the compound on wound healing and angiogenesis

Fig 5a: the output of IP after exposure to 3 microM Complex 1a looks to be higher that after exposure to 1 microM

Fig 6a. it is not clear that VEGF and EPO are induced in dose dependent by complex 1a

Figure 7: the staining should be quantified. The CD31 seems to reflect an increase in cellularity (as shown by DAPI)

Fig. 8d is missing

GAPDH is not an appropriate control since it is regulated by HIF.

The quantification of the WB analysis should be presented.

The quantification of the angiogenesis in zebra fishes is needed

Referee Report 1

This manuscript reports an Iridium complex (compound 1a) as inducer of HIF-1 α activity. The main claim of the paper is that the compound works by binding to VHL and competitively disrupts the well-known interaction between VHL and HIF-1 α – which would lead to stabilization of HIF-1 α and upregulation of HIF-dependent transcriptional activity. Indeed, a VHL inhibitor (VH298) active in cells via this exact mechanism was previously reported and characterized in a recent article published in the same journal (Frost et al. Nat. Comm. 2016, 7: 13312).

The authors' claim is based on experimental evidence provided on Figure 5, from co-IP and CETSA experiments. The evidence gathered to support such a claim is insufficient, inappropriate and wholly unconvincing, for the reasons explained below, thus raising severe reservations and doubts about the validity of the claim. There are several key experiments that should absolutely be performed to provide evidence to support or refute the claim of whether the compound binds to VHL and block the VHL-HIF interaction. I strongly suspect that the compound is merely a HIF stabilizer, via an unknown mechanism that has got nothing to do with VHL. There are many examples of compounds that stabilize HIF in cells, indeed via a wide variety of mechanisms (iron chelation, oxygen depletion, PHD inhibition, proteasomal inhibition and so on), and the reported compound 1a could act via any of those mechanisms. There are also metal complexes, such as CoCl₂, that act as HIF stabilizers and thus hypoxia mimetic in cells – via mechanism that remain unclear. Until this issue is resolved, the claim that 1a is a VHL-HIF inhibitor will remain unsubstantiated, and studying the effect of compound 1a in vivo seems far-fetched and unwarranted, and could lead to no more information that assessing the effect of CoCl₂ in mice!

In light of these serious and substantive concerns, my strong recommendation is that the manuscript in this current form should be precluded from publication - not only in this journal but in any reputable peer-reviewed scientific journal.

Provided below are constructive criticism and suggestions of crucial experiments that will help the authors in their future work.

***Comment 1:** The cellular HRE luciferase assays reports on HIF transcriptional activity. It does not report on blockade of the VHL-HIF interaction. Increase in cellular HIF activity can be the result of any other mechanism, as noted above.*

Our response: We thank the reviewer for the constructive suggestions. We have revised the relevant statements in the revised manuscript according to this suggestion, for example:

1. "Screening and structure-based optimization of small molecules as VHL–HIF-1 α PPI inhibitors" has been revised to "Screening and structure-based optimization of small molecules as potential HIF-1 α stabilizers by monitoring HRE-driven luciferase activity"
2. "This demonstrates the role of Ir(III) center in coordinating the structure of the entire complex in order to confer VHL–HIF-1 α inhibitory activity." has been

revised to "This demonstrates the role of Ir(III) center in coordinating the structure of the entire complex in order to confer HIF-1 α transcriptional activity."

Moreover, to evaluate the selectivity of **1a**, we have tested the effect of **1a** on proteasome activity in the revised manuscript. We found that **1a** had no significant effect on proteasomal activity as compared with MG132 treatment alone in HEK293 cell lysis samples (Supplementary Fig. 5a). We also measured the effect of **1a** on oxygen consumption in HEK293 living cells. Glucose oxidase (GO) was used as a reference for oxygen depletion. Compared with glucose oxidase, complex **1a** induced no significant increase in oxygen consumption (Supplementary Fig. 5b). Moreover, immunoblotting and the cellular thermal shift assay (CETSA) were performed to evaluate the effect of our complex on PHD2 and PHD3 levels or binding. As shown in Supplementary Figure 5c, **1a** had no significant effect on PHD2 and PHD3 levels. Of the three PHD isoforms (PHD1, 2 and 3), PHD2 is thought to be the key oxygen sensor in regulating HIF-1 α (*Oncogene*, **2017**, 36(3): 397; *The EMBO Journal*, **2003**, 22(16): 4082-4090). Thus, we also performed CETSA to verify the engagement between complex **1a** and VHL and PHD2. After incubating HEK293 cell lysates with complex **1a** followed by heating to set temperatures, an obvious shift of about 6 °C in the VHL melting curve was observed in the presence of complex **1a** (3 μ M). In contrast, complex **1a** had no appreciable effect on the thermal stabilization of PHD2 (Fig. 2c).

Taken together, these results suggest that complex **1a** does not stabilize HIF *via* modulating proteasome activity, oxygen consumption and/or PHD2/PHD3 inhibition, providing stronger support for our hypothesis that **1a** is a highly potent and selective inhibitor of VHL.

Supplementary Figure 5. Complex **1a** does not interfere with proteasome activity, oxygen consumption or PHD2/PHD3 levels. (a) A commercial fluorometric assay kit

was used to monitor the effect of **1a** on proteasome activity. Homogenized HEK293 lysis samples were prepared by using 0.5 % NP-40 lysis buffer in PBS. 50 μ L of cell extract was added to 50 μ L of assay buffer. 1 μ L of complex **1a** (0.1 and 0.3 mM) or proteasome inhibitor MG132 (1 mM) was added to one of the paired wells. The fluorescence intensity of the cells was measured at Ex/Em = 350/440 nm in a microplate reader after incubation at 37° C for 60 min in the dark. (b) Complex **1a** has no effect on the oxygen consumption in living HEK293 cells. HEK293 cells were seeded in a black bottom 96-well tissue culture plate and incubated overnight. After treatment with complex **1a** for 2 h, MitoXpress Xtra solution was added. 100 μ L of HS Mineral Oil was added and the fluorescence intensity was monitored immediately at Ex/Em = 380/650 nm in a microplate reader. Glucose oxidase (GO) was used as a reference for oxygen depletion. (c) Immunoblotting assay to monitor the effect of **1a** on PHD2 and PHD3 levels. HEK293 cells were incubated with the indicated concentration of complex **1a** (1 and 3 μ M) for 2 h. The protein samples were collected and detected by Western blotting using PHD2 or PHD3 antibodies. Data are expressed as means \pm SD (n = 3). *P* values were calculated using a two-sided t-test. ***P* < 0.01 vs. DMSO group. NS (not significant, *P* > 0.05) vs. DMSO group.

Figure 2b | Cellular thermal shift assays to monitor cellular target engagement of VHL and HIF-1 α . HEK293 cell lysates were treated with **1a** (3 μ M) at room temperature for 30 min and then heated at different temperature ranging from 48 °C to 68 °C for 5 min. The protein samples were collected and detected by Western blotting using either VHL or PHD2 antibodies. Data are expressed as means \pm SD (n = 3). *P* values were calculated using a two-sided t-test. **P* < 0.05, ***P* < 0.01 vs. DMSO group.

Comment 2: I have serious issues with the co-IP data and CETSA data reported in Fig. 5. Specificity of bands detected using antibodies on western blots should be validated for any target using si-RNA or other means of target knockout/knockdown. It is very surprising that the authors detect a band for HIF-1 α under normoxic conditions for the control DMSO treatments (Fig. 5a left band for input HIF; Fig. 5b bands corresponding to the “minus 1a” treatments). HIF-1 α levels are barely detectable under normal conditions in cancer cells such as the used HEK-293. For

this reason, it does not make sense that the author to monitor HIF-1alpha levels in CETSA! For the same reason, immunoprecipitating HIF-1alpha makes no sense at all, as HIF-1alpha levels will be highly variable between DMSO and compound treatments. The authors should instead perform the IP in the reversed fashion – i.e. IP'ing with VHL antibody, and monitoring levels of co-IP'ed HIF-1alpha (hydroxylated form) as a result of different treatments (e.g. proteasome inhibitor and VH298 inhibitor as positive controls; PHD inhibitor as negative control). It is surprising and concerning that the authors have not used any solid positive and negative controls in their cellular experiments. Strongly recommended for future work will be to use MG132/bortezomid as proteasome inhibitor, VH298 as VHL inhibitor and FG-4592 as PHD inhibitor. The authors should also use an antibody specific to the hydroxylated form of HIF-1alpha to distinguish whether the stabilized HIF is hydroxylated or not.

In any cases, aside of all due reservations outlined above, the effects reported in both the co-IP and CETSA blots presented are less than ideal (unreliable?) and difficult to interpret (barely detectable) - and no quantification is included.

Our response: We thank the reviewer for the constructive suggestion.

In our study, HIF-1 α levels could be detected under normoxic conditions for the control DMSO treatments. The intensity of the protein band is relative and could depend on many factors, including the exposure time, the type of antibody, operational differences, etc. In the literature, HIF-1 α has been detected under normal conditions in many cancer cell lines, including HEK-293 cells (*EMBO Reports*, **2014**, 15(1): 77-85; *Proceedings of the National Academy of Sciences*, **2012**, 109(49): E3367-E3376; *Journal of Medicinal Chemistry*, **2011**, 54(24): 8471-8489; *Blood*, **2004**, 103(3): 1124-1130; *Molecular and Cellular Biology*, **2003**, 23(24): 9361-9374).

We have repeated the co-IP assay using anti-VHL to monitor hydroxylated HIF-1 α in the presence of DMSO, complex **1a**, VH298, MG132 (proteasome inhibitor), or FG-4592 (PHD inhibitor). HEK293 cells were incubated with complex **1a** for 2 h, and cell lysates were subjected to co-IP using VHL antibody. The results showed that complex **1a** dose-dependently reduced the amount of HIF-1 α -OH co-precipitating with VHL, suggesting that it was able to inhibit the VHL–HIF-1 α interaction in the treated cells (Fig. 2a). We have also repeated the CETSA (Fig. 2b). All Western blotting bands have been quantified by densitometry in the revised manuscript.

Figure 2a and 2b. (a) Complex **1a** inhibit the interaction of VHL–HIF-1α *in cellulo*. HEK293 cells were treated with vehicle DMSO (1% for 2 h), **1a** (1 and 3 μM for 2 h), VH298 (100 μM for 2 h), MG132 (20 μM for 3 h), or FG-4592 (100 μM for 2 h) before lysis. Cell lysates were collected and incubated with VHL antibody overnight at 4 °C. The proteins were immunoprecipitated using agarose beads. The levels of hydroxy-HIF-1α co-precipitated with VHL were detected using a hydroxy-HIF-1α antibody, and then visualized using ECL Western Blotting Detection Reagent (GE Healthcare). (b) Cellular thermal shift assays to monitor cellular target engagement of VHL and HIF-1α. HEK293 cell lysates were treated with **1a** (3 μM) at room temperature for 30 min and then heated at different temperature ranging from 48 °C to 68 °C for 5 min. The protein samples were collected and detected by Western blotting using either VHL or PHD2 antibodies. Data are expressed as means ± SD (n = 3). *P* values were calculated using a two-sided t-test. **P* < 0.05, ***P* < 0.01 vs. DMSO group.

Comment 3: *If the compound were really acting as inhibitor of the VHL-HIF interaction, their binding to VHL and ability to block the VHL-HIF interaction should be observed in vitro against recombinant proteins, using biophysical assays widely reported in the literature (see Frost et al. Nat Comm 2016, PMID: 27811928; van Molle et al. Chem Biol 2012, PMID: 23102223). Useful controls in cells to resolve*

whether the compound act on-target against VHL (or not) would be to use a VHL null cancer cell line such as a renal cancer cell lines and compare the cellular effect of the compounds in this cellular background with that in cells containing functional VHL.

Our response: We thank the reviewer for the constructive suggestion. After consideration of the biophysical assays used in the literature, we have performed an ITC assay to confirm the ability of complex **1a** on binding to VHL. The results showed that complex **1a** binds to VHL with a K_d of $4.62 \pm 0.67 \mu\text{M}$ (Fig. 2c). Moreover, an immunoblotting assay was performed to detect the level of HIF-1 α in the renal cancer cell line (A498), which lacks functional VHL, under treatment with DMSO, **1a**, VH298, or FG-4592 (Fig. 3b). We found that the level of HIF-1 α did not increase significantly on **1a** treatment. In contrast, HEK293 cells expressing VHL showed accumulation of HIF-1 α in the presence of **1a**, suggesting the on-target specificity of **1a**. Taken together, these data provide evidence for the stabilization of HIF-1 α by **1a** via binding to VHL and antagonizing the interaction of VHL–HIF-1 α *in cellulo*.

Figure 2c | Complex 1a binds to VHL *in vitro*. ITC titration of recombinant VHL protein (180 μM) into complex **1a** (18 μM). ITC experiments were carried in a MicroCal PEAQ-ITC Isothermal Titration Calorimeter (Malvern Panalytical).

Figure 3b | Complex 1a has no effect on HIF-1 α accumulation in a VHL null cancer cell line. A498 (VHL-null) or HEK293 cells were treated with 1% DMSO, **1a**

(3 μ M), VH298 (100 μ M), and FG-4592 (100 μ M) for 2 h before lysis. Data are expressed as means \pm SD (n = 3). *P* values were calculated using a two-sided t-test. **P* < 0.05, ***P* < 0.01 vs. DMSO group.

Comment 4: *The authors barely acknowledge the existence of a well-characterized and validated VHL inhibitor: compound VH298. The compound is available under no restriction of use from commercial vendors (Tocris) and should have been included as positive control (instead of much less characterized compound P1). The paper describing VH298 is briefly mentioned (cited as Ref. 12) and incorrectly (and questionably!) referred to swiftly when describing PHD inhibitors. VH298 and other analogues are now used world-wide by academic institutes and pharma industry, both in their own right as VHL inhibitors as well as for incorporation into conjugates. There is no question that VH298 should have been front and centre and a direct comparison included. If I were cynical, I would say they were trying to brush it under the carpet...*

Our response: We thank the reviewer for the constructive suggestion. We have corrected the inappropriate description and citation of VH298 in the revised manuscript. Moreover, VH298 has now been used as a positive control compound in the experiments of the revised manuscript, including the co-immunoprecipitation (co-IP) experiment to evaluate the effect of **1a** on the VHL–HIF-1 α interaction, and the immunoblotting (IB) assay to evaluate the effect of **1a** on HIF-1 α target gene products *in cellulo*.

Comment 5: *Another question that the authors may wish to address in future work is the effect of their compounds on the expression and activity of HIF-2 α .*

Our response: We thank the reviewer for the constructive suggestion. We have performed a Western blotting assay to detect the level of HIF-2 α in HEK293 cells in the presence of DMSO, **1a**, and VH298, respectively. The levels of HIF-2 α were maintained even after treatment with **1a** or VH298 for 2 h (Supplementary Fig. 6a). HIF-2 α is essential in activating the COX-2 signaling axis in cancer cells, including in colon cancers (*Carcinogenesis*, **2012**, 34(1): 163-169; *Cell death & disease*, **2015**, 6(1): e1600). Therefore, we also detected the level of COX-2 in HEK293 cells (Supplementary Fig. 6b). After incubating HEK293 cells with complex **1a** (1 and 3 μ M) for 2 h, no significant change in COX-2 levels was observed. In contrast, In contrast, an obvious increase of COX-2 level was observed after MG132 (30 μ M) treatment. The results suggest that **1a** has no significant effect on the level and activity of HIF-2 α .

Supplementary Figure 6 | Effect of complex 1a on specific proteins mediated by HIF-2α. Effect of complex 1a on the levels of (a) HIF-2α and (b) COX-2 in HEK293 cells. HEK293 cells were treated with 1a for 2 h. Cell lysates were collected and analyzed by the Western blotting. Data are expressed as means ± SD (n = 3). *P* values were calculated using a two-sided t-test. **P* < 0.05 vs. DMSO group. NS (not significant, *P* > 0.05) vs. DMSO group.

Referee Report 2

This is an interesting and highly relevant manuscript. The authors use a systematic approach to identify new compounds that could improve wound healing in diabetic ulcers, specifically focusing on antagonizing the interaction between HIF-1α and VHL. They identified a complex (1a) which dose-dependently reduced the amount of VHL co-precipitating with HIF-1α and increased the expression of HIF-1α, VEGF, GLUT1 and EPO. These proteins were also shown to be increased in wound tissue by complex 1a 8 days after wounding of mice. The strengths of the paper are that it is methodologically sound, presents both cellular and in vivo data and touches on a highly relevant area with potential clinical applications, although the road to translation is still long. The weaknesses of the paper are that it is to a large extent observational and lacks full support for the molecular mechanism of action and that the connection with the human diabetic situation is fairly weak.

Comment 1: *Why were Ir(III)/Rh(III) complexes chosen? Please motivate this starting point better in the Results part.*

Our response: We thank the reviewer for the useful suggestions. The motivation of choosing Ir(III)/Rh(III) complexes has been added in the revised manuscript.

“Recently, Group 9 metal complexes have emerged as versatile scaffolds for drug discovery or bioanalysis due to their easily adjustable reactivity (from liable to inert), high aqueous solubility, inertness to air and synthetic accessibility^{23, 43}. The unique but adaptable properties of metal compounds, including their electronic properties, chemical reactivities, and molecular architectures, and can enable them to adopt three-

dimensional geometries that can probe hitherto unexplored hotspots of the binding site of proteins.”

References:

23. *Chem Sci* **6**, 871-884 (2015).

43. *Acc Chem Res* **47**, 3614-3631 (2014).

Comment 2: Please report statistical comparisons in figure 4 between compound 1a and compound 1 and P1.

Our response: We thank the reviewer for the constructive suggestion. We have included statistical comparisons between **P1**, **1** and **1a** in Figure 4 in the revised manuscript.

Figure 1d | Effect of complexes 1a–1n on HIF-1 α activation of the HRE-driven reporter. HEK293 cells were treated with complexes or **P1** for 8 h. Error bars represent the standard deviations (SD) of the results from three independent experiments. *P* values were calculated using a two-sided t-test. #*P* < 0.05 **1a** vs. **P1**, NS (not significant, *P* > 0.05) **1** vs. **P1**. **P* < 0.05, ***P* < 0.01 vs. DMSO group.

Comment 3: It is not clear whether Fig. 5C is based on only one experiment. If so, it needs to be repeated so that *n*=3 at least. The DMSO curve is rather different between the left and right graph, which is unexpected, and may by chance exaggerate the differences if this is only one experiment each.

Our response: We thank the reviewer for the constructive suggestion. The difference in the DMSO curves between the left and right graph could be due to the fact that different proteins have different intrinsic thermal stabilities (see *Science*, **2013**, 341(6141): 84-87; *Nature Communications*, **2016**, 7: 13312). We have added error bars to the revised Figure 2b based on three independent experiments, and performed statistical analysis. In the presence of complex **1a** (3 μ M), an obvious shift of about 6 $^{\circ}$ C in the VHL melting curve was observed. This result indicates that complex **1a**

could stabilize VHL inside cell lysates. In contrast, complex **1a** had no appreciable effect on the thermal stabilization of PHD2.

Figure 2b | Cellular thermal shift assays to monitor cellular target engagement of VHL and HIF-1 α . HEK293 cell lysates were treated with **1a** (3 μ M) at room temperature for 30 min and then heated at different temperature ranging from 48 $^{\circ}$ C to 68 $^{\circ}$ C for 5 min. The protein samples were collected and detected by Western blotting using either VHL or PHD2 antibodies. Data are expressed as means \pm SD (n = 3). *P* values were calculated using a two-sided t-test. **P* < 0.05, ***P* < 0.01 vs. DMSO group.

Comment 4: In Figure 6, what happens if **1a** is added in the absence of *VRI*? It would be interesting to see the effect of **1a** on control cells alone.

Our response: We thank the reviewer for the constructive suggestion. The original "Figure 6" described a zebrafish model for studying angiogenesis, which has now been removed from the updated manuscript. Previous research has showed that activated HIF-1 α could increase the transcription of over 100 downstream genes, such as EPO, GLUT1, VEGF and transforming growth factor beta3 (TGF- β 3), which regulate glucose metabolism, cell proliferation, migration and angiogenesis, in addition to other processes (*Acta Pharmaceutica Sinica B*, **2015**, 5(5): 378-389; *Proceedings of the National Academy of Sciences*, **2008**, 105(49): 19426-19431). Due to the wide role of HIF-1 α in wound healing, we decided to remove the zebrafish experiments which were limited mainly to angiogenesis. Instead, our main *in vivo* focus was on a diabetic mice model, which is a more relevant model for wound healing model in type 1/2 diabetes (*Diabetes*, **2004**, 53(suppl 3): S215-S219; *Current protocols in pharmacology*, **2015**, 70(1): 5.47. 1-5.47. 20).

Comment 5: The *in vivo* model is a short-term model of diabetes. The lack of good complication models is well known, but it should be acknowledged in the text that it is

very difficult to directly draw parallels of the findings from the STZ mouse, with a brief period of diabetes, with the human situation of years/decades with the disease. This model is a reflection of the acute detrimental effects of hyperglycemia on wound healing rather than diabetic complications.

Our response: We thank the reviewer for the constructive suggestion.

To simulate long-term diabetic complications, a type 2 diabetic mice model has been included in the revised manuscript (Fig. 4a). Type 2 diabetic mice (T2D) was generated by 8 weeks high-fat diet feeding, followed by low-dose STZ injection for 7 days. 3 days after STZ injection, the mice with fasting blood glucose levels between 15 and 28 mmol/l were considered as diabetic mice and used in wound healing experiments. In the wound healing experiment, two full-thickness skin lesions were excised in interscapular area of each mouse, and the wound area was monitored every other day.

Treatment of complex **1a** dramatically accelerated the wound healing in NC and T2D mice (Fig. 4b). In the NC group, the rates of wound closure were approximately 22%, 41%, 57%, and 74% after 2 days, 4 days, 6 days, and 8 days post-injury, respectively. The NC + **1a** group of mice exhibited more rapid wound closure compared to the NC group, approaching more than 65% of closure after 4 days and almost complete wound closure by 8 days (Fig. 4b). The wound closure rates of the diabetic mice group were lower than those of the NC group on corresponding days; only about 62% wound was healed after 8 days post-injury for T2D mice (Fig. 4b). As expected, the rate of wound closure in **1a** treated T2D mice was 62% after 4 days and 82% after 8 days post injury (Fig. 4b). Taken together, these results indicate that complex **1a** could stimulate wound healing in both normal and diabetic mice, with a greater effect being observed in the diabetic group.

The epithelial thickness of the regenerated skin in the different groups was also compared using H&E staining and Masson's trichrome staining. Encouragingly, in both normal and diabetic mice groups, complex **1a** increased skin thickness after 8 days post-injury (Fig. 4c), and also increased collagen deposition (Fig. 4c). One of the key processes related to wound healing is tissue angiogenesis. Skin perfusion pressure tests indicated that complex **1a** significantly increased skin blood flow rate after 2 days post-injury in both normal and diabetic mice (Fig. 4d). Moreover, CD31 immunostaining showed that complex **1a** significantly enhanced microvessel density in the wound areas in both normal and diabetic groups (Fig. 4e). Taken together, these results indicate that complex **1a** is effective at both increasing wound healing and angiogenesis *in vivo*, in normal and type 2 diabetic mice.

Figure 4 | Complex 1a (1.25 mg/kg) accelerates wound closure in type 2 diabetic mice. (a) Timeline for *in vivo* experiments. (b) Image of representative wound (left) and wound closure rate (right) ($n = 5$). (c) H&E and Masson's trichrome staining of dorsal skin section and skin thickness from the top of the epidermis to the bottom of the dermis in mice after 8 days post-injury ($n = 4$). Scale bar = 200 μm . (d) Laser doppler imager in dorsal skin: representative images were shown for each group (left) and baseline perfusion on back skin of mice (right) after 2 days post-injury (dotted line circle represents the wound bed in mice of each group and arrow represents perfusion intensity). ($n = 5$) (e) CD31 and DAPI double staining in the wound bed of mice after 8 days post-injury ($n = 4$). Scale bar = 50 μm . Data are expressed as means \pm SD. # $P < 0.05$, ### $P < 0.01$ T2D vs. NC, ** $P < 0.01$ 1a vs. vehicle.

Comment 6: The authors overstate the case when asserting that they provide a mechanistic explanation. The observations that VEGF, EPO and GLUT1 as well as other HIF1a target genes increase is interesting but only observational. The authors should, at least in cell studies, investigate the effect of the compound after knock-down of HIF1a to exclude that the compound has effects that are independent of HIF1a. It would also be of interest to assess the relative contribution of the different pathways of HIF1a-induced genes to the effects by e.g. knocking down VEGF, EPO and GLUT1 and investigate the resultant effect of the compound in zebra fish.

Our response: We thank the reviewer for the valuable suggestion.

A knockdown assay has been performed to investigate the effect of complex on the expression level of downstream genes (EPO, GLUT1, VEGF). As expected, knockdown of HIF-1 α reduced the accumulation of EPO, GLUT1, and VEGF

compared with control cells. Importantly, when compared to its effect in normal cells, **1a** was less able to increase the accumulation of EPO, GLUT1, and VEGF in HIF-1 α knockdown cells (Fig. 3c). This provides evidence that **1a** acts *via* accumulating HIF-1 α in order to exert its angiogenic effects in HEK293 cells.

Compared with the zebrafish model, the diabetic mice model was considered to be a more appropriate model to investigate the effect of **1a** on wound healing in type 1/2 diabetes. After careful consideration, we have removed zebrafish assays in the revised manuscript. Please refer to the response of comment 4 (Referee 2) for a detailed response.

Figure 3c | Effect of complex 1a on EPO, GLUT1, and VEGF levels in HEK293 cells with or without HIF-1 α knockdown. Left: The expression levels of EPO, GLUT1, and VEGF in HEK293 cells with or without knockdown HIF-1 α in the presence or absence of **1a**. Right: Densitometry analysis of EPO, GLUT1, and VEGF levels on the Western blot. HIF-1 α siRNA (sense, 5'-CUGAUGACCAGCAACUUGA-3', antisense, 5'-UCAAGUUGCUGGUCAUCAG-3'). Negative control (NC) siRNA (sense, 5'-UAGCGACUAAACACAUCA-3', antisense, 5'-UUGAUGUGUUUAGUCGCUA-3'). Data are expressed as means \pm SD (n = 3). *P* values were calculated using a two-sided t-test. #*P* < 0.05, ##*P* < 0.01 HIF-1 α siRNA vs. NC siRNA, **P* < 0.05, ***P* < 0.01 **1a** vs. vehicle, respectively.

Comment 7: *I lack a rational motivation for why antagonists of the interaction between HIF-1 α and VHL would be more harmless than PHDs which have been clinically tested. Is this just an optimistic anticipation, or can the authors support their strategy with literature evidence that this route is likely to avoid the side effects associated with PHDs?*

Our response: We thank the reviewer for the constructive suggestion. In the revised manuscript, we have included additional discussion with literature support to explain why antagonists of the interaction between HIF-1 α and VHL might be more harmless than PHD inhibitors.

“Previous studies have shown that hyperglycemia destabilizes HIF-1 α and impairs its function in diabetes mouse through a VHL-dependent mechanism¹¹. Thus, an

alternative strategy to stabilize HIF-1 α *via* blocking the downstream HIF-1 α and VHL interaction is potentially superior to inhibiting upstream PHDs, in order to avoid HIF-independent off-target effects as has been observed with PHD inhibitors in clinical development. In this context, interrupting the VHL–HIF-1 α protein-protein interaction may be a highly effective strategy for the treatment of diabetic wounds^{38, 39, 40}. Until now, very few inhibitors of the interaction between VHL and HIF-1 α have been discovered, and none of these have yet shown the ability to enhance wound healing *in vivo*^{25, 38, 41, 42}.”

References:

11. *Proc Natl Acad Sci U S A* **105**, 19426-19431 (2008).
25. *J Am Chem Soc* **134**, 4465-4468 (2012).
38. *Nat Commun* **7**, 13312 (2016).
39. *Expert Opin Investig Drugs* **20**, 645-656 (2011).
40. *Angew. Chem. Int. Ed.* **51**, 11463-11467 (2012).
41. *Chem Commun (Camb)* **52**, 12837-12840 (2016).
42. *PeerJ* **4**, e2757 (2016).

Referee Report 3

The work is interesting but it needs further work to strengthen the conclusions of the manuscript.

Comment 1: *The logic behind the choice of the library should be clarified.*

Our response: We thank the reviewer for the constructive suggestion. The library of 14 in-house Rh(III)/Ir(III) complexes was chosen based on the diversity of their C^N and N^N ligands. For example, ligands with different methylation patterns (e.g. ppy, eppy), regiochemistries (e.g. tpy, 3-mpy, 2-mpy), steric properties (e.g. phen, dpphen), electron-donating properties (e.g. dmeophen, dmobpy) and electron-withdrawing properties (e.g. dbrbpy, nitrophen) were employed in the first round of screening. Furthermore, we also aimed at choosing complexes with ligands that are commonly used in medicinal chemistry. For example, ligands were used that contain the bioactive diethyl methylphosphonate group (e.g. dphobpy) or methoxy group (e.g. dmobpy) (*Journal of medicinal chemistry*, **2007**, 50(15): 3434-3441; *Journal of medicinal chemistry*, **2012**, 55(22): 10282-10286; *Journal of Medicinal Chemistry*, **2013**, 56(22): 9089-9099). Based on the preliminary structural-activity relationships (SAR) identified from the first round of screening, we could develop a bioactive lead structure for the development of derivatives with higher potency. In the second round of screening, we aimed at further optimizing the structural and electronic effects of different C^N and N^N ligands. For example, C^N ligands with different methylation and positional effects (e.g. complexes **1a**, **1e** and **1g**), N^N ligands with different

steric effects (e.g. complexes **1j** and **1n**) and electronic effects (e.g. complexes **1j** and **1l**) were designed and synthesized according to the structure of the lead compound.

Comment 2: *What are the thoughts of the authors about the mechanisms of actions of complex 1a for angiogenesis in zebrafish since they still observe it after blocking the VEGFR2 receptor?*

Our response: We thank the reviewer for the constructive suggestion. Compared with the zebrafish model, the diabetic mice model was considered to be a more appropriate model to investigate the effect of **1a** on wound healing in type 1/2 diabetes. After careful consideration, we have removed zebrafish assays in the revised manuscript. Please refer to the response of comment 4 (Referee 2) for a detailed response.

Comment 3: *The local application of complex 1a is needed for evaluate the local effect of the compound on wound healing and angiogenesis?*

Our response: We thank the reviewer for the constructive suggestion. A type 1 diabetic mice model has been established for testing the local effect of complex **1a** on wound healing. Neither T1D diabetic nor normal control mice (NC) showed obvious changes in body weight (Supplementary Fig. 12a) or blood glucose levels (Supplementary Fig. 12b) during treatment of complex **1a**. In the wound healing experiment, two full-thickness skin lesions were excised in the interscapular area of each mouse, and the wound area was monitored every other day. Treatment of complex **1a** dramatically accelerated wound healing in NC and T1D mice (Supplementary Fig. 12c). In the NC group, the rates of wound closure were approximately 25%, 55%, 72%, and 84% after 2 days, 4 days, 6 days, and 8 days post-injury, respectively. The NC + **1a** group of mice exhibited more rapid wound closure compared to the NC group, approaching more than 72% of closure after 4 days (Supplementary Fig. 12c). For T1D mice, the wound closure rates were lower than those of the NC group on corresponding days; only about 41% wound was healed after 8 days post-injury (Supplementary Fig. 12c). As expected, the rate of wound closure in **1a** treated T1D mice was 41% after 4 days and 60% after 8 days post injury (Supplementary Fig. 12c). These results indicate that the local application of complex **1a** could also stimulate wound healing in both normal and diabetic mice.

Supplementary Figure 12. Local effects of complex **1a** on wound healing in NC and diabetic mice. (a) Body weight of NC and T1D mice. (b) Blood glucose levels of NC and T1D mice. (c) Image of representative wounds (left) and wound closure rate (right). Data are expressed as means \pm SD ($n = 5$). $###P < 0.01$ T1D vs. NC, $*P < 0.05$, $**P < 0.01$ **1a** vs. vehicle.

Comment 4: Fig 5a: the output of IP after exposure to 3 microM Complex 1a looks to be higher than after exposure to 1 microM.

Our response: We thank the reviewer for the constructive suggestion. We have repeated the co-IP assay using anti-VHL to monitor hydroxylated HIF-1 α in the presence of DMSO, complex **1a**, VH298, MG132, or FG-4592. HEK293 cells were incubated with complex **1a** for 2 h, and cell lysates were subjected to co-IP using HIF-1 α antibodies (Fig. 2a). The results showed that complex **1a** dose-dependently reduced the amount of HIF-1 α -OH co-precipitating with VHL, suggesting that it was able to inhibit the VHL-HIF-1 α interaction in the treated cells. All of Western blotting bands have been quantified in the revised manuscript.

Figure 2a | Complex 1a inhibits the interaction of VHL–HIF-1α in cellulo. HEK293 cells were treated with vehicle DMSO (1%), **1a** (1 and 3 μM), VH298 (100 μM), MG132 (20 μM), or FG-4592 (100 μM) for 2 h before lysis. Cell lysates were collected and incubated with VHL antibody overnight at 4 °C. The proteins were immunoprecipitated using agarose beads. The levels of hydroxy-HIF-1α co-precipitated with VHL were detected using a hydroxy-HIF-1α antibody, and then visualized using ECL Western Blotting Detection Reagent (GE Healthcare). Data are expressed as means ± SD ($n = 3$). P values were calculated using a two-sided t-test. * $P < 0.05$, ** $P < 0.01$ vs. DMSO group.

Comment 5: Fig 6a. it is not clear that VEGF and EPO are induced in dose dependent by complex 1a

Our response: We thank the reviewer for the useful suggestion. We have repeated the immunoblotting assay as described above in the revised manuscript. As shown in the Figure 3a, after incubation of HEK293 cells with complex **1a** for 2 h, the expression levels of HIF-1α, VEGF, GLUT1 and EPO were increased in a dose-dependent manner.

Figure 3a | Effect of complex 1a on specific proteins (GLUT1, VEGF, and EPO) mediated by HIF-1α in HEK293 cells. HEK293 cells were treated with **1a** for 2 h. Cell lysis were collected and analyzed by the Western blotting. Data are expressed as means ± SD ($n = 3$). P values were calculated using a two-sided t-test. * $P < 0.05$, ** $P < 0.01$ vs. DMSO group.

Comment 6: Figure 7: the staining should be quantified. The CD31 seems to reflect an increase in cellularity (as shown by DAPI)

Our response: We thank the reviewer for the constructive suggestion. The quantification of the staining analysis has been performed in the revised manuscript. The results demonstrated that complex **1a** significantly enhanced CD31 immunostaining density in the wound areas in both normal and diabetic groups (Fig. 4e and and Supplementary Fig. 10d).

Figure 4e. CD31 and DAPI double staining in the wound bed of mice after 8 days post-injury ($n = 4$). Scale bar = 50 μm . Data are expressed as means \pm SD. # $P < 0.05$ T2D vs. NC, ** $P < 0.01$ 1a vs. vehicle.

Supplementary Figure 10d. CD31 and DAPI double staining in the wound bed of mice after 8 days post-injury ($n = 4$). Scale bar = 50 μm . Data are expressed as means \pm SD. # $P < 0.05$ T1D vs. NC, ** $P < 0.01$ 1a vs. vehicle.

Comment 7: Fig. 8d is missing

Our response: We thank the reviewer for the constructive suggestion and apologise for the numbering mistake. The corrected figure number "Supplementary Fig. 11b" has been updated in the manuscript accordingly.

Comment 8: *GAPDH is not an appropriate control since it is regulated by HIF.*

Our response: We thank the reviewer for the useful suggestion. We have repeated all of the relevant assays using β -actin instead of GAPDH in the revised manuscript.

Comment 9: *The quantification of the WB analysis should be presented.*

Our response: We thank the reviewer for the useful suggestion. The quantification of the WB analysis has been performed in the revised manuscript.

Comment 10: *The quantification of the angiogenesis in zebra fishes is needed*

Our response: We thank the reviewer for the constructive suggestion. Compared with the zebrafish model, the diabetic mice model was considered to be a more appropriate model to investigate the effect of **1a** on wound healing in type 1/2 diabetes. After careful consideration, we have removed zebrafish assays in the revised manuscript. Please refer to the response of comment 4 (Referee 2) for a detailed response.

Reviewers' Comments:

Reviewer #1:

Remarks to the Author:

The authors have attempted to address the major concerns expressed at first review. They have included in Figure 2 new co-IP data, this time immunoprecipitating VHL and monitoring the level of pulled-down HIF-1 α -OH in various conditions – as well as new ITC data. The authors use the new data in support of their claim that “in vitro biophysical analyses demonstrated that the compound binds to VHL and inhibits the VHL-HIF interaction”. However, in my opinion, the new data is insufficient evidence to support such a claim because it is ambiguous and inconclusive, for the following reasons:

Re co-IP data: a) as proteasome blockade greatly enhances levels of HIF-1 α -OH a significant increase in HIF-1 α -OH pulled down by VHL would be expected to be detected in going from DMSO treated to MG132 treated. However this is not observed. b) It is difficult to assess specificity of the effect from just two concentrations of compounds. I also remain concerned that the bands detected by western blot have not been confirmed for identity using siRNA of the respective protein – which was a point raised in the previous report.

Re ITC data: an ITC titration of “recombinant VHL protein” (180 μ M) into 1a (18 μ M) is included in Figure 2c. However, the conditions (buffer, temperature and so on) for this experiment are not given. Most importantly, it is not specified how “recombinant VHL protein” was expressed and purified. Also the data is ambiguous as it appears that the n value (the point at which the curve reaches 50% saturation) appears to be at molar ratio around 0.3 – so at a point significantly lower than the expected value, should be around 1 .

In summary, in spite of the new data, the evidence provided remains inconclusive with regards to supporting direct binding to VHL. Moreover, crucially, no evidence is provided that could support the claim that 1a competes with HIF-1 α for binding to VHL. In the absence of any supporting biophysical data for such VHL-HIF1 α competition by 1a, and/or any co-crystal structure or allied structural/biophysical evidence for genuine binding interaction between 1a and VHL at the HIF binding site, compound 1a cannot be qualified as an inhibitor of the VHL-HIF interaction.

Reviewer #2:

Remarks to the Author:

The authors have responded to many of my concerns. (Please note that my focus area is diabetes pathophysiology and the paper has been reviewed with my glasses of expertise. I recognize that reviewer 1 has a number of concerns which seem substantial - it is beyond my expertise to fully evaluate whether the technical concerns raised by him/her has been appropriately met and I leave that at the discretion of the editor and reviewer 1).

Two concerns remain from my viewpoint:

1) In figure 3c the authors only see partial effects by HIF1 α -knockdown and it should therefore be acknowledged that other pathways could also be involved.

2) It is incorrect to use the terms type 1 and type 2 diabetes for mice. The mice studied in this paper do not have type 1 or type 2 diabetes - they have hyperglycemia induced in ways to mimic type 1 and type 2 diabetes, respectively. There is no really good rodent type 2 model. The HFD/STZ model used here and termed “type 2” has very different features compared to the human situation. The HFD used to induce insulin resistance is appropriate but the insulin secretion failure seen in human T2D is not caused by such a severe destruction as the STZ treatment models. It is rather a matter of long-term hyperinsulinemia which subsequently leads to loss of specialized function and/or cell death (the exact mechanisms are debated).

The model used here is not worse than many other models but it should not be called type 2 diabetes but a mimic of type 2 diabetes and its limitations should be clearly stated in the text. Moreover, the diabetic ulcers usually occur after years/decades and the very different timing in

this study compared to the real situation has to be acknowledged as it may limit the clinical usefulness of the findings.

Reviewer #3:

Remarks to the Author:

If the aim of the authors is to suggest a new therapeutically approach for diabetic wounds they should test the compound locally and not systemically.

The animal model used for "type 2 diabetes" is mainly a model for type 1 diabetes since the animals were injected with STZ. Other established models for wound healing impairment linked with type 2 diabetes are availabl.

Reviewers' comments:

Reviewer #1 (Remarks to the Author):

The authors have attempted to address the major concerns expressed at first review. They have included in Figure 2 new co-IP data, this time immunoprecipitating VHL and monitoring the level of pulled-down HIF-1 α -OH in various conditions – as well as new ITC data. The authors use the new data in support of their claim that “in vitro biophysical analyses demonstrated that the compound binds to VHL and inhibits the VHL-HIF interaction”. However, in my opinion, the new data is insufficient evidence to support such a claim because it is ambiguous and inconclusive, for the following reasons:

Comment 1: *Re co-IP data: a) as proteasome blockade greatly enhances levels of HIF-1 α -OH a significant increase in HIF-1 α -OH pulled down by VHL would be expected to be detected in going from DMSO treated to MG132 treated. However this is not observed. b) It is difficult to assess specificity of the effect from just two concentrations of compounds. I also remain concerned that the bands detected by western blot have not been confirmed for identity using siRNA of the respective protein – which was a point raised in the previous report.*

Our response: We thank the reviewer for the constructive suggestion. We have repeated the co-IP assay with new MG132 reagent (Fig. 2a). Notably, treatment with 20 μ M of MG132 for 3 h led to a significant increase in the HIF-1 α -OH level pulled down by VHL. The HIF-1 α -OH level with MG132 treatment is approximately 1.2 times higher than that with DMSO treatment, confirming that proteasome blockade could greatly enhance HIF-1 α -OH levels. Meanwhile, the results of co-IP assay with five concentrations of complex **1a** showed the concentration-dependent reduction of HIF-1 α -OH co-precipitating with VHL, demonstrating that **1a** was able to inhibit the VHL–HIF-1 α interaction in the treated cells. Reduced levels of HIF-1 α -OH were also after treatment with reported positive control compound VH298 and PHD inhibitor FG-4598 (Figure 2a).

In order to verify the specificity of HIF-1 α -OH and VHL antibodies, siRNA knockdown experiments were performed with HIF-1 α and VHL siRNA in HEK293 cells. As shown by Western blotting experiments, the levels of both HIF-1 α -OH (Supplementary Fig. 5a) and VHL (Supplementary Fig. 5b) were significantly

reduced in the presence of siRNA, providing evidence for the identity of the bands of HIF-1 α -OH and VHL in Western blotting experiments.

Figure 2a | Complex **1a** inhibits the interaction of VHL–HIF-1 α *in cellulo*. HEK293 cells were treated with vehicle DMSO (1% for 2 h), **1a** (0.3, 1, 3, 6 and 9 μ M for 2 h), VH298 (100 μ M for 2 h), MG132 (20 μ M for 3 h), or FG-4592 (100 μ M for 2 h) before lysis. Cell lysates were collected and incubated with VHL antibody overnight at 4 $^{\circ}$ C. The proteins were immunoprecipitated using agarose beads. The levels of hydroxy-HIF-1 α co-precipitated with VHL were detected using a hydroxy-HIF-1 α antibody, and then visualized using ECL Western Blotting Detection Reagent (GE Healthcare).

Supplementary Figure 5. Knockdown assay to verify the specificity of HIF-1 α -OH and VHL antibodies. (a) The expression levels of HIF-1 α -OH in HEK293 cells with or without knockdown HIF-1 α . (b) The expression levels of VHL in HEK293 cells with or without knockdown VHL. HIF-1 α siRNA (sense, 5'-CUGAUGACCAGCAACUUGA-3', antisense, 5'-UCAAGUUGCUGGUCAUCAG-3'), VHL siRNA (5'-ACACAGGAGCGCAUUGCACAU-3', antisense, 5'-AUGUGCAAUGCGCUCCUGUGU-3')^{1, 2} and NC siRNA (Negative control (NC) siRNA (sense, 5'-UAGCGACUAAACACAUCAA-3', antisense, 5'-UUGAUGUGUUUAGUCGCUA- 3').

Comment 2: *Re ITC data: an ITC titration of “recombinant VHL protein” (180 μ M) into 1a (18 μ M) is included in Figure 2c. However, the conditions (buffer, temperature and so on) for this experiment are not given. Most importantly, it is not specified how “recombinant VHL protein” was expressed and purified. Also the data is ambiguous as it appears that the n value (the point at which the curve reaches 50% saturation) appears to be at molar ratio around 0.3 – so at a point significantly lower than the expected value, should be around 1.*

Our response: We thank the reviewer for the constructive suggestion. We have added the VHL protein expression and purification methods and the conditions of the ITC assay in the revised manuscript, as follows:

"VHL protein expression and purification. The expression and purification of full-length human recombinant VHL protein (GST-tagged) was carried out as described previously^{1, 2}. VHL-pGex2TK was a gift from William Kaelin (Addgene plasmid # 20790)³. Briefly, *E. coli* DH5-Alpha was cultured in lysogeny broth (LB) medium with 100 μ g/mL ampicillin and grown at 37 °C until an OD₆₀₀ of 0.5-0.8 was reached. The cells were induced with 1 mM isopropyl- β -D-thiogalactopyranoside (IPTG) at 37 °C for 4 h. Cells were harvested by centrifugation at 4000 g for 20 min and then homogenized by sonication in buffer (50 mM Tris-HCl, 400 mM NaCl, 5% glycerol, pH 6.2). Precleared lysates were applied to GST GraviTrap columns (GE Healthcare), following the kit protocol. The protein was assayed by preparative SDS-PAGE using TGX and TGX Stain-Free FastCast acrylamide kits (Bio-Rad Laboratories GmbH, Munich, Germany).

Isothermal titration calorimetry. ITC experiments were carried in a MicroCal PEAQ-ITC Isothermal Titration Calorimeter (Malvern Panalytical), as previously described with minor modification⁴. Briefly, complex **1a** and recombinant VHL were dialyzed into the ITC buffer (20 mM Bis-Tris, 150 mM NaCl, 2 mM DTT, 5% DMSO) overnight. Complex **1a** (2 mM) was titrated against 200 μ M VHL protein, over 19 injections of 2 μ L complex **1a** solution at a rate of 2 sec/ μ L at 150 s time intervals. An initial injection of ligand (0.4 μ L) was made and discarded during data analysis. The experiment was carried out at 25 °C while stirring at 750 rpm. The generated data was analyzed using the Setup MicroCal PEAQ-ITC Analysis Software provided by the manufacturer."

To address the data ambiguity issue raised by reviewer 1, we have repeated the ITC experiment three times using new purified VHL protein (GST-tagged, full length human recombinant VHL protein). We found by ITC that complex **1a** bound to VHL *in vitro* with a K_d value of 6.27 μ M with N (sites) = 0.951 ± 0.015 , indicating the modest affinity of complex **1a** for VHL (Figure 2e).

Figure 2c | ITC titration of complex **1a** (2 mM) into recombinant VHL protein (200 μ M). ITC experiments were carried in a MicroCal PEAQ-ITC Isothermal Titration Calorimeter (Malvern Panalytical).

Comment 3: *In summary, in spite of the new data, the evidence provided remains inconclusive with regards to supporting direct binding to VHL. Moreover, crucially, no evidence is provided that could support the claim that 1a competes with HIF-1 α for binding to VHL. In the absence of any supporting biophysical data for such VHL-HIF1 α competition by 1a, and/or any co-crystal structure or allied structural/biophysical evidence for genuine binding interaction between 1a and VHL at the HIF binding site, compound 1a cannot be qualified as an inhibitor of the VHL-HIF interaction.*

Our response: We thank the reviewer for the constructive suggestion. We have investigated the ability of the complex **1a** to compete with HIF-1 α for binding to VHL using a competitive fluorescence polarization (FP) assay. In the assay, a labeled high-affinity HIF-1 α peptide, FAM-DEALAHyp-YIPD, was used to probe specific binding to the VHL-HIF-1 α interface by monitoring the displacement of ligand signals [*J. Med. Chem.* 2018, 61(16), 7387–7393; *Chem. Biol.* 2012, 19, 1300–1312; *J. Am. Chem. Soc.* 2012, 134, 4465–4468]. The polarization was measured in the presence of VHL and serial dilutions of complex **1a** in order to observe the displacement of the fluorescent peptide. From this assay, the IC₅₀ of complex **1a** was determined to be 14.65 μ M ($K_d = 7.11 \mu$ M) in the presence of 450 nM VHL and 125 nM fluorescent HIF-1 α peptide (Fig. 2d), demonstrating that complex **1a** could displace the high-affinity HIF-1 α peptide from VHL.

A surface plasmon resonance (SPR) assay showed that the HIF-1 α peptide bound to the immobilized recombinant VHL protein with a K_d value of 234 nM (Supplementary Figure 6). In addition, SPR was employed to characterize the binding kinetics of **1a** with VHL (Fig. 2e). As shown in Figure 2e, **1a** showed an association rate constant (k_{on}) of 6.33×10^3 (1/Ms) and a dissociation rate constant (k_{off}) of 7.14×10^{-2} (1/s) with immobilized VHL protein using a 1:1 binding model. Taken together, these data demonstrate that **1a** is able to bind to VHL and displace a high-affinity HIF-1 α peptide from VHL.

Figure 2d | A competitive fluorescence polarization binding assay was performed to evaluate the displacement of a fluorescent peptide (FAM-DEALAHyp-YIPD) from VHL ($K_d = 178.5$ nM) by complex **1a**. The IC_{50} of complex **1a** was determined to be 14.65 μ M in the presence of 125 nM of the fluorescent peptide and 450 nM of VHL using a four-parameter logistic equation.

Supplementary Fig. 6 | SPR sensograms showing the binding affinity of HIF-1 α peptide, FAM-DEALAHyp-YIPD, to surface-immobilized full-length recombinant VHL protein. The K_d value of the interaction was determined to be 234 nM.

Figure 2e | SPR sensograms showing the binding of complex **1a** to surface-immobilized VHL.

Reviewer #2 (Remarks to the Author):

The authors have responded to many of my concerns. (Please note that my focus area is diabetes pathophysiology and the paper has been reviewed with my glasses of expertise. I recognize that reviewer 1 has a number of concerns which seem substantial - it is beyond my expertise to fully evaluate whether the technical concerns raised by him/her has been appropriately met and I leave that at the discretion of the editor and reviewer 1).

Two concerns remain from my viewpoint:

Comment 1: *In figure 3c the authors only see partial effects by HIF1 α -knockdown and it should therefore be acknowledged that other pathways could also be involved.*

Our response: We thank the reviewer for the useful suggestion. We have acknowledged the key point as following in the revised manuscript.

"However, as only partial effects of HIF-1 α knockdown are observed, we do not rule out the possibility that other pathways could be involved in mediating the on-target effects of complex **1a**."

Comment 2: *It is incorrect to use the terms type 1 and type 2 diabetes for mice. The mice studied in this paper do not have type 1 or type 2 diabetes - they have hyperglycemia induced in ways to mimic type 1 and type 2 diabetes, respectively. There is no really good rodent type 2 model. The HFD/STZ model used here and termed "type 2" has very different features compared to the human situation. The*

HFD used to induce insulin resistance is appropriate but the insulin secretion failure seen in human T2D is not caused by such a severe destruction as the STZ treatment models. It is rather a matter of long-term hyperinsulinemia which subsequently leads to loss of specialized function and/or cell death (the exact mechanisms are debated). The model used here is not worse than many other models but it should not be called type 2 diabetes but a mimic of type 2 diabetes and its limitations should be clearly stated in the text. Moreover, the diabetic ulcers usually occur after years/decades and the very different timing in this study compared to the real situation has to be acknowledged as it may limit the clinical usefulness of the findings.

Our response: We thank the reviewer for the constructive suggestion. According to the reviewer's suggestion, we revised the terms "type 1 diabetes" and "type 2 diabetes" into "STZ" and "HFD/STZ", respectively. We completely agree with the reviewer that no animal model is exactly the same as human T2D.

In the current study, a high-fat diet with subsequent multiple injections of a low dose of streptozotocin (HFD/STZ) in mice was used to mimic type 2 diabetes. The key advantage of this model is to mimic the slow pathogenesis of type 2 diabetes that occurs in humans, encompassing the slow development from adult-onset diet-induced obesity to glucose intolerance, insulin resistance, the resulting compensatory insulin release and finally streptozotocin-induced partial β -cell death. Streptozotocin is a chemical used to destroy insulin-producing β -cells, which causes severe insulin secretion failure. On the contrary, long-term hyperinsulinemia occurs in human diabetes, which subsequently leads to loss of specialized function and/or cell death. Additionally, streptozotocin treatment causes liver and kidney toxicity and mild carcinogenic adverse effect (Nature reviews Endocrinology, 2018, 14: 140-162). Due to these limitation, local application of complex **1a** was performed on another widely used diabetic model, *db/db* mice, to confirm the therapeutic effect of this compound on diabetic wounds. The results further supported that complex **1a** accelerates wound closure in *db/db* mice (Fig. 4).

Diabetic ulcers usually occur in humans after years or decades. Due to the limitations of animal models, the timescale for the treatment of diabetic ulcers in this study is very different compared to how diabetic ulcers develops in humans. To further investigate the clinical usefulness of complex **1a**, the therapeutic effect of this

compound in aged diabetic mice, such as old *db/db* mice, might be tested in a future study.

According to the reviewer's suggestion, we have acknowledge the limitations as following in the revised manuscript.

"However, it should be noted that the *db/db*, HFD/STZ and STZ models used in this study each have their own limitations and cannot fully mimic human diabetes^{49, 50}. Thus, further research is needed in order to validate HIF-1 α -VHL inhibition as a target for wound healing in clinical practice."

Moreover, in order to avoid any misunderstanding regarding the clinical usefulness of the findings in diabetic ulcers therapy, we have also revised the introduction as following in the revised manuscript.

"... Wound healing in diabetic patients is compromised due to impaired wound contraction, reduced blood supply, and infection³⁻⁵. To date, several treatment strategies for diabetic wounds are available, including regular debridement, surgical revascularization, infection therapy, pressure-offloading and bioengineered alternative tissue products⁶⁻⁸. However, most treatments are effective only for mild to moderate wounds and are not 100% effective in preventing the risk of amputation and recovering full skin functionalities^{9, 10}. Therefore, the development of novel drugs or therapies for the treatment of diabetic wound healing in clinical practice is desperately needed."

Figure 4 | Complex 1a (0.25 mg/mL) accelerates wound closure in *db/db* mice. (a) Timeline for *in vivo* experiments. (b) Image of representative wound (left) and wound closure rate (right) ($n = 5$). (c) H&E and Masson's trichrome staining of dorsal skin section and skin thickness from the top of the epidermis to the bottom of the dermis in mice after 8 days post-injury ($n = 4$). Scale bar = 200 μm . (d) Laser doppler imager in dorsal skin: representative images were shown for each group (left) and baseline perfusion on back skin of mice (right) after 2 days post-injury (dotted line circle represents the wound bed in mice of each group and arrow represents perfusion intensity ($n = 5$)). (e) CD31 and DAPI double staining in wound bed of mice after 8 days post-injury ($n = 4$). Scale bar = 50 μm . Data are expressed as means \pm SD. $^{\#}P < 0.05$, $^{\#\#}P < 0.01$ WT vs. *db/db*, $^*P < 0.05$, $^{**}P < 0.01$ **1a** vs. vehicle.

Reviewer #3 (Remarks to the Author):

If the aim of the authors is to suggest a new therapeutically approach for diabetic wounds they should test the compound locally and not systemically.

The animal model used for "type 2 diabetes" is mainly a model for type 1 diabetes since the animals were injected with STZ. Other established models for wound healing impairment linked with type 2 diabetes are availabl.

Our response: We thank the reviewer for the constructive suggestion. To confirm the therapeutic effect of complex **1a** on diabetic wounds, the local application of this compound was performed in *db/db* mice. The results showed that local application of complex **1a** accelerates wound closure in *db/db* mice (Fig. 4).

We agree with the reviewer that high-fat diet fed, and streptozotocin-treated mice model (HFD/STZ) cannot completely mimic human type 2 diabetes. Streptozotocin causes acute insulin secretion failure, however, long-term hyperinsulinemia is the key characteristic of human type 2 diabetes. However, many studies still use the HFD/STZ model as a type 2 diabetes model due to the simplicity of its use (Journal of Diabetes Investigation, 2014, 5(4): 349–358. Diabetes, 2014, 63(3): 1048-1057; Molecular Therapy, 2018, 26(8): 1921-1930).

Type 2 diabetes is characterized by insulin resistance and the inability of β -cell to sufficiently compensate. Therefore, animal models of type 2 diabetes tend to replicate insulin resistance and/or β -cell failure. Many rodent models have been established to

investigate type 2 diabetes, including polygenic models (C57BL/6J-DIO, SWR/J-DIO, NZO-carbohydrate enriched diet, DR Sprague Dawley-DIO, and Goto-Kakizaki), monogenic models (C57BL/6J-*ob/ob*, C57BLKS/J-*db/db*, Zucker Diabetic Fatty, and *fa/fa*), and experimental models (HFD/STZ and ventromedial hypothalamus lesion). Each animal model has its advantages and disadvantages. The choice of model depends on the purpose of the study. Type 2 diabetes models usually come with a variety of associated pathologies, such as obesity, hyperglycemia, insulin resistance, dyslipidemia, and pathologic islet changes. Although these co-morbidities are common in human type 2 diabetes, no model is completely reflective of the human condition. Ideally, more than one animal model should be used to represent the diversity seen in human type 2 diabetes.

Among these models, C57BL/6J-*ob/ob*, C57BLKS/J-*db/db*, C57BL/6J DIO, and NZO-carbohydrate enriched diet models can exhibit impaired wound healing (JAX Mouse Model of Type 2 Diabetes, <https://www.jax.org/news-and-insights/jax-blog/2015/july/choosing-among-type-ii-diabetes-mouse-models>; Nature reviews Endocrinology, 2018, 14: 140-162; Br J Pharmacol. 2012 Jun; 166(3): 877–894). C57BLKS/J-*db/db* mice model is widely used for investigating diabetic wound healing (Circulation, 2006, 113(20): 2413-2424; Proceedings of the National Academy of Sciences of the United States of America 2008, 105: 19426-19431; Diabetes 2014, 63: 1763-1778; Circulation research, 2003, 92(11): 1247-1253). C57BLKS/J-*db/db* mice manifest morbid obesity, chronic hyperglycemia, and pancreatic β -cell atrophy, and can develop phase I, II and III of type 2 diabetes. Therefore, to confirm the therapeutic effect of complex **1a** on diabetic wounds, the local application of this compound was performed in *db/db* mice, which showed that complex **1a** could accelerate wound closure *in vivo* (Figs. 4 and 5).

Figure 4 | Complex 1a (0.25 mg/mL) accelerates wound closure in *db/db* mice. (a) Timeline for *in vivo* experiments. (b) Image of representative wound (left) and wound closure rate (right) ($n = 5$). (c) H&E and Masson's trichrome staining of dorsal skin section and skin thickness from the top of the epidermis to the bottom of the dermis in mice after 8 days post-injury ($n = 4$). Scale bar = 200 μm . (d) Laser doppler imager in dorsal skin: representative images were shown for each group (left) and baseline perfusion on back skin of mice (right) after 2 days post-injury (dotted line circle represents the wound bed in mice of each group and arrow represents perfusion intensity ($n = 5$)). (e) CD31 and DAPI double staining in wound bed of mice after 8 days post-injury ($n = 4$). Scale bar = 50 μm . Data are expressed as means \pm SD. $\#P < 0.05$, $\#\#P < 0.01$ WT vs. *db/db*, $*P < 0.05$, $**P < 0.01$ **1a** vs. vehicle.

Figure 5 | Complex 1a (0.25 mg/ml) activates gene expression regulated by HIF-1 α in *db/db* mice at 8 days post-injury. (a) Western blot analyses and quantitation of HIF-1 α , VEGF, GLUT1, and EPO in wound tissue ($n = 3$). All proteins were normalized by β -actin. (b) The mRNA levels of HIF-1 α target genes involved in wound healing were analysed by qRT-PCR in wound tissues ($n = 5$). Data are expressed as means \pm SD. # $P < 0.05$, ## $P < 0.01$ WT vs. *db/db*, * $P < 0.05$, ** $P < 0.01$ **1a** vs. vehicle.

Reviewers' Comments:

Reviewer #1:

Remarks to the Author:

IN this new version, the authors have made an attempt to address the serious concerns raised at previous rounds of reviews. They now provide more information on protein purification protocols, replace the old ITC data with a new one, and add some fluorescence polarization (FP) and surface plasmon resonance (SPR) data for binding of compound 1a and control HIF peptide to VHL. While this is a laudable effort in the right direction, the data remain inconclusive for the reasons highlighted below. This reviewer remain unconvinced that compound 1a binds to VHL and genuinely blocks the VHL-HIF interaction as claimed - and thus I maintain that I cannot recommend publication. Nevertheless, even if it was to be believed, the affinity reported (KD ~6-15uM) appears to be too weak to warrant any realistic expectations to infer any meaningful claims from the cellular data.

There are issues and concerns with all of the methods reported:

- Protein purification

The authors claim to have purified GST-VHL construct. The new biophysical data relies on the behaviour of this recombinantly purified GST-VHL construct. VHL is renowned to be highly unstable and insoluble in the absence of the adaptor proteins Elongin B and Elongin C - and remains insoluble in inclusion bodies if expressed alone in *E. coli*, even with the aid of potentially solubilizing tags such as GST. This reviewer therefore seriously doubts that the authors have in hand active protein. In the experimental they mention expression in DH5a, which is very unusual for protein expression. The authors should co-express VHL together with ElonginC and ElonginB, as established protocol in the literature, and used this resultant VCB complex for biophysical studies. This would help to begin clear-up any doubts about protein quality.

- ITC

First and foremost, the authors should absolutely perform a control ITC titration of the hydroxylated HIF peptide to their VHL protein, to quality check the functionality and binding capacity of their protein.

I also have serious reservations too with the titration shown for compound 1a titrated to protein. Firstly, the curve now looks amazingly good, with an almost perfect sigmoidal shape - while in the previous version of the paper the previous ITC data look much weaker signal and hyperbolic in shape. It seems unlikely that these changes are to be due merely to the difference in concentrations used. Secondly, how do the authors know that their protein was fine under 5 % DMSO? Thirdly, They also don't provide any other values apart from the Kds, including delta H, delta G, n-value, etc, which would be useful in interpreting the data and validating the protein quality. The Delta H appears to be also remarkably and perhaps suspiciously large (-14 kcal/mol). Together these observations, and being 1a a metal organic complex, I wonder whether other equilibria e.g. metal dissociation, protonation effects might take place under these conditions. The authors should absolutely perform and report control titrations in which 1) species in syringe is titrated in to buffer alone; 2) buffer in syringe is titrated into species in cell.

- SPR

SPR data is the only raw data provided for the control HIF peptide, yet it is unclear why this is done with a FAM-labelled peptide and not with an unlabelled peptide. This is the only technique in which it can be possible based on the shown data to properly compare HIF peptide binding to 1a binding for their protein. However, there seems to be apparent inconsistencies between the two datasets. Given the much higher MW of the HIF peptide compared to 1a, would one not expect a much higher response compared to compound 1a assuming roughly equivalent amount of immobilization? In fact the reverse is observed in the image (RU up to 4 with the peptide, and up to 15 with the compound 1a). The level of immobilization of the protein in the two experiments is not specified too, so it is not possible to interpret how these responses compare relative to the

maximal response (R_{max}) expected based on protein immobilized – this will allow to get an idea of the activity of their SPR chip surface.

- FP

Again, key controls are missing here. The compound 1a is likely fluorescent, how does this affect the signal? At the very least, a control titration of 1a titrated to buffer in the absence of protein should be included. As a positive control, again, the authors should perform a titration of high-affinity unlabelled HIF peptide in this assay – and show how the data compare to the literature data.

Additional general comment. The authors might want to check whether their compounds destabilizes the protein. This could be one way in which it leads to apparent loss of HIF binding. The authors might want to check what effect the compound has on the folding and stability of the protein (e.g. using CD) – that would also be useful to quality control their protein preps too!.

Reviewer #2:

Remarks to the Author:

The authors have responded well to my particular concerns.

Reviewer #3:

Remarks to the Author:

No further comments.

Reviewers' comments:

Reviewer #1 (Remarks to the Author):

IN this new version, the authors have made an attempt to address the serious concerns raised at previous rounds of reviews. They now provide more information on protein purification protocols, replace the old ITC data with a new one, and add some fluorescence polarization (FP) and surface plasmon resonance (SPR) data for binding of compound 1a and control HIF peptide to VHL. While this is a laudable effort in the right direction, the data remain inconclusive for the reasons highlighted below. This reviewer remain unconvinced that compound 1a binds to VHL and genuinely blocks the VHL-HIF interaction as claimed - and thus I maintain that I cannot recommend publication. Nevertheless, even if it was to be believed, the affinity reported ($KD \sim 6-15\mu M$) appears to be too weak to warrant any realistic expectations to infer any meaningful claims from the cellular data.

There are issues and concerns with all of the methods reported:

• Protein purification

*The authors claim to have purified GST-VHL construct. The new biophysical data relies on the behaviour of this recombinantly purified GST-VHL construct. VHL is renown to be highly unstable and insoluble in the absence of the adaptor proteins Elongin B and Elongin C – and remains insoluble in inclusion bodies if expressed alone in *E. coli*, even with the aid of potentially solubilizing tags such as GST. This reviewer therefore seriously doubts that the authors have in hand active protein. In the experimental they mention expression in DH5a, which is very unusual for protein expression. The authors should co-expressed VHL together with ElonginC and ElonginB, as established protocol in the literature, and used this resultant VCB complex for biophysical studies. This would help to begin clear-up any doubts about protein quality.*

Our response: We thank the reviewer for the constructive suggestion. We have constructed the plasmids to express and purify the human recombinant VHL:ElonginB:ElonginC (VBC) complex (Supplementary Fig. 6a), which was then

verified by using pull-down and CD studies (Supplementary Fig. 6b and 6c). The methods have been updated in the revised manuscript as follows:

"VBC protein expression and purification. Fragment of VHL coded sequence (corresponding to the 54th-213th amino acids of VHL protein) was first amplified from a VHL-pGEX-2TK plasmid, a gift from Dr. William Kaelin (Addgene plasmid # 20790), with primers with recognized sites of restriction endonucleases *EcoRI* and *XhoI* (Forward, 5'-CCGGAATTCATGGAGGCCGGGCGGCCGCG-3'; reverse, 5'-CCGCTCGAGATCTCCCATCCGTTGATGTGCAAT-3') and inserted into T-Vector pMD™19 (Simple). The recombinant plasmid and pET28a vector were digested, extracted, and ligated to give rise to a pET28a_VHL plasmid with a six Histidine-tag. pACYC-1, an Elongin B and Elongin C co-expression plasmid, was a gift from Dr. Nicola Burgess-Brown (Addgene plasmid # 110274). Plasmids pET28a_VHL and pACYC-1 were cotransformed into the expression strain *E. coli* (BL21) (DE3), and verified by using PCR (VHL₅₄₋₂₁₃ primers, Forward, 5'-CCGGAATTCATGGAGGCCGGGCGGCCGCG-3'; reverse, 5'-CCGCTCGAGATCTCCCATCCGTTGATGTGCAAT-3'; pACYC-1 primers: forward, 5'-ATGATGTATGTCAAATTGATATCATCT-3'; reverse, 5'-CTAACAATCTAAGAAGTTCGCAGCCATC-3'). To obtain the VBC complex, a preculture from one colony of BL21(DE3)(pET28a_VHL + pACYC_1) was grown overnight at 37°C in lysogeny broth (LB) medium supplemented with 50 µg/ml kanamycin and chloramphenicol (Kan⁺/Chl⁺). The human recombinant VBC protein were expressed and purified as described previously¹, with minor modifications. Brief, the *E. coli* BL21(DE3) containing plasmids pET28a_VHL and pACYC_1 were cultured in LB medium with Kan⁺/Chl⁺ (50 µg/ml), and grew at 37°C until the OD₆₀₀ of 0.6-0.8. Then the bacterial solutions were induced with 1 mM isopropyl-β-D-thiogalactopyranoside (IPTG) at 37°C for 4 h. The recombinant *E. coli* cells were harvested by centrifugation at 5000 g for 20 min and then homogenized by sonication in buffer (20 mM Tris-HCl, 400 mM NaCl, 5 mM imidazole, PH 7.4) and precleared lysates were applied to His GraviTrap columns (GE Healthcare, Catalog No. 11-0033-99), following kit protocols for purifications. The protein was assayed by using pull down assay and circular dichroism measurement."

Supplementary Figure 6 | (a) Agarose gel electrophoresis of DNA fragments from the colonies of BL21(DE3)(pET28a_VHL + pACYC_1) grown overnight at 37°C in lysogeny broth (LB) medium supplemented with 50 µg/ml kanamycin and chloramphenicol (Kan⁺/Chl⁺). Plasmids pET28a_VHL and pACYC-1 were co-transformed into the expression strain *E. coli* (BL21) (DE3), and verified by using PCR (VHL₅₄₋₂₁₃ primers, Forward, 5'-CCGGAATTCATGGAGGCCGGGCGGCCGCG-3'; reverse, 5'-CCGCTCGAGATCTCCCATCCGTTGATGTGCAAT-3'; Elongin BC primers: Forward, 5'-ATGATGTATGTCAAATTGATATCATCT-3'; reverse, 5'-CTAACAATCTAAGAAGTTCGCAGCCATC-3'). Lane M: low molecular weight DNA ladder; Lane W: PCR products of water; Lane P: PCR products of the partial sequences from plasmids pET28a_VHL (left, 498 bp) and pACYC-1 (right, 294 bp); Lane 1: the colony of BL21(DE3)(pET28a_VHL + pACYC_1). (b) The pull down assay was performed by using a gravity flow column with Ni-NTA agarose beads. 50 µg His-tag VHL protein with 400 µL equilibrium buffer was immobilized in the column, and 200 µL bacterial extract with or without IPTG induction with 200 µL equilibrium buffer was loaded onto the gravity flow column. The gravity flow column was then washed, and the fractions eluted by elution buffer contained 500 mM imidazole were analyzed by Western blotting using VHL antibody anti-VHL antibody (GeneTex, GTX101087, 1:1,000 dilution) and Elongin B Antibody (Santa Cruz Biotechnology, Inc., sc-133090, 1:1,000 dilution). Lane 1: Bacterial extract (BL21(DE3)/Elongin BC) without induction with IPTG; Lane 2: Bacterial extract (BL21(DE3)/Elongin BC) with 4h induction with IPTG; Lanes 3: 5 X diluted bacterial extract without induction with IPTG; Lanes 4: 5 X diluted bacterial extract with 4h induction with IPTG.

Supplementary Figure 6c | Circular dichroism of VBC complex with or without **1a** and 1% DMSO. CD spectroscopy measurements were carried in JASCO-815 spectropolarimeter at room temperature to evaluate the folding and stability of VBC complex. VBC was at a final concentration of 5 μM in CD buffer (pH 7.4).

• *ITC*

First and foremost, the authors should absolutely perform a control ITC titration of the hydroxylated HIF peptide to their VHL protein, to quality check the functionality and binding capacity of their protein.

I also have serious reservations too with the titration shown for compound 1a titrated to protein. Firstly, the curve now looks amazingly good, with an almost perfect sigmoidal shape – while in the previous version of the paper the previous ITC data look much weaker signal and hyperbolic in shape. It seems unlikely that these changes are to be due merely to the difference in concentrations used. Secondly, how do the authors know that their protein was fine under 5 % DMSO? Thirdly, They also don't provide any other values apart from the K_{ds} , including ΔH , ΔG , n -value, etc, which would be useful in interpreting the data and validating the protein quality. The ΔH appears to be also remarkably and perhaps suspiciously large (-14 kcal/mol). Together these observations, and being 1a a metal organic complex, I wonder whether other equilibria e.g. metal dissociation, protonation effects might take place under these conditions. The authors should absolutely perform and

report control titrations in which 1) species in syringe is titrated in to buffer alone; 2) buffer in syringe is titrated into species in cell.

Our response: We thank the reviewer for the constructive suggestion. We agree with the reviewer that a control ITC titration of the hydroxylated HIF peptide to the VHL protein is indispensable to check the functionality and binding capacity of human recombinant VBC protein, and have added this ITC titration result in Supplementary Fig. 6d. The hydroxylated HIF peptide DEALA-Hyp-YIPD showed a K_d of $0.46 \pm 0.16 \mu\text{M}$, which verified that hydroxylated HIF peptide has high-affinity for interacting with VBC complex¹. With regards to the difference in the shapes of the titration curves in the two experiments, we suspect that it may be caused by the use of different batches of protein: one was purchased earlier (VHL protein complex, Catalog No. 23-044M) from Merck Millipore while the other one was purified by ourselves using the VHL-pGex2TK plasmid (Addgene plasmid # 20790). To address the data ambiguity issue, we have used the same batch of protein VBC complex to perform the biophysical studies in the revised manuscript, including CD (Supplementary Figure 6c), ITC (Figure 2c and Supplementary Figure 6d), FP (Figure 2d and Supplementary Figure 6e) and SPR (Figure 2e and Supplementary Figure 6f) studies.

We agree that the DMSO concentration can directly affect protein structure and activity. As 2-5% DMSO is often used in SPR to help solubilize chemical compounds injected over proteins or nucleic acids, we have performed additional experiments to investigate the effect of DMSO concentration on the protein. In this revision, the concentration of DMSO did not exceed 1% in all of the biophysical experiments, including CD, ITC, SPR and FP studies. At this concentration, DMSO had no significant effect towards the VBC complex as analyzed using CD. As shown in Supplementary Fig. 6c, there was no significant shift of the signal when VBC was incubated with 1% DMSO, indicating that 1% DMSO has no significant effect on the folding and stability of VBC.

Furthermore, the values of the binding constants (K_d), reaction stoichiometry (N), binding enthalpy (ΔH), changes in free energy (ΔG), and $-T\Delta S$ have been added in Figure 2c, which showed that **1a** bound to VBC complex with a 1:1 stoichiometry and a dissociation constant of $K_d 1.08 \pm 0.20 \mu\text{M}$. Moreover, the ITC data in Figure 2c indicates that the binding between **1a** and VBC is strongly enthalpy-driven ($\Delta G = -$

8.14 kcal/mol, $-T\Delta S = -0.29$ kcal/mol, $\Delta H = -7.85$ kcal/mol), suggesting that hydrogen bonds and electrostatic interactions may play a key role in this binding.

Moreover, we suspect that the high value of ΔH caused may be due to the higher concentration of **1a** in the syringe and GST-tagged VHL protein in the cell during the experiment. In the previous ITC study, the 2 mM of complex **1a** was titrated against 200 μ M of GST-tagged VHL protein. To overcome this issue, we have used 300 μ M of **1a** to titrate against 30 μ M of the new purified VBC complex in the ITC experiment. We found by ITC that complex **1a** bound to VHL *in vitro* with a ΔH value of -7.85 ± 0.28 kcal/mol (Figure 2c).

According to the referee's constructive suggestion, we have also evaluated the binding affinity of **1a** to the VBC complex by using composite model with three control titrations, which including (1) 300 μ M of **1a** in syringe titrated into buffer alone; (2) buffer in syringe titrated into 30 μ M of VBC complex in the cell; (3) buffer in syringe titrated into buffer (Figure a). The result in Figure a showed that the three control titrations produced visible DP signals with fluctuations ranging from -0.15 to 0.1 μ cal/s, which were subtracted during analysis by using MicroCal PEAQ-ITC Analysis Software following the software's instructions (Figure 2c). The method conditions of the ITC assay have also been updated in the revised manuscript as follows:

"Isothermal titration calorimetry. ITC experiments were carried in a MicroCal PEAQ-ITC Isothermal Titration Calorimeter (Malvern Panalytical), as previously described with minor modification². Briefly, complex **1a** and recombinant VBC complex were dialyzed into the ITC buffer (20 mM Bis-Tris, 150 mM NaCl, 2 mM DTT, 1% DMSO) overnight. Complex **1a** (300 μ M) was titrated against 30 μ M VBC complex, consisting of 19 injections of 2 μ L complex **1a** solution at a rate of 2 sec/ μ L at 150 s time intervals. An initial injection of ligand (0.4 μ L) was made and discarded during data analysis. The experiment was carried out at 25 °C while stirring at 750 rpm. The generated data were fitted to a single binding site model using the Setup MicroCal PEAQ-ITC Analysis Software provided by the manufacturer. Three control titrations, in which (1) **1a** is titrated into VBC buffer; (2) VBC buffer is titrated into VBC complex; (3) VBC buffer is titrated into VBC buffer, were also analyzed by using composite model."

Supplementary Figure 6d | ITC titration of hydroxylated HIF peptide DEALA-Hyp-YIPD (300 μM) into recombinant VBC complex (30 μM). ITC experiments were carried in a MicroCal PEAQ-ITC Isothermal Titration Calorimeter (Malvern Panalytical) and analyzed by using composite model with three control titrations.

Supplementary Figure 6c | Circular dichroism of VBC complex with or without **1a** and 1% DMSO. CD spectroscopy measurements were carried in JASCO-815 spectropolarimeter at room temperature to evaluate the folding and stability of VBC complex. VBC was at a final concentration of 5 μM in CD buffer (pH 7.4).

Figure 2c | ITC titration of **1a** (300 μM) into recombinant VBC complex (30 μM). ITC experiments were carried in a MicroCal PEAQ-ITC Isothermal Titration Calorimeter (Malvern Panalytical) and analyzed by using composite model with three control titrations.

Figure a | Three representative control titrations for the ITC titration of **1a** (300 μM) into recombinant VBC complex (30 μM), including (1) 300 μM of **1a** in syringe titrated into buffer alone; (2) buffer in syringe titrated into 30 μM of VBC complex in the cell; (3) buffer in syringe titrated into buffer. ITC experiments were carried in a MicroCal PEAQ-ITC Isothermal Titration Calorimeter (Malvern Panalytical) and analyzed by using MicroCal PEAQ-ITC Analysis Software.

- *SPR*

SPR data is the only raw data provided for the control HIF peptide, yet it is unclear why this is done with a FAM-labelled peptide and not with an unlabelled peptide. This

is the only technique in which it can be possible based on the shown data to properly compare HIF peptide binding to 1a binding for their protein. However, there seems to be apparent inconsistencies between the two datasets. Given the much higher MW of the HIF peptide compared to 1a, would one not expect a much higher response compared to compound 1a assuming roughly equivalent amount of immobilization? In fact the reverse is observed in the image (RU up to 4 with the peptide, and up to 15 with the compound 1a). The level of immobilization of the protein in the two experiments is not specified too, so it is not possible to interpret how these responses compare relative to the maximal response (Rmax) expected based on protein immobilized – this will allow to get an idea of the activity of their SPR chip surface.

Our response: We thank the reviewer for the constructive suggestion. We have performed the SPR study by using unlabelled HIF-1 α peptide, DEALAHyp-YIPD, which showed a high binding affinity binding to surface-immobilized recombinant VBC protein complex with a K_d value of 379.80 ± 5.74 nM (Supplementary Fig. 6f). In regards to the difference of RU response, we suspect that one reason is that we indeed did not capture the same amount of target and that the concentration range of **1a** and FAM-labelled HIF peptide used in the previous version were too different between these two experiments, and so it is not appropriate to compare the responses. The max concentration of **1a** was 25,000 nM, while that of FAM-labelled HIF peptide was 1,000 nM. In the current study, we have performed the SPR assay capturing the same amount of His-tagged VBC and we injected partially overlapping concentrations (625, 1250, 2500 and 5000 nM) to compare the binding ability of HIF peptide and **1a** for recombinant VBC protein. The final surface density of VBC was approximately 2000 RU. The results showed that complex **1a** bound to the immobilized recombinant VBC protein complex with a K_d value of 1.72 ± 0.15 μ M (Figure 2e), while the positive control HIF peptide bound with a K_d value of 379.80 ± 5.74 nM (Supplementary Figure 6f). As shown in Figure 2e, **1a** showed an association rate constant (k_{on}) of 1.93×10^4 (1/Ms) and a dissociation rate constant (k_{off}) of 3.33×10^{-2} (1/s) with immobilized VBC protein using a 1:1 binding model. In short, these biophysical experiments are giving the same K_d for **1a** and stoichiometry as measured by ITC. Taken together, these data demonstrate that **1a** is able to bind to VBC complex and displace a high-affinity HIF-1 α peptide from VBC.

The methods have been updated in the revised manuscript as follows:

"Surface plasmon resonance. The surface plasmon resonance study was carried out as described previously with minor modification^{3,4}. SPR analyses were performed at 25°C with a Biacore X100 instrument (GE Healthcare BioSciences AB, Uppsala, Sweden). Histidine-tagged VBC was immobilized onto the CM5 sensor chip surface using the His Capture Kit, following the procedure recommended by the producer. Firstly, the CM5 sensor chip was first equilibrated with HBS-EP buffer [10 mmol/L HEPES (pH 7.4), 150 mmol/L NaCl, 3 mmol/L EDTA, 0.05% (v/v) Tween 20, 0.1%(v/v) DMSO] overnight. An anti-histidine antibody (GE Healthcare) was then immobilized using standard amine coupling chemistry. Briefly, the carboxymethylated dextran surface of the CM5 sensor chip (GE Healthcare) was first activated by a 7-minute injection of a 1:1 mixture of 0.4 M N-ethyl-N-(3-diethylaminopropyl) carbodiimide and 0.1 M N-hydroxysuccinimide at 10 µL/min to give reactive succinimide esters. Then a solution of the anti-histidine antibody (50 µg/µL) in immobilization buffer (10 mM sodium acetate pH 4.5, GE Healthcare) was fluxed on the reactive matrix using a flow rate of 10 µL/min. We immobilized more than 9,000 resonance units (RU) of the anti-histidine antibody in flow cells 1 and 2. The surfaces of both flow cells were deactivated with ethanolamine (GE Healthcare) with a flow rate of 10 µL/min. The complex **1a** or HIF peptide (DEALAHyp-YIPD) was serially diluted in HBS-EP buffer and injected for 60 seconds (contact phase) to the immobilized histidine-tagged VBC (~2000 RU) as the chip surface, followed by 180 seconds for the dissociation phase, followed by regeneration (10 mM glycine–HCl pH 1.5, GE Healthcare). The K_d values of the tested compounds were determined using the Biacore evaluation software (GE Healthcare). Reference flow cell response was subtracted from the sample response with captured VBC protein to correct for systematic noise and baseline drift. Data were solvent corrected and the response from the blank injections was used to double reference the binding data. k_{on}/k_{off} values were obtained using a 1:1 binding model fit. HIF peptide purchased from IGE Biotechnology Co (Guangzhou, China)."

Figure 2e | SPR sensorgrams showing the binding of complex **1a** to surface-immobilized VBC. The K_d value of the interaction was determined to be 1.72 ± 0.15 μM .

Supplementary Figure 6f | SPR sensorgrams showing the binding affinity of unlabelled HIF-1 α peptide, DEALAHyp-YIPD, to surface-immobilized recombinant VBC protein complex. The K_d value of the interaction was determined to be 379.80 ± 5.74 nM.

• *FP*

Again, key controls are missing here. The compound 1a is likely fluorescent, how does this affect the signal? At the very least, a control titration of 1a titrated to buffer in the absence of protein should be included. As a positive control, again, the authors should perform a titration of high-affinity unlabelled HIF peptide in this assay – and show how the data compare to the literature data.

Our response: We thank the reviewer for the useful suggestion. We completely agree with the reviewer that a control titration of **1a** into buffer in the absence of protein should be performed. In the revised study, the FP signal of compound **1a** was

recorded as a control and subtracted from the experimental data sets. Moreover, as suggested by the reviewer, the non-fluorescent HIF-1 α peptide, DEALA-Hyp-YIPD, was used as a positive control and found to bind to human recombinant VBC complex with $IC_{50} = 1.09 \mu\text{M}$ and $K_d = 773.86 \text{ nM}$ (Supplementary Fig. 6e), which is similar to the reported literature¹. Furthermore, the polarization was also measured in the presence of VBC and serial dilutions of complex **1a** in order to observe the displacement of the fluorescent peptide. From this assay, the IC_{50} of complex **1a** was determined to be $3.79 \mu\text{M}$ ($K_d = 2.06 \mu\text{M}$) in the presence of 450 nM VBC and 125 nM fluorescent HIF-1 α peptide (Fig. 2d), demonstrating that complex **1a** could displace the high-affinity HIF-1 α peptide from VBC. The methods have been updated in the revised manuscript as follows:

"Fluorescence polarization assay. The fluorescence polarization assay was carried out as described previously^{5, 6}. The fluorescent ligand FAM-DEALA-Hyp-YIPD was firstly diluted with VBC buffer (100 mM Tris, 100 mM NaCl, 1 mM DTT, pH 7.0). For K_d determination of fluorescent peptides, the fluorescence polarization of a 2-fold serial dilution of 10 μM VBC in 125 nM of FAM-DEALA-Hyp-YIPD was prepared in VBC buffer. For the dose response curves for complex **1a** or peptide ligand, wells of a 384-well plate were added 9 μL of 1 μM VBC (450 nM final), 9 μL of 278 nM FAM-DEALAHyp-YIPD (125 nM final), and 2 μL of complex **1a** or unlabelled peptide ligand (2-fold serial dilutions starting from 400 μM , 1% DMSO final). The reference FP response of **1a** was subtracted from the sample response with immobilized VBC protein. Control wells contained VBC and FAM-labelled peptide in the absence of compound (maximum signal), and FAM-labelled peptide in the absence of protein (background signal). Before reading the fluorescence polarization signal on a SpectraMax M5 microplate reader (Molecular Devices, excitation 485 nm, emission 520 nm), the plate was shaken for 1 minute, then centrifuged for 1 minute. K_d values were then back-calculated from the measured IC_{50} values as previously described⁶."

Supplementary Fig. 6e | A competitive fluorescence polarization binding assay was performed to evaluate the displacement of a fluorescent peptide (FAM-DEALAHyp-YIPD) from VBC ($K_d = 421.50 \pm 65.23$ nM) by unlabelled HIF peptide DEALAHyp-YIPD. The IC_{50} of DEALAHyp-YIPD ($K_d = 773.86$ nM) was determined to be $1.09 \mu\text{M}$ in the presence of 125 nM of the fluorescent peptide and 450 nM of VBC using a four-parameter logistic equation.

Figure 2d | A competitive fluorescence polarization binding assay was performed to evaluate the displacement of a fluorescent peptide (FAM-DEALAHyp-YIPD) binding to VBC ($K_d = 421.50 \pm 65.23$ nM) by complex **1a**. The IC_{50} of complex **1a** ($K_d = 2.06 \mu\text{M}$) was determined to be $3.79 \mu\text{M}$ in the presence of 125 nM of the fluorescent peptide and 450 nM of VBC using a four-parameter logistic equation.

Additional general comment. The authors might want to check whether their compounds destabilizes the protein. This could be one way in which it leads to apparent loss of HIF binding. The authors might want to check what effect the

compound has on the folding and stability of the protein (e.g. using CD) – that would also be useful to quality control their protein preps too!.

Our response: We thank the reviewer for the constructive suggestion. We have studied the possible denaturation of VBC by using CD spectroscopy after incubation of the compound.

Circular dichroism (CD) is an excellent tool for examining the interactions and stability of proteins⁷. Herein, the purified human recombinant VBC complex was incubated with **1a** or 1% DMSO to study any changes on the conformation upon their binding using circular dichroism. The VBC complex showed no significant shift upon incubation with 1% DMSO, while a distinct shift was observed when VBC was incubated with 3 μ M of **1a** in 1% DMSO. We suspect that the changes in its secondary structure induced by **1a** was due to the interaction between **1a** and VBC.

Supplementary Figure 6a | Circular dichroism of VBC complex with or without **1a** and 1% DMSO. CD spectroscopy measurements were carried in JASCO-815 spectropolarimeter at room temperature to evaluate the folding and stability of VBC complex. VBC was at a final concentration of 5 μ M in CD buffer (pH 7.4).

"**Circular dichroism measurement.** Protein concentration was 5 μ M in CD buffer (20 mM Tris, 200 mM NaCl, 1 mM DTT, pH 7.4). CD spectra were recorded on a JASCO-815 spectropolarimeter using 1 cm path length quartz cuvettes at 20 °C. Spectra were collected between 200 nm and 320 nm, using a data pitch length of 0.5

nm, bandwidth of 2 nm, averaging time of 5 s for each measurement, and an accumulation cycle of 3 runs per measurement. The smoothed curves were plotted using GraphPad Prism software. The data were baseline corrected using CD spectra of buffer alone. The buffer used for the experiment were filtered with 0.22 µm nylon membrane filter and degassed."

Reviewer #2 (Remarks to the Author):

The authors have responded well to my particular concerns.

Reviewer #3 (Remarks to the Author):

No further comments.

References

1. Buckley, D. L. *et al.* Targeting the von Hippel–Lindau E3 Ubiquitin Ligase Using Small Molecules To Disrupt the VHL/HIF-1 α Interaction. *J. Am. Chem. Soc.* **134**, 4465-4468 (2012).
2. Galdeano, C. *et al.* Structure-guided design and optimization of small molecules targeting the protein–protein interaction between the von Hippel–Lindau (VHL) E3 ubiquitin ligase and the hypoxia inducible factor (HIF) alpha subunit with in vitro nanomolar affinities. *J. Med. Chem.* **57**, 8657-8663 (2014).
3. Quevedo, C. E. *et al.* Small molecule inhibitors of RAS-effector protein interactions derived using an intracellular antibody fragment. *Nat. Commun.* **9**, 3169 (2018).
4. Moscetti, I. *et al.* MDM2–MDM4 molecular interaction investigated by atomic force spectroscopy and surface plasmon resonance. *Int. J. Nanomedicine* **11**, 4221-4229 (2016).
5. Yang, C. *et al.* Discovery of a VHL and HIF1 α interaction inhibitor with in vivo angiogenic activity via structure-based virtual screening. *Chem. Commun.* **52**, 12837-12840 (2016).
6. Van Molle, I. *et al.* Dissecting fragment-based lead discovery at the von Hippel-Lindau protein: hypoxia inducible factor 1 α protein-protein interface. *Chem. Biol.* **19**, 1300-1312 (2012).
7. Greenfield, N. J. Determination of the folding of proteins as a function of denaturants, osmolytes or ligands using circular dichroism. *Nat. Protoc.* **1**, 2733 (2006).

Reviewers' Comments:

Reviewer #1:

Remarks to the Author:

The authors provide new evidence to support their claim that their Iridium complex 1a binds to VHL. They now express and purify VHL in complex with ElonginB and ElonginC (VCB) using proper material and following due protocol. The protein looks fine and of good quality in their control experiments - it binds the known substrate HIF-1alpha peptide as expected in ITC, FP and SPR.

The data aims to show that complex 1a binds to VCB and provide new ITC, SPR and FP data - which all would seem fine in principle. However in my own lab we synthesised exactly the same compound 1a, and tested it in our own well-validated ITC and FP assays. Our data is consistent with 1a not binding to VCB - and crucially is clearly not consistent with the evidence provided herein by the authors. In addition, I have other concerns:

1) I've inspected carefully the new ITC data for 1a (Figure 2c). The integrated data points (bottom panel) appear to follow a similar pattern (same no. of points, very similar trend) as in the same Figure 2c in the previous version (revision 2).

2) It is also not clear to me how very good quality data was previously obtained for ITC, FP and SPR, with the previously purified VHL protein expressed alone. One possibility is that the previously purified protein was not active / not soluble.

3) the CD trace of VCB in the presence of 1a (red trace in Supp Fig. 6a) strongly suggests that the protein unfolds / loses substantial integrity and tertiary structure in the presence of the compound - invalidating the now presumably positive results by ITC/SPR/FP. In contrast, this CD data very much is in line with our own biophysical data, where using differential scanning Fluorimetry (thermal shift) we observed a significant negative thermal shift ($\Delta T_m = -5$ oC), consistent with destabilization / loss of structural integrity in the presence of compound 1a.

In summary, I remain unconvinced that 1a genuinely binds to VHL and inhibits the VHL-HIF-1 α interaction. I feel the authors have now properly purified soluble and active VCB for biophysical studies, but their data conflicts with my own lab's attempt to reproduce it.

Our Responses to the Comments of the Referees

Referee 1

Recommendation: Revisions required

Comments:

The authors provide new evidence to support their claim that their Iridium complex 1a binds to VHL. They now express and purify VHL in complex with ElonginB and ElonginC (VCB) using proper material and following due protocol. The protein looks fine and of good quality in their control experiments - it binds the known substrate HIF-1alpha peptide as expected in ITC, FP and SPR.

The data aims to show that complex 1a binds to VCB and provide new ITC, SPR and FP data - which all would seem fine in principle. However in my own lab we synthesised exactly the same compound 1a, and tested it in our own well-validated ITC and FP assays. Our data is consistent with 1a not binding to VCB - and crucially is clearly not consistent with the evidence provided herein by the authors. In addition, I have other concerns:

Our response: We thank the reviewer for the constructive suggestion. We sincerely hope that our further explanations and amendments will be satisfactory.

1) I've inspected carefully the new ITC data for 1a (Figure 2c). The integrated data points (bottom panel) appear to follow a similar pattern (same no. of points, very similar trend) as in the same Figure 2c in the previous version (revision 2).

Our response: We thank the reviewer for the constructive comment. The protein used for the ITC data in Figure 2c is different between the previous two versions. In one of the previous versions (revision 2) we used the protein GST-VHL, while in the other version we used the protein complex VBC (suggested by reviewer 1). The raw data (.itc) have been attached for reference.

2) *It is also not clear to me how very good quality data was previously obtained for ITC, FP and SPR, with the previously purified VHL protein expressed alone. One possibility is that the previously purified protein was not active / not soluble.*

Our response: We thank the reviewer for the useful suggestion. As suggested by reviewer 1, we have prepared plasmids to express and purify the VBC complex. On this basis, we did follow-up experiments, including ITC, FP, SPR and FTS.

Compared with the ITC experiment carried out by the reviewer, our experiment was performed using different equipment (ITC200 microcalorimeter for reviewer 1; MicroCal PEAQ-ITC Isothermal Titration Calorimeter for us) and materials, such as ITC buffer (buffer of 20 mM Bis-Tris propane, 100 mM NaCl, 1 mM TCEP, pH 7 for reviewer 1; 20 mM Bis-Tris, buffer of 150 mM NaCl, 2 mM DTT, 1% DMSO for us).

Similarly, compared with the experimental conditions of the reviewer, the fluorescence polarization experiment we performed also had different parameters, such as different HIF-1 α peptide (FAM-DEALAHyp-YIPMDDDFQLRSF for reviewer 1; FAM-DEALA-Hyp-YIPD for us), FP assay buffer (100 mM Bis-Tris, 100 mM NaCl, 1 mM TCEP, pH 7.0 for reviewer 1; 100 mM Tris, 100 mM NaCl, 1 mM DTT, pH 7.0 for us), plate (Corning #3820 for reviewer 1; Corning #3575 for us), and equipment (PHERAstar FS for reviewer 1; SpectraMax M5 microplate reader for us).

Meanwhile, a surface plasmon resonance (SPR) experiment using the protein complex VBC was performed by Abiocenter Biotechnology Co., Ltd. (Beijing, China), which revealed that **1a** bound to the immobilized recombinant VBC protein complex with a K_d value of 6.94 μ M, an association rate constant (k_{on}) of 2.20×10^4 (1/Ms) and a dissociation rate constant (k_{off}) of 1.53×10^{-1} (1/s) (Fig. 2f). These results are in agreement with previous data performed by ourselves. The report (including the

experimental conditions) has been attached and the results of the SPR experiment are as follows:

Fig. 2e. SPR sensorgrams showing the binding of complex **1a** to surface-immobilized VBC.

An Fluorescence-based protein thermal shift (FTS) assay has also been performed and revealed a marked positive shift of the melting curve (ca. + 4 °C) of purified VBC in the presence of complex **1a** by using the Protein Thermal Shift Dye Kit (ThermoFisher, No. 4461146). (Fig. 2c).

Fig. 2c Fluorescence-based protein thermal shift assay of VBC in the presence or absence of VH298 (100 μ M) and complex **1a** (100 μ M). Error bars represent the standard deviations (SD) of the results from obtained in triplicate. *P* values were calculated using a two-sided t test. **P* < 0.05, ***P* < 0.01 vs. DMSO group.

In summary, we suspect that the different conditions used in these assays may have led to different results. The raw data or experimental report of these assays have been attached for reference.

3) *the CD trace of VCB in the presence of 1a (red trace in Supp Fig. 6a) strongly suggests that the protein unfolds / loses substantial integrity and tertiary structure in the presence of the compound - invalidating the now presumably positive results by ITC/SPR/FP. In contrast, this CD data very much is in line with our own biophysical data, where using differential scanning Fluorimetry (thermal shift) we observed a significant negative thermal shift ($\Delta T_m = -5$ oC), consistent with destabilization / loss of structural integrity in the presence of compound 1a.*

Our response: We thank the reviewer for the useful suggestion. The FTS assay using the Protein Thermal Shift Dye Kit (ThermoFisher, No. 4461146) to verify the stabilization of VBC by complex **1a**. The results revealed a marked positive shift of the melting curve (ca. +4 °C) of purified VBC in the presence of complex **1a** (Fig. 2c). The raw data has also been attached for reference.

Fig. 2c Fluorescence-based protein thermal shift assay of VBC in the presence or absence of VH298 (100 μ M) and complex **1a** (100 μ M). Error bars represent the

standard deviations (SD) of the results from obtained in triplicate. *P* values were calculated using a two-sided t test. **P* < 0.05, ***P* < 0.01 vs. DMSO group.

The method has been updated in the revised manuscript as follows: “**Fluorescence-based protein thermal shift assay.** The protein thermal shift assay with purified protein was performed by using the Protein Thermal Shift Dye Kit (ThermoFisher, No. 4461146). Briefly, purified human recombinant VBC protein was appropriately diluted. All assay experiments used 2.5 μ L 8 \times Sypro Orange and 100 μ M of VH298 or **1a** up to a total volume of 20 μ L with VBC protein. The PCR plates were sealed with an optical seal, shaken, and centrifuged after protein and compounds were added. Thermal scanning (25 to 98 $^{\circ}$ C at 1 $^{\circ}$ C/min) was performed using a real-time PCR setup (Mx3005P Q-PCR system) and fluorescence intensity was measured after every 1 min.”

In summary, I remain unconvinced that 1a genuinely binds to VHL and inhibits the VHL-HIF-1 α interaction. I feel the authors have now properly purified soluble and active VCB for biophysical studies, but their data conflicts with my own lab's attempt to reproduce it.

Our response: We thank the reviewer for the useful suggestion. We have performed a range of experiments including CD, FTS, ITC, FP, SPR using the human recombinant VBC complex. CD measurements showed that **1a** could regulate VBC secondary structure via directly binding to the VBC complex. The FTS assay revealed that complex **1a** could promote the stabilization of VBC with a marked positive shift of the melting curve (ca. + 4 $^{\circ}$ C) (Fig. 2c). The SPR assay performed by Abiocenter Biotechnology Co., Ltd. (Beijing, China) revealed that **1a** bound to the immobilized recombinant VBC protein complex with a K_d value of 6.94 μ M, an association rate constant (k_{on}) of 2.20×10^4 (1/Ms) and a dissociation rate constant (k_{off}) of 1.53×10^{-1} (1/s) (Fig. 2f). Taken together, these data demonstrate that **1a** is able to bind to the VBC complex.

Reviewers' Comments:

Reviewer #1:

Remarks to the Author:

The authors invoke the different experimental conditions as the main reason for the differences in results between the two labs. Those experimental differences are very minor, as the conditions are essentially highly comparable - and therefore, I do not believe that differences in conditions are expected to make relevant impact on the overall results of the experiments. I so remain unconvinced.

Meanwhile the authors provide new data.

Firstly the new SPR data performed by Abiocenter Biotechnology Co., Ltd. (Beijing, China) appears to be problematic because it does not show the sensogram plot for the immobilization of the protein. One has to assume that the total level of immobilized protein is 13,000 RU, based on the report, which means that the R_{max} - i.e. the maximum signal expected from a 1:1 stoichiometric binding, assuming 100% activity of the surface (which here is not known) should be 231.4 RU (based on the MW values input from the report). However, the data shows no saturation of the signal, so makes it more difficult to accurately quantify the K_d . Yet at the highest concentration tested (12.5 μ M) the signal has already reached the R_{max} value - suggesting weak and super-stoichiometric binding.

For these reasons, I remain unconvinced by this additional SPR data. Moreover, the authors should provide evidence for the level of activity of their immobilised SPR chip surface, which they could easily measure using the HIF peptide and/or VH298 as a positive binding control in these experiments-

Secondly they present additional thermal shift data. This new data is inconsistent with our own data, where we observed destabilization of T_m with compound 1a instead.

Moreover, we observed stabilization of VBC between 5 and 7 degrees C in the presence of VH298 - which is much greater stabilization than what shown in the new data presented by the authors (compare red vs blue line, showing that hardly 1 degree C of VBC stabilization is observed in the presence of VH298)

In summary, I remain unconvinced that 1a genuinely binds to VHL and inhibits the VHL-HIF-1 α interaction, in the way as claimed by the authors. Therefore I cannot recommend publication in the journal.

Reviewer #4:

Remarks to the Author:

Leung and co-workers report the discovery of a metal complex that appears to disrupt the VHL/HIF1 α interaction by binding to VHL, and speeds wound healing in a diabetic mouse model. There were initial concerns with their biophysical characterization, which has been improved over several rounds of revisions. Most notably, the authors have started to use a VHL/ElonginB/ElonginC complex, which is the biologically relevant unit. The ITC data appear to reflect this change: in revision 2 with VHL alone, the peaks are exceedingly broad, consistent with partial aggregation of the material, while in v4, the ITC peaks are sharp, indicative of rapid binding kinetics. The results from ITC, peptide displacement and thermal upshift assays appear to be consistent to within the accuracies of the methods. I have some concerns about the quality of the SPR data in Figure 2f. The scale of the x-axis is overly-large (more than twice what is plotted in Figure S6 and about 5-fold larger than the region of interest). It appears that the fitted curves do not match the data nearly as well here as they do for peptide binding in Fig S6 (where the x-scale is appropriate). However the way the data are plotted make it very hard for the reader to judge. These data should be plotted clearly and any inconsistencies in the fits should be addressed in the text.

Secondly, I feel that the authors have somewhat overstated the novelty of this work. The statement on page 14 "Until now, very few inhibitors of the interaction between VHL and HIF-1 α have been discovered, and none of these have yet shown the ability to enhance wound healing in vivo) is not entirely true. Although the authors cited several previous studies reporting new VHL antagonists, they failed to mention 2014 J Med Chem ([dx.doi.org/10.1021/jm5011258](https://doi.org/10.1021/jm5011258)) and 2016 Nat Comm (DOI: [10.1038/ncomms13312](https://doi.org/10.1038/ncomms13312)) studies that culminated in the development of VH298, a ligand with about 10-fold stronger affinity than the one reported here. Furthermore this compound has been reported to speed wound healing in at least two different in vivo mouse models (doi.org/10.1155/2019/1897174 and doi.org/10.1016/j.bbrc.2018.09.172) This makes the authors' compound the fourth distinct scaffold for VHL antagonists and the second to promote wound healing. The authors should better place their work in the context of current research in the field.

Our Responses to the Comments of the Referees

Referee 1

Comments:

1) The authors invoke the different experimental conditions as the main reason for the differences in results between the two labs. Those experimental differences are very minor, as the conditions are essentially highly comparable - and therefore, I do not believe that differences in conditions are expected to make relevant impact on the overall results of the experiments. I so remain unconvinced.

Meanwhile the authors provide new data.

Our response: We thank the reviewer for the constructive comments. We have performed a protein thermal shift assay by fluorescence-based protein thermal shift (FTS) and a kinetic assay by biolayer interferometry (BLI) technique to validate the interaction between VBC and VH298/1a. We sincerely hope that our revised manuscript would be satisfactory for acceptance in the journal.

2) Firstly the new SPR data performed by Abiocenter Biotechnology Co., Ltd. (Beijing, China) appears to be problematic because it does not show the sensogram plot for the immobilization of the protein. One has to assume that the total level of immobilized protein is 13,000 RU, based on the report, which means that the R_{max} – i.e. the maximum signal expected from a 1:1 stoichiometric binding, assuming 100% activity of the surface (which here is not known) should be 231.4 RU (based on the MW values input from the report). However, the data shows no saturation of the signal, so makes it more difficult to accurately quantify the K_d . Yet at the highest concentration tested (12.5 μ M) the signal has already reached the R_{max} value – suggesting weak and super-stoichiometric binding.

For these reasons, I remain unconvinced by this additional SPR data. Moreover, the authors should provide evidence for the level of activity of their immobilised SPR chip surface, which they could easily measure using the HIF peptide and/or VH298 as a

positive binding control in these experiments–

Our response: We thank the reviewer for the constructive comments. We agree with the reviewer that the data shows no saturation of the signal, which makes it more difficult to accurately quantify the K_d . We have confirmed with Abiocenter Biotechnology that the exact level of immobilized protein should be 13026.1 RU. Meanwhile, the molecular weight of VBC is around 44 kDa based on the pull-down assay (Supplementary Figure 6b). Thus, the R_{max} was estimated to be 262.2 RU, rather than 231.4 RU.

In order to evaluate the kinetics, we have repeated the experiment using biolayer interferometry (BLI) and extended the association time from 60 s to 300 s and the dissociation time from 180 s to 300 s (up to 200 s was shown for facile reading). The result revealed that **1a** bound to the immobilized recombinant His-tagged VBC protein complex with K_d values of $6.74 \pm 0.19 \mu\text{M}$ (kinetic fit) and $5.80 \pm 0.64 \mu\text{M}$ (steady state fit).

Figure 2f. BLI kinetic analysis of the interaction between complex **1a** and VBC. VBC was surface-immobilized to Ni-NTA biosensors. BLI sensorgrams showing the binding of complex **1a** to surface-immobilized VBC. The K_d values for a 1:1 interaction were calculated from the kinetic fit ($K_d = 6.74 \pm 0.19 \mu\text{M}$) and steady state fit ($K_d = 5.80 \pm 0.64 \mu\text{M}$), respectively. The Ni-NTA biosensor tips coated with His-tagged VBC were dipped in increasing concentrations of **1a** (0.78, 1.56, 3.13, 6.25, and 12.5 μM) to measure binding affinity of **1a** to VBC ($k_{on} = 5.22 \times 10^2 \text{ M}^{-1}\text{s}^{-1}$) and subsequently moved to wells containing buffer to measure dissociation rates ($k_{off} = 3.52 \times 10^{-2} \text{ s}^{-1}$).

Meanwhile, as suggested, we have also studied the kinetics of HIF-1 α peptide DEALAHyp-YIPD-VBC as a positive control to validate the level of activity of the

VBC-immobilized biosensor surface (Supplementary Fig. 6f). The result demonstrated the high-affinity interaction of VBC with the HIF-1 α peptide with K_d values of $0.66 \pm 0.01 \mu\text{M}$ (kinetic fit) and $0.72 \pm 0.06 \mu\text{M}$ (steady state fit).

Supplementary Figure 6f. BLI kinetic analysis of the interaction between HIF-1 α peptide, DEALAHyp-YIPD, and VBC. VBC was surface-immobilized to Ni-NTA biosensors. BLI sensorgrams showing the binding of HIF-1 α peptide to surface-immobilized VBC. The K_d values for a 1:1 interaction were calculated from the kinetic fit ($K_d = 0.66 \pm 0.01 \mu\text{M}$) and steady state fit ($K_d = 0.72 \pm 0.06 \mu\text{M}$), respectively. The Ni-NTA biosensor tips coated with His-tagged VBC were dipped in increasing concentrations of peptide (0.19, 0.38, 0.75, 1.50, and 3.00 μM) to measure binding affinity of **1a** to VBC ($K_{\text{on}} = 5.78 \times 10^4 \text{ M}^{-1}\text{s}^{-1}$) and subsequently moved to wells containing buffer to measure dissociation rates ($K_{\text{off}} = 3.84 \times 10^{-2} \text{ s}^{-1}$).

3) Secondly they present additional thermal shift data. This new data is inconsistent with our own data, where we observed destabilization of T_m with compound **1a** instead.

Moreover, we observed stabilization of VBC between 5 and 7 degrees C in the presence of VH298 - which is much greater stabilization than what shown in the new data presented by the authors (compare red vs blue line, showing that hardly 1 degree C of VBC stabilization is observed in the presence of VH298)

Our response: We thank the reviewer for the constructive suggestion. We have evaluated again the stabilization of VBC by complex **1a** by the fluorescence-based protein thermal shift (FTS) assay using the Protein Thermal Shift Dye Kit (No. 4461146) purchased from ThermoFisher (CA, USA). Purified human recombinant VBC protein was diluted by using Protein Thermal Shift™ Buffer (1X) provided in

the Kit (No. 4461146). The assay was carried out in 5.0 μL of Protein Thermal Shift™ Buffer, 2.5 μL 8 \times Sypro Orange, 100 μM of VH298 or 100 μM of **1a** in 1.0% (v/v) DMSO solution up to a total volume of 20 μL with VBC protein. Thermal scanning (25 to 85 $^{\circ}\text{C}$ at 1 $^{\circ}\text{C}/\text{min}$) was performed using a real-time PCR setup (Mx3005P Q-PCR system) and the fluorescence intensity was measured after every one minute. Reaction samples were conducted in three replicates. The melting temperature (T_m) was then calculated by using the GraphPad Prism software. Shifts of the melting curve of VBC treated with **1a** (*ca.* 2.5 $^{\circ}\text{C}$) and VH298 (*ca.* 1.5 $^{\circ}\text{C}$) were observed (as shown in Figure C1), suggesting that both **1a** and VH298 could stabilize the VBC protein.

Figure C1. Fluorescence-based protein thermal shift assay of VBC in the presence or absence of VH298 (100 μM) and complex **1a** (100 μM) in 1.0% DMSO solution. Error bars represent the standard deviations (SD) of the results from obtained in triplicate.

In response to the reviewer's concern raised in Revision 2, "*how do the authors know that their protein was fine under 5 % DMSO?*", we further evaluated the effect of DMSO on the **1a**-induced stability of VBC by using the FTS assay (as shown in Figure C2). Compared with the melting curve of **1a** in 1% (v/v) DMSO solution, significant lower shifts of 1.2 and 3.5 $^{\circ}\text{C}$ in the melting curves were observed in the presence of 2.5% and 5% (v/v) DMSO, respectively. The results demonstrated that DMSO could have an impact on the stability of the VBC protein.

Figure C2. Fluorescence-based protein thermal shift assay of VBC in the presence or absence of **1a** (100 μ M) in 1.0%, 2.5% and 5.0% (v/v) DMSO. Error bars represent the standard deviations (SD) of the results from obtained in triplicate.

4) In summary, I remain unconvinced that 1a genuinely binds to VHL and inhibits the VHL-HIF-1 α interaction, in the way as claimed by the authors. Therefore I cannot recommend publication in the journal.

Our response:

We sincerely hope that our revised manuscript would be satisfactory for acceptance in the journal.

Referee 4

Comments:

1) Leung and co-workers report the discovery of a metal complex that appears to disrupt the VHL/HIF1 α interaction by binding to VHL, and speeds wound healing in a diabetic mouse model. There were initial concerns with their biophysical characterization, which has been improved over several rounds of revisions. Most notably, the authors have started to use a VHL/ElonginB/ElonginC complex, which is the biologically relevant unit. The ITC data appear to reflect this change: in revision 2 with VHL alone, the peaks are exceedingly broad, consistent with partial aggregation of the material, while in v4, the ITC peaks are sharp, indicative of rapid

binding kinetics. The results from ITC, peptide displacement and thermal upshift assays appear to be consistent to within the accuracies of the methods.

Our response: We thank the reviewer for the valuable comment. We agree with the reviewer that the underlying cause of the ITC result in revision 2 may be due to the aggregation of VHL protein.

2) I have some concerns about the quality of the SPR data in Figure 2f. The scale of the x-axis is overly-large (more than twice what is plotted in Figure S6 and about 5-fold larger than the region of interest). It appears that the fitted curves do not match the data nearly as well here as they do for peptide binding in Fig S6 (where the x-scale is appropriate). However the way the data are plotted make it very hard for the reader to judge. These data should be plotted clearly and any inconsistencies in the fits should be addressed in the text.

Our response: We thank the reviewer for the constructive suggestion.

In order to evaluate the kinetics, we have extended the association time from 60 s to 300 s and the dissociation time from 180 s to 300 s (up to 200 s was shown for facile reading) using biolayer interferometry (BLI). The result revealed that **1a** bound to the immobilized recombinant His-tagged VBC protein complex with K_d values of $6.74 \pm 0.19 \mu\text{M}$ (kinetic fit) and $5.80 \pm 0.64 \mu\text{M}$ (steady state fit).

Figure 2f. BLI kinetic analysis of the interaction between complex **1a** and VBC. VBC was surface-immobilized to Ni-NTA biosensors. BLI sensorgrams showing the binding of complex **1a** to surface-immobilized VBC. The K_d values for a 1:1 interaction were calculated from the kinetic fit ($K_d = 6.74 \pm 0.19 \mu\text{M}$) and steady state fit ($K_d = 5.80 \pm 0.64 \mu\text{M}$), respectively. The Ni-NTA biosensor tips coated with His-tagged VBC were dipped in increasing

concentrations of **1a** (0.78, 1.56, 3.13, 6.25, and 12.5 μM) to measure binding affinity of **1a** to VBC ($k_{\text{on}} = 5.22 \times 10^2 \text{ M}^{-1}\text{s}^{-1}$) and subsequently moved to wells containing buffer to measure dissociation rates ($k_{\text{off}} = 3.52 \times 10^{-2} \text{ s}^{-1}$).

Meanwhile, as suggested, we have also studied the kinetics of HIF-1 α peptide DEALAHyp-YIPD-VBC as a positive control to validate the level of activity of the VBC-immobilized biosensor surface (Supplementary Fig. 6f). The result demonstrated the high-affinity interaction of VBC with the HIF-1 α peptide with K_{d} values of $0.66 \pm 0.01 \mu\text{M}$ (kinetic fit) and $0.72 \pm 0.06 \mu\text{M}$ (steady state fit).

Supplementary Figure 6f. BLI kinetic analysis of the interaction between HIF-1 α peptide, DEALAHyp-YIPD, and VBC. VBC was surface-immobilized to Ni-NTA biosensors. BLI sensorgrams showing the binding of HIF-1 α peptide to surface-immobilized VBC. The K_{d} values for a 1:1 interaction were calculated from the kinetic fit ($K_{\text{d}} = 0.66 \pm 0.01 \mu\text{M}$) and steady state fit ($K_{\text{d}} = 0.72 \pm 0.06 \mu\text{M}$), respectively. The Ni-NTA biosensor tips coated with His-tagged VBC were dipped in increasing concentrations of peptide (0.19, 0.38, 0.75, 1.50, and 3.00 μM) to measure binding affinity of **1a** to VBC ($K_{\text{on}} = 5.78 \times 10^4 \text{ M}^{-1}\text{s}^{-1}$) and subsequently moved to wells containing buffer to measure dissociation rates ($K_{\text{off}} = 3.84 \times 10^{-2} \text{ s}^{-1}$).

3) Secondly, I feel that the authors have somewhat overstated the novelty of this work. The statement on page 14 “Until now, very few inhibitors of the interaction between VHL and HIF-1 α have been discovered, and none of these have yet shown the ability to enhance wound healing in vivo) is not entirely true. Although the authors cited several previous studies reporting new VHL antagonists, they failed to mention 2014 *J Med Chem* ([dx.doi.org/10.1021/jm5011258](https://doi.org/10.1021/jm5011258)) and 2016 *Nat Comm* (DOI: [10.1038/ncomms13312](https://doi.org/10.1038/ncomms13312)) studies that culminated in the development of VH298, a

ligand with about 10-fold stronger affinity than the one reported here. Furthermore this compound has been reported to speed wound healing in at least two different in vivo mouse models (doi.org/10.1155/2019/1897174 and doi.org/10.1016/j.bbrc.2018.09.172) This makes the authors' compound the fourth distinct scaffold for VHL antagonists and the second to promote wound healing. The authors should better place their work in the context of current research in the field.

Our response: We thank the reviewer for the constructive suggestion. We have rewritten the statement and the corresponding references have been cited as follows:

"Until now, very few inhibitors of the interaction between VHL and HIF-1 α have been discovered^{30, 32, 39, 47-49}, and only VH298 has been reported to promote entesis healing and wound healing *in vivo*^{50, 51}."

This manuscript was originally submitted to *Nature Communications* in August 2017. The aforementioned statement was based on the situation of three years ago. We sincerely apologize for not updating the literature in time. In the study by Qiu et al, a one-shot high-dosage of streptozotocin (STZ)-induced hyperglycemic model was used to evaluate the wound healing effect of VH298⁵³. As STZ causes acute insulin secretion failure in mouse resulting in a brief period of diabetes, it is difficult to directly draw parallels with the human condition of years/decades with diabetes⁵³.

In our study, a high-fat diet with subsequent multiple injections of a low dose of streptozotocin (HFD/STZ) in mice was used to mimic type 2 diabetes. The key advantage of this model is to mimic the slow pathogenesis of type 2 diabetes that occurs in humans, encompassing the slow development from adult-onset diet-induced obesity to glucose intolerance, insulin resistance, the resulting compensatory insulin release and finally streptozotocin-induced partial β -cell death. Additionally, the local application of complex **1a** was performed on another widely used diabetic model, *db/db* mice, to confirm that our compound accelerates wound closure in *db/db* mice.

Diabetic ulcers usually occur in humans after years or decades. Due to the limitations of animal models, we admit that the timescale for the treatment of diabetic ulcers in this study is very different compared to how diabetic ulcers develops in humans and

clinical trials will have to be carried out in the future to validate the efficacy of the compound.

Reviewers' Comments:

Reviewer #1:

Remarks to the Author:

The authors have now dropped the SPR data previously included in the previous version, and following the technical issues noticed and highlighted by the reviewers at the time. They have now replaced that old data with some new BLI data instead. Their BLI assay was first benchmarked with the HIF-1 α 10-mer peptide, which show a K_d of 600nM for VHL, consistent with previous data from the literature. Using the same technique, they then obtain reasonably good data showing a K_d of 6-7 μ M (new Fig. 2f). It is unclear if the technical issue from the SPR assay were resolved or not, but one would assume it has not. The BLI data, although not deemed to be to the same level of "gold standardness" as SPR, seems nonetheless acceptable. The authors should, however, remove any reference to SPR carried over from previous versions of the main text (there is still a reference to SPR in the Discussion).

With regards to the thermal shift data as differential scanning fluorimetry (DSF) – I remain unconvinced by the shown results. The high-affinity VH298 inhibitor (K_d 80nM) should yield a much greater ΔT_m stabilization than what shown by the authors, especially when used at such relatively high concentration (100 μ M) as reported by them. Considering their compound is almost 100-fold weaker binder than VH298 (compare K_d BLI = 6 μ M for 1a versus K_d 80nM for VH298 as reported in literature), VH298 is expected to shift the T_m of the VBC complex by a much greater extent than 1a. Yet, Fig. C1 shows the contrary i.e. 1a shifting more than VH298 – making it look as though 1a is a better binder than VH298, albeit shifts are relatively small for both compounds. As previously explained, we in our Lab reproducibly observe stabilization of VBC between 5 and 7 degrees C in the presence of VH298.

Finally, I very much agree with Reviewer 4 that: "the authors have somewhat overstated the novelty of this work.... Although the authors cited several previous studies reporting new VHL antagonists, they failed to mention 2014 J Med Chem ([dx.doi.org/10.1021/jm5011258](https://doi.org/10.1021/jm5011258)) and 2016 Nat Comm (DOI: 10.1038/ncomms13312) studies that culminated in the development of VH298, a ligand with about 10-fold stronger affinity than the one reported here." Based on the new BLI data presented in this revised version, actually, 1a is \sim 100-fold weaker affinity binder to VHL than VH298. The authors should acknowledge this fact in a much more scholarly fashion, in my opinion, and should be upfront in acknowledging the priority development of bona-fide VHL-HIF protein-protein interaction inhibitor VH298 as high-quality chemical probe of the VHL – HIF axis in the hypoxia signaling pathway (Galdeano et al. J Med Chem 2014; Frost et al. Nat Commun 2016; Soares et al. J Med Chem 2018 <https://doi.org/10.1021/acs.jmedchem.7b00675>).

The above issues should be satisfactorily resolved before publication.

Reviewer #4:

Remarks to the Author:

The authors have addresses all of my concerns. I would note that I don't think that the BLI data give realistic kinetics, but the affinity results, which are the focus of the paper, appear consistent and convincing.

**A small molecule HIF-1 α stabilizer that accelerates diabetic wound healing**

Guodong Li,^{1,#} Chung-Nga Ko,^{2,#} Dan Li,^{1,#} Chao Yang,^{1,#} Wanhe Wang,² Guan-Jun Yang,¹ Carmelo Di
Primo,^{3,4} Vincent Kam Wai Wong,⁵ Yaozu Xiang,⁶ Ligen Lin,^{1,*} Dik-Lung Ma,^{2,*} and Chung-Hang
Leung^{1,*}

¹ State Key Laboratory of Quality Research in Chinese Medicine, Institute of Chinese Medical
Sciences, University of Macau, Macao, China. *E-mail: duncanleung@um.edu.mo; ligenl@um.edu.mo

² Department of Chemistry, Hong Kong Baptist University, Kowloon Tong, Hong Kong, China.

*E-mail: edmondma@hkbu.edu.hk

³ Université de Bordeaux, Laboratoire ARNA, 146 rue Léo Saignat, Bordeaux, France.

[revised manuscript text omitted]

of silver triflate in 25 mL acetonitrile and stirred at room temperature under a nitrogen
atmosphere for 15 h. The mixture was filtered and washed with two portions of ether (2×30
384 mL) to yield titled product.

**General synthesis of $[M(C^{\wedge}N)_2(N^{\wedge}N)]PF_6$ complexes.** These complexes were synthesized
using a modified literature method⁵⁶. The precursor Ir(III) complex dimers were prepared as
previously reported. Briefly, a suspension of $[M_2(C^{\wedge}N)_4Cl_2]$ (0.2 mM) and corresponding
$N^{\wedge}N$ (0.44 mM) ligands in a mixture of dichloromethane:methanol (1:1, 20 mL) was refluxed
overnight under a nitrogen atmosphere. The resulting solution was allowed to cool to room
temperature, and was filtered to remove the unreacted cyclometalated dimer. To the filtrate, an
aqueous solution of ammonium hexafluorophosphate (excess) was added and the filtrate was
reduced in volume by rotary evaporation until precipitation of the crude product occurred.
The precipitate was then filtered and washed with several portions of water (2×50 mL)
followed by diethyl ether (2×50 mL). The product was recrystallized by acetonitrile:diethyl
ether vapor diffusion to yield the titled compound. Complexes **1–14** and **1a–1m** were
characterized by 1H -NMR, ^{13}C -NMR, high resolution mass spectrometry (HRMS) and
elemental analysis.

**Complex 1.** Yield: 72%. 1H NMR (400 MHz, Acetone- d_6) δ 8.43 (s, 2H), 8.35 – 8.27 (m, 4H),

8.04 (t, $J = 8.2$ Hz, 2H), 7.96 (d, $J = 8.4$ Hz, 2H), 7.73 (d, $J = 5.8$ Hz, 2H), 7.52 (d, $J = 6.1$ Hz,
2H), 7.35 (d, $J = 8.3$ Hz, 2H), 7.13 (t, $J = 7.4$ Hz, 2H), 6.51 (d, $J = 2.0$ Hz, 2H), 4.26 (s, 6H).
401 ^{13}C NMR (101 MHz, Acetone) δ 170.87, 170.54, 165.11, 164.86, 153.06, 150.30, 146.81,
402 144.09, 139.95, 135.75, 127.51, 127.20, 125.27, 125.24, 125.01, 124.02, 121.64, 121.40,
121.38, 107.64, 57.86. MALDI-TOF-HRMS: Calcd. for $\text{C}_{36}\text{H}_{26}\text{Br}_2\text{N}_4\text{O}_2\text{Rh} [\text{M}-\text{PF}_6]^+$:
808.9457 Found: 808.9435. Anal.: ($\text{C}_{36}\text{H}_{26}\text{Br}_2\text{N}_4\text{O}_2\text{RhPF}_6$) C, H, N: calcd. 45.31, 2.75, 5.87;
found 45.13, 2.72, 5.76.

**Complex 2.** Yield: 81%. ^1H NMR (400 MHz, Acetone- d_6) δ 9.16 (d, $J = 13.8$ Hz, 2H), 8.65
(dd, $J = 8.1, 1.4$ Hz, 2H), 8.34 – 8.24 (m, 4H), 8.01 (d, $J = 8.8$ Hz, 2H), 7.96 – 7.85 (m, 4H),
7.66 – 7.58 (m, 4H), 7.24 (t, $J = 7.6$ Hz, 2H), 6.36 (d, $J = 7.2$ Hz, 2H), 4.30 – 4.06 (m, 8H),
1.33 – 1.27 (m, 12H). ^{13}C NMR (101 MHz, CDCl_3) δ 164.55, 164.22, 155.88, 155.75, 155.12,
410 152.10, 151.97, 149.96, 143.85, 142.01, 140.50, 138.32, 134.96, 131.04, 130.49, 130.43,
411 130.34, 128.37, 127.17, 127.06, 124.88, 123.64, 122.60, 63.95, 63.93, 63.9016.66, 16.64,
16.61, 16.58. MALDI-TOF-HRMS: Calcd. for $\text{C}_{44}\text{H}_{42}\text{N}_4\text{O}_6\text{P}_2\text{Rh} [\text{M}-\text{PF}_6]^+$: 887.1635 Found:
886.1672. Anal.: ($\text{C}_{44}\text{H}_{42}\text{N}_4\text{O}_6\text{P}_2\text{RhPF}_6$) C, H, N: calcd. , 51.18, 4.10, 5.43; found 50.88, 4.20,
5.45.

**Complex 3.** Yield: 68%. ^1H NMR (400 MHz, Acetone- d_6) δ 8.82 (d, $J = 8.2$ Hz, 2H), 8.32 (t,
$J = 7.9$ Hz, 2H), 8.22 – 8.17 (m, 2H), 8.11 (t, $J = 8.2$ Hz, 2H), 7.99 – 7.93 (m, 2H), 7.91 (d, J
= 8.6 Hz, 2H), 7.74 – 7.70 (m, 4H), 7.10 (t, $J = 7.3$ Hz, 2H), 6.70 (dd, $J = 8.6, 2.5$ Hz, 2H),

5.86 (dd, $J = 2.5, 1.1$ Hz, 2H), 3.59 (s, 6H). ^{13}C NMR (101 MHz, Acetone) δ 170.59, 170.27,
419 165.83, 165.82, 161.58, 161.56, 155.57, 151.16, 149.78, 140.78, 139.39, 137.54, 128.86,
126.97, 125.01, 123.19, 120.18, 120.17, 118.94, 109.39, 55.07. MALDI-TOF-HRMS: Calcd.
for $\text{C}_{34}\text{H}_{28}\text{N}_4\text{O}_2\text{Rh} [\text{M}-\text{PF}_6]^+$: 627.1267 Found: 627.1266. Anal.: ($\text{C}_{34}\text{H}_{28}\text{N}_4\text{O}_2\text{RhPF}_6$) C, H, N:
calcd. 52.86, 3.65, 7.25; found 52.62, 3.66, 7.36.

**Complex 4.** Yield: 79%. ^1H NMR (400 MHz, Acetone- d_6) δ 8.52 (s, 2H), 8.31 (d, $J = 5.1$ Hz,
2H), 8.25 (d, $J = 8.2$ Hz, 2H), 7.97 (t, $J = 7.8$ Hz, 4H), 7.91 – 7.85 (m, 2H), 7.63 (d, $J = 5.7$
425 Hz, 2H), 7.14 (t, $J = 7.6$ Hz, 2H), 7.08 – 6.99 (m, 4H), 6.45 (d, $J = 7.6$ Hz, 2H), 3.00 – 2.95
(m, 6H). ^{13}C NMR (101 MHz, Acetone) δ 168.81, 168.49, 166.01, 150.94, 150.08, 149.85,
427 146.22, 145.01, 139.48, 133.68, 131.27, 130.91, 127.97, 125.56, 125.19, 124.31, 124.24,
120.86, 19.19. MALDI-TOF-HRMS: Calcd. for $\text{C}_{36}\text{H}_{28}\text{N}_4\text{Rh} [\text{M}-\text{PF}_6]^+$: 619.1369 Found:
619.1404. Anal.: ($\text{C}_{36}\text{H}_{28}\text{F}_6\text{N}_4\text{PRh}$) C, H, N: calcd. , 56.56, 3.69, 7.23; found 56.36, 3.77,
7.39.

**Complex 5.** Yield: 68%. ^1H NMR (400 MHz, Acetone- d_6) δ 8.55 (d, $J = 5.2$ Hz, 2H), 8.27 (s,
2H), 8.23 (d, $J = 8.3$ Hz, 2H), 8.03 – 7.95 (m, 4H), 7.92 (d, $J = 8.0$ Hz, 2H), 7.76 (d, $J = 5.8$
433 Hz, 2H), 7.70 – 7.61 (m, 10H), 7.07 – 6.98 (m, 4H), 6.35 (s, 2H), 2.44 (q, $J = 7.6$ Hz, 4H),
1.05 (t, $J = 7.6$ Hz, 6H). ^{13}C NMR (101 MHz, Acetone) δ 168.92, 168.60, 166.20, 166.19,
435 151.74, 151.10, 150.22, 147.15, 147.11, 142.66, 139.42, 136.92, 132.86, 130.63, 130.51,
129.95, 129.67, 127.66, 126.73, 125.53, 124.13, 123.80, 120.55, 15.43. MALDI-TOF-HRMS:

Calcd. for $C_{50}H_{40}N_4Rh [M-PF_6]^+$: 799.2308 Found: 799.2307. Anal.: ($C_{50}H_{40}N_4RhPF_6$) C, H,
438 N: calcd. 63.57, 4.27, 5.93; found 63.54, 4.34, 6.02.

**Complex 6.** Yield: 31%. 1H NMR (400 MHz, Acetone- d_6) δ 8.56 (d, $J = 8.7$ Hz, 2H), 8.46 (d,
$J = 8.9$ Hz, 2H), 8.20 (d, $J = 8.0$ Hz, 2H), 8.14 (d, $J = 6.3$ Hz, 2H), 7.93 (d, $J = 8.1$ Hz, 2H),
7.86 (d, $J = 2.6$ Hz, 2H), 7.62 (d, $J = 8.9$ Hz, 2H), 7.44 (t, $J = 8.0$ Hz, 2H), 7.24 – 7.15 (m,
4H), 7.09 (d, $J = 8.0$ Hz, 2H), 6.44 (d, $J = 1.3$ Hz, 2H), 3.94 (s, 6H), 2.25 (q, $J = 7.6$ Hz, 4H),
0.80 (t, $J = 7.6$ Hz, 6H). ^{13}C NMR (101 MHz, Acetone) δ 169.94, 168.87, 167.25, 156.87,
444 149.93, 147.81, 147.17, 144.17, 140.37, 135.44, 131.14, 129.83, 128.89, 127.77, 127.35,
126.04, 124.47, 118.83, 114.30, 110.69, 56.96, 15.27. MALDI-TOF-HRMS: Calcd. for
$C_{46}H_{40}N_4O_2Rh [M-PF_6]^+$: 783.2209 Found: 783.2269. Anal.: ($C_{46}H_{40}N_4O_2RhPF_6 + 3H_2O$) C,
H, N: calcd. 56.22, 4.72, 5.70; found 55.95, 4.29, 5.61.

**Complex 7.** Yield: 35%. 1H NMR (400 MHz, Acetone- d_6) δ 8.86 (d, $J = 1.7$ Hz, 2H), 8.68 (t,
$J = 5.5$ Hz, 2H), 8.65 – 8.51 (m, 4H), 8.34 (d, $J = 7.9$ Hz, 2H), 8.10 (d, $J = 5.5$ Hz, 2H), 7.93
(d, $J = 8.1$ Hz, 2H), 7.52 (q, $J = 8.9$ Hz, 2H), 7.43 (dd, $J = 8.0, 6.9$ Hz, 2H), 7.34 – 7.23 (m,
2H), 7.18 (dd, $J = 8.7, 6.9$ Hz, 2H), 6.94 (dd, $J = 7.8, 7.2$ Hz, 2H), 6.60 (t, $J = 7.9$ Hz, 2H),
3.93 (s, 6H). ^{13}C NMR (101 MHz, Acetone) δ 168.10, 167.80, 167.25, 164.40, 155.69, 150.41,
453 147.50, 146.50, 141.47, 141.04, 136.00, 131.83, 131.31, 131.29, 130.11, 129.13, 128.24,
127.88, 127.83, 125.70, 125.08, 124.12, 119.30, 53.68. MALDI-TOF-HRMS: Calcd. for
$C_{44}H_{32}N_4O_4Rh [M-PF_6]^+$: 783.1479 Found: 783.1495. Anal.: ($C_{44}H_{32}N_4O_4RhPF_6 + 2H_2O$) C,

H, N: calcd. , 54.78, 3.76, 5.81; found 54.65, 3.48, 5.71.

**Complex 8.** Yield: 71%. ^1H NMR (400 MHz, Acetone- d_6) δ 8.51 (d, J = 8.0 Hz, 2H), 8.13 (d,
J = 7.2 Hz, 2H), 8.09 (d, J = 7.9 Hz, 2H), 8.00 (d, J = 7.8 Hz, 2H), 7.84 (d, J = 7.6 Hz, 2H),
7.47 (d, J = 7.6 Hz, 2H), 7.10 (dd, J = 7.7, 5.8 Hz, 2H), 6.95 (t, J = 7.2 Hz, 2H), 6.72 (t, J =
7.5 Hz, 2H), 6.19 (dd, J = 7.8, 1.4 Hz, 2H), 2.87 (s, 6H), 1.87 (s, 6H). ^{13}C NMR (101 MHz,
Acetone) δ 153.88, 149.62, 149.60, 149.59, 149.57, 143.20, 140.48, 132.72, 129.47, 129.44,
129.43, 129.27, 129.23, 123.29, 123.28, 122.55, 26.36, 23.26. MALDI-TOF-HRMS: Calcd.
for $\text{C}_{36}\text{H}_{32}\text{IrN}_4 [\text{M}-\text{PF}_6]^+$: 713.2256 Found: 713.2238. Anal.: ($\text{C}_{36}\text{H}_{32}\text{IrN}_4\text{PF}_6 + 2\text{H}_2\text{O}$) C, H, N:
calcd. 48.37, 4.06, 6.27; found 48.17, 3.64, 6.29.

**Complex 9.** Yield: 55%. ^1H NMR (400 MHz, Acetone- d_6) δ 9.42 (s, 1H), 9.34 (d, J = 8.8 Hz,
1H), 9.23 (d, J = 8.3, 1H), 8.78 (dd, J = 15.8, 5.1 Hz, 2H), 8.45 – 8.33 (m, 2H), 8.27 (ddd, J =
10.3, 8.5, 5.1 Hz, 2H), 8.07 – 7.94 (m, 2H), 7.82 (t, J = 5.6 Hz, 2H), 7.05 (dd, J = 7.4, 5.8 Hz,
2H), 6.82 (dd, J = 12.1, 9.3 Hz, 2H), 5.88 (d, J = 8.6 Hz, 2H). ^{13}C NMR (101 MHz, Acetone)
δ 154.68, 153.19, 150.21, 148.28, 147.29, 141.23, 139.83, 135.74, 129.14, 128.55, 127.24,
124.61, 124.02, 123.73, 123.53, 114.01, 113.84. MALDI-TOF-HRMS: Calcd. for
$\text{C}_{34}\text{H}_{19}\text{F}_4\text{IrN}_5\text{O}_2 [\text{M}-\text{PF}_6]^+$: 798.1104 Found: 798.0186. Anal.: ($\text{C}_{34}\text{H}_{19}\text{F}_4\text{IrN}_5\text{O}_2\text{PF}_6$) C, H, N:
calcd. 43.32, 2.03, 7.43; found 43.46, 1.87, 7.42.

**Complex 10.** Yield: 70%. ^1H NMR (400 MHz, DMSO- d_6) δ 9.58 (d, J = 8.3 Hz, 2H), 8.88 (d,
J = 2.9 Hz, 2H), 8.41 (d, J = 5.1 Hz, 2H), 8.13 (dd, J = 8.3, 5.2 Hz, 2H), 7.72 (d, J = 8.1 Hz,

2H), 7.16 (d, $J = 2.3$ Hz, 2H), 7.09 (t, $J = 7.7$ Hz, 2H), 6.90 (t, $J = 7.4$ Hz, 2H), 6.62 (t, $J = 2.8$,
2H), 6.29 (d, $J = 7.5$ Hz, 2H), 3.31 -3.25 (m, 4H), 2.18 – 2.01 (m, 4H). ^{13}C NMR (101 MHz,
DMSO- d_6) δ 155.86, 151.61, 148.19, 143.09, 139.20, 136.52, 134.57, 132.63, 131.62, 129.55,
128.66, 127.66, 126.38, 123.13, 112.11, 108.36, 32.47, 21.94. MALDI-TOF-HRMS: Calcd.
for $\text{C}_{36}\text{H}_{28}\text{IrN}_8 [\text{M-PF}_6]^+$: 765.2066 Found: 765.2020. Anal.: ($\text{C}_{36}\text{H}_{28}\text{F}_6\text{IrN}_8\text{P}$) C, H, N: calcd.
47.52, 3.10, 12.32; found 47.66, 3.19, 12.45.

**Complex 11.** Yield: 72%. ^1H NMR (400 MHz, DMSO- d_6) δ 9.63 (d, $J = 8.3$ Hz, 2H), 9.27 (s,
1H), 8.33 – 8.25 (m, 4H), 8.17 (dd, $J = 8.3, 5.2$ Hz, 2H), 7.96 (d, $J = 8.0$ Hz, 2H), 7.89 (t, $J =$
8.2 Hz, 2H), 7.61 – 7.54 (m, 2H), 7.07 (t, $J = 7.6$ Hz, 2H), 7.03 – 6.93 (m, 4H), 6.29 (d, $J =$
7.7 Hz, 2H), 2.92 (s, 3H). ^{13}C NMR (101 MHz, DMSO) δ 166.76, 156.31, 151.61, 151.32,
485 149.70, 149.49, 149.45, 148.08, 147.59, 147.20, 144.02, 138.75, 138.24, 136.81, 134.80,
486 134.56, 131.17, 130.27, 130.02, 129.86, 128.20, 128.10, 125.08, 123.81, 122.45, 119.96,
22.23. MALDI-TOF-HRMS: Calcd. for $\text{C}_{37}\text{H}_{26}\text{IrN}_6 [\text{M-PF}_6]^+$: 747.1848 Found: 747.1835.
Anal.: ($\text{C}_{37}\text{H}_{26}\text{IrN}_6\text{PF}_6 + 2\text{H}_2\text{O}$) C, H, N: calcd. , 47.90, 3.26, 9.06; found 47.76, 3.13, 9.22.

**Complex 12.** Yield: 60%. ^1H NMR (400 MHz, Acetone- d_6) δ 8.68 (d, $J = 1.7$ Hz, 2H), 8.13
(dd, $J = 8.2, 1.4$ Hz, 2H), 7.97 (d, $J = 5.7$ Hz, 2H), 7.94 (dd, $J = 8.0, 1.3$ Hz, 2H), 7.83 (t, $J =$
7.8 Hz, 2H), 7.49 (dd, $J = 5.7, 1.7$ Hz, 2H), 7.14 – 7.02 (m, 4H), 6.87 (t, $J = 7.4, 1.3$ Hz, 2H),
6.45 (d, $J = 7.6$ Hz, 2H), 2.86 – 2.79 (m, 4H), 1.89 (s, 6H), 1.75 – 1.63 (m, 4H), 1.33 – 1.23
(m, 24H), 0.89 – 0.84 (m, 6H). ^{13}C NMR (101 MHz, Acetone) δ 169.60, 162.51, 156.94,

494 156.84, 149.59, 149.13, 146.52, 139.94, 133.85, 130.47, 128.86, 126.26, 125.31, 124.74,
495 123.56, 118.47, 35.78, 32.57, 30.85, 30.17, 30.09, 29.99, 29.95, 29.90, 29.81, 26.36, 23.31,
14.35. MALDI-TOF-HRMS: Calcd. for $C_{52}H_{64}IrN_4 [M-PF_6]^+$: 937.4760 Found: 937.5958.
Anal.: ($C_{52}H_{64}F_6IrN_4P$) C, H, N: calcd. 57.71, 5.96, 5.18; found 57.87, 6.05, 5.16.

**Complex 13.** Yield: 46% 1H NMR (400 MHz, DMSO- d_6) δ 9.63 (d, J = 8.2 Hz, 2H), 9.10 (d,
J = 8.1 Hz, 2H), 8.41 – 8.31 (m, 4H), 8.29 (d, J = 8.4 Hz, 2H), 8.19 (dd, J = 8.3, 5.0 Hz, 2H),
7.95 – 7.91 (m, 4H), 7.71 (d, J = 5.9 Hz, 2H), 7.60 (d, J = 7.6 Hz, 2H), 7.55 (t, J = 8.1 Hz,
2H), 7.05 (t, J = 6.8 Hz, 2H), 6.96 (d, J = 8.3 Hz, 2H), 6.21 – 6.14 (m, 2H), 2.16 (s, 6H). ^{13}C
NMR (101 MHz, DMSO) δ 166.93, 151.60, 150.06, 149.30, 148.23, 141.43, 140.87, 139.73,
503 138.76, 137.62, 135.01, 131.95, 131.32, 130.90, 129.98, 128.23, 128.11, 125.62, 125.16,
123.58, 123.34, 123.22, 119.72, 21.62. MALDI-TOF-HRMS: Calcd. for $C_{50}H_{34}IrN_6 [M-PF_6]^+$:
911.2474 Found: 911.2397. Anal.: ($C_{50}H_{34}IrN_6PF_6 + 2.5H_2O$) C, H, N: calcd. 54.54, 3.57, 7.63;
found 54.35, 3.07, 7.35.

**Complex 14.** Yield: 50%. 1H NMR (400 MHz, Acetone- d_6) δ 9.18 (d, J = 2.0 Hz, 2H), 8.20 (d,
J = 8.3 Hz, 2H), 8.15 (d, J = 6.0 Hz, 2H), 8.04 (dd, J = 5.9, 2.0 Hz, 2H), 7.98 (d, J = 7.7 Hz,
2H), 7.47 (dd, J = 8.2, 7.3 Hz, 2H), 7.26 (dd, J = 8.5, 7.3 Hz, 2H), 7.13 (t, J = 7.5 Hz, 2H),
6.92 (t, J = 7.5 Hz, 2H), 6.47 (t, J = 8.4 Hz, 2H), 6.42 (t, J = 7.6 Hz, 2H). ^{13}C NMR (101 MHz,
Acetone) δ 182.56, 157.64, 152.51, 150.28, 150.00, 141.21, 137.36, 134.06, 133.53, 132.99,
132.54, 129.70, 129.19, 127.74, 127.09, 125.09, 124.35, 118.59. MALDI-TOF-HRMS: Calcd.

for $C_{36}H_{23}Br_2IrN_4S_2[M-PF_6]^+$: 926.9261 Found: 926.9214. Anal.: ($C_{36}H_{23}Br_2IrN_4S_2PF_6$) C, H,
514 N: calcd. 40.35, 2.07, 5.23; found 40.45, 2.02, 5.16.

**Complex 1a.** Yield: 59%. 1H NMR (400 MHz, Acetone- d_6) δ 8.42 (s, 2H), 8.22 (dt, $J = 8.3$,
1.2 Hz, 2H), 8.17 (d, $J = 6.1$ Hz, 2H), 7.94 – 7.86 (m, 4H), 7.75 (ddd, $J = 5.8, 1.5, 0.8$ Hz, 2H),
7.52 (d, $J = 6.1$ Hz, 2H), 7.08 – 6.96 (m, 4H), 6.93 (td, $J = 7.4, 1.4$ Hz, 2H), 6.45 (dd, $J = 7.5$,
1.2 Hz, 2H), 4.25 (s, 6H). ^{13}C NMR (101 MHz, Acetone- d_6) δ 205.36, 167.94, 163.95, 152.39,
519 150.58, 149.18, 147.29, 144.38, 138.32, 131.88, 130.18, 124.81, 123.59, 123.29, 122.23,
520 121.06, 119.66, 106.87, 57.02, 29.56, 29.37, 29.23, 29.17, 29.04, 28.98, 28.79, 28.60, 28.40.
HRMS [$C_{36}H_{28}IrN_4O_2$] $^+$ calculated: 741.1842, found: 741.1884. Anal. ($C_{36}H_{28}O_2IrN_4PF_6$) C,
H, N: calculated: 48.81, 3.19, 6.32; found: 48.53, 3.20, 6.30.

**Complex 1b.** Yield: 68%. 1H NMR (400 MHz, Acetone- d_6) δ 8.41 (s, 1H), 8.26 – 8.20 (m, 2H),
8.00 – 7.94 (m, 2H), 7.73 – 7.68 (m, 1H), 7.49 (d, $J = 6.0$ Hz, 1H), 7.12 (td, $J = 7.5, 1.2$ Hz,
1H), 7.08 – 6.98 (m, 2H), 6.45 (dt, $J = 7.3, 1.1$ Hz, 1H), 4.24 (s, 3H), ^{13}C NMR (101 MHz,
Acetone) δ 166.02, 164.92, 152.76, 150.15, 146.92, 145.07, 139.39, 133.74, 130.81, 125.48,
124.27, 124.13, 123.89, 121.56, 120.77, 120.76, 107.49, 57.81; MALDI-TOF-HRMS: Calcd.
for $C_{36}H_{28}N_4Rh[M-PF_6]^+$: 651.1261 Found: 651.1235; Anal. ($C_{36}H_{28}N_4RhPF_6$) C, H, N: calcd
54.29, 3.54, 7.03; found 54.01, 3.73, 7.01.

**Complex 1c.** Yield: 63%. 1H NMR (400 MHz, Acetone- d_6) δ 8.42 (s, 2H), 8.11 (dd, $J = 8.2$,
1.3 Hz, 2H), 8.06 (d, $J = 6.1$ Hz, 2H), 7.73 (d, $J = 6.1$ Hz, 4H), 7.49 (d, $J = 6.1$ Hz, 2H), 7.07

(ddd, $J = 8.2, 7.2, 1.4$ Hz, 2H), 6.93 – 6.84 (m, 4H), 6.42 (dd, $J = 7.6, 1.4$ Hz, 2H), 4.25 (s,
6H), 2.88 (s, 6H). ^{13}C NMR (100 MHz, Acetone- d_6) δ 206.17, 170.66, 170.34, 164.88, 163.89,
534 163.87, 152.41, 148.22, 146.89, 146.75, 143.03, 133.81, 133.80, 133.51, 129.73, 129.72,
535 129.30, 123.90, 123.67, 123.32, 121.54, 117.65, 107.39, 57.76, 30.42, 30.22, 30.03, 29.84,
29.65, 29.45, 29.26, 23.26. HRMS $[\text{C}_{38}\text{H}_{32}\text{IrN}_4\text{O}_2]^+$ calculated: 769.2155, found: 769.2194.
Anal. ($\text{C}_{38}\text{H}_{32}\text{O}_2\text{IrN}_4\text{PF}_6 \cdot 2\text{H}_2\text{O}$) C, H, N: calculated: 48.05, 3.82, 5.90; found: 47.96, 3.54,
6.08.

**Complex 1d**. Yield: 61%. ^1H NMR (400 MHz, Acetone- d_6) δ 8.40 (s, 2H), 8.20 – 8.11 (m, 4H),
7.84 (ddd, $J = 8.2, 7.4, 1.5$ Hz, 2H), 7.79 (d, $J = 8.0$ Hz, 2H), 7.69 (ddd, $J = 5.9, 1.6, 0.8$ Hz,
2H), 7.50 (d, $J = 6.1$ Hz, 2H), 6.97 – 6.85 (m, 4H), 6.31 – 6.24 (m, 2H), 4.25 (s, 6H), 2.11 (s,
6H). ^{13}C NMR (100 MHz, Acetone- d_6) δ 206.16, 206.14, 168.86, 164.77, 153.22, 151.80,
543 149.85, 148.14, 142.63, 140.84, 139.00, 133.48, 125.62, 124.41, 124.11, 123.56, 121.90,
120.11, 107.71, 57.86, 30.42, 30.23, 30.04, 29.85, 29.65, 29.46, 29.27, 21.83. HRMS
$[\text{C}_{38}\text{H}_{32}\text{IrN}_4\text{O}_2]^+$ calculated: 769.2155, found: 769.2197. Anal. ($\text{C}_{38}\text{H}_{32}\text{O}_2\text{IrN}_4\text{PF}_6$) C, H, N:
calculated: 49.94, 3.53, 6.13; found: 49.47, 3.48, 6.07.

**Complex 1e**. Yield: 66%. ^1H NMR (400 MHz, Acetone- d_6) δ 8.67 (s, 2H), 8.46 (d, $J = 5.5$ Hz,
2H), 8.41 (d, $J = 5.6$ Hz, 2H), 8.29 (dt, $J = 8.3, 1.3$ Hz, 2H), 7.99 – 7.89 (m, 4H), 7.81 (ddd, J
= 5.8, 1.5, 0.7 Hz, 2H), 7.29 (dd, $J = 8.4, 2.0$ Hz, 2H), 7.03 (ddd, $J = 7.3, 5.8, 1.4$ Hz, 2H),
6.45 (d, $J = 2.0$ Hz, 2H). ^{13}C NMR (100 MHz, Acetone- d_6) δ 167.21, 152.95, 152.25, 150.89,

551 148.32, 144.46, 140.15, 137.52, 134.76, 132.72, 132.59, 129.18, 127.68, 126.95, 125.76,
125.10, 121.40, 30.42, 30.23, 30.04, 29.90, 29.85, 29.65, 29.46, 29.27. HRMS
$[\text{C}_{34}\text{H}_{20}\text{IrN}_4\text{Br}_4]^+$ calculated: 996.8010, found: 996.7966. Anal. ($\text{C}_{34}\text{H}_{20}\text{Br}_4\text{IrN}_4\text{PF}_6$) C, H, N:
calculated: 35.78, 1.77, 4.91; found: 35.94, 3.48, 6.07.

**Complex 1f.** Yield: 54% ^1H NMR (400 MHz, Acetone- d_6) δ 8.43 (s, 2H), 8.32 – 8.22 (m, 4H),
7.97 – 7.87 (m, 4H), 7.76 (ddd, $J = 5.8, 1.5, 0.8$ Hz, 2H), 7.54 (d, $J = 6.1$ Hz, 2H), 7.27 (dd, J
= 8.3, 2.0 Hz, 2H), 7.06 (ddd, $J = 7.3, 5.8, 1.4$ Hz, 2H), 6.50 (d, $J = 2.0$ Hz, 2H), 4.26 (s, 6H).
558 ^{13}C NMR (100 MHz, Acetone- d_6) δ 166.87, 164.45, 152.93, 149.62, 147.40, 143.95, 139.15,
559 134.23, 126.88, 125.78, 125.00, 124.25, 123.95, 121.39, 120.52, 117.04, 107.24, 57.32.

HRMS $[\text{C}_{36}\text{H}_{26}\text{O}_2\text{IrN}_4\text{Br}_2]^+$ calculated: 899.0021, found: 899.0053. Anal.
($\text{C}_{36}\text{H}_{26}\text{Br}_2\text{IrN}_4\text{O}_2\text{PF}_6$) C, H, N: calculated: 41.43, 2.51, 5.37; found: 41.28, 2.59, 5.41.

**Complex 1g.** Yield: 54%. ^1H NMR (400 MHz, Acetone- d_6) δ 8.61 (d, $J = 5.3$ Hz, 2H), 8.39 –
8.33 (m, 2H), 8.31 (s, 2H), 8.07 (d, $J = 5.3$ Hz, 2H), 8.03 – 7.94 (m, 4H), 7.85 (dt, $J = 5.7, 1.2$
564 Hz, 2H), 7.73 – 7.62 (m, 10H), 7.32 (dd, $J = 8.4, 2.0$ Hz, 2H), 7.11 (ddd, $J = 7.3, 5.8, 1.4$ Hz,
2H), 6.53 (d, $J = 2.0$ Hz, 2H). ^{13}C NMR (100 MHz, Acetone- d_6) δ 206.16, 167.43, 153.40,
566 152.10, 151.96, 150.61, 148.35, 144.50, 140.06, 136.60, 134.75, 130.74, 130.68, 130.45,
567 130.01, 128.25, 127.70, 127.21, 126.73, 125.78, 125.13, 121.43, 30.42, 30.23, 30.04, 29.85,
29.65, 29.46, 29. HRMS $[\text{C}_{46}\text{H}_{30}\text{IrN}_4\text{Br}_2]^+$ calculated: 996.0449, found: 996.0440. Anal.
($\text{C}_{46}\text{H}_{30}\text{Br}_2\text{IrN}_4\text{PF}_6$) C, H, N: calculated: 48.65, 2.66, 4.93; found: 49.20, 3.05, 5.00.

**Complex 1h.** Yield: 64%. ^1H NMR (400 MHz, Acetone- d_6) δ 8.95 (dd, $J = 8.3, 1.4$ Hz, 2H),
8.53 (dd, $J = 5.1, 1.4$ Hz, 2H), 8.44 (s, 2H), 8.33 – 8.27 (m, 2H), 8.11 (dd, $J = 8.3, 5.0$ Hz, 2H),
8.00 – 7.90 (m, 4H), 7.70 (ddd, $J = 5.8, 1.6, 0.8$ Hz, 2H), 7.31 (dd, $J = 8.3, 2.0$ Hz, 2H), 7.05
(ddd, $J = 7.4, 5.8, 1.4$ Hz, 2H), 6.51 (d, $J = 2.0$ Hz, 2H). ^{13}C NMR (100 MHz, Acetone- d_6) δ
574 167.40, 152.86, 152.51, 150.54, 147.67, 144.52, 140.05, 140.03, 134.84, 132.72, 129.45,
128.10, 127.64, 126.78, 125.75, 125.06, 121.35. HRMS $[\text{C}_{34}\text{H}_{22}\text{IrN}_4\text{Br}_2]^+$ calculated:
839.9897, found: 839.9859. Anal. ($\text{C}_{34}\text{H}_{22}\text{Br}_2\text{IrN}_4\text{PF}_6$) C, H, N: calculated: 41.52, 2.25, 5.70;
found: 41.72, 2.50, 5.66.

**Complex 1i.** Yield: 51% ^1H NMR (400 MHz, Acetone- d_6) δ 8.38 (d, $J = 2.7$ Hz, 2H), 8.29 (dt,
$J = 8.3, 1.2$ Hz, 2H), 8.02 (td, $J = 7.8, 1.5$ Hz, 2H), 7.93 – 7.86 (m, 6H), 7.29 – 7.22 (m, 6H),
6.40 (d, $J = 2.0$ Hz, 2H), 4.08 (s, 6H). ^{13}C NMR (100 MHz, Acetone- d_6) δ 206.18, 169.12,
581 167.51, 158.28, 153.96, 152.56, 150.20, 144.36, 139.93, 134.70, 127.62, 126.43, 125.72,
125.08, 121.30, 115.09, 112.54, 57.38, 30.43, 30.24, 30.05, 29.85, 29.66, 29.47, 29.28. HRMS
$[\text{C}_{34}\text{H}_{26}\text{O}_2\text{IrN}_4\text{Br}_2]^+$ calculated: 875.0031, found: 875.0041. Anal. ($\text{C}_{34}\text{H}_{26}\text{Br}_2\text{IrN}_4\text{O}_2\text{PF}_6$) C, H,
584 N: calculated: 40.05, 2.57, 5.50; found: 40.88, 2.77, 5.59.

**Complex 1j.** Yield: 58% ^1H NMR (400 MHz, Acetone- d_6) δ 8.38 (d, $J = 2.7$ Hz, 2H), 8.29 (dt,
$J = 8.3, 1.2$ Hz, 2H), 8.02 (td, $J = 7.8, 1.5$ Hz, 2H), 7.93 – 7.86 (m, 6H), 7.29 – 7.22 (m, 6H),
6.40 (d, $J = 2.0$ Hz, 2H), 4.08 (s, 6H). ^{13}C NMR (100 MHz, Acetone- d_6) δ 206.18, 169.12,
588 167.51, 158.28, 153.96, 152.56, 150.20, 144.36, 139.93, 134.70, 127.62, 126.43, 125.72,

125.08, 121.30, 115.09, 112.54, 57.38, 30.43, 30.24, 30.05, 29.85, 29.66, 29.47, 29.28. HRMS
$[\text{C}_{38}\text{H}_{32}\text{O}_2\text{RhN}_4]^+$ calculated: 679.1575, found: 679.1534. Anal. ($\text{C}_{38}\text{H}_{32}\text{RhN}_4\text{O}_2\text{PF}_6+\text{H}_2\text{O}$) C,
H, N: calculated: 54.17, 4.07, 6.65; found: 54.20, 4.17, 6.65.

**Complex 1k.** Yield: 63% ^1H NMR (400 MHz, Acetone- d_6) δ 8.40 (s, 2H), 8.22 (dd, J = 6.0,
0.5 Hz, 2H), 8.16 (dt, J = 8.2, 1.1 Hz, 2H), 7.97 – 7.88 (m, 2H), 7.85 (d, J = 7.9 Hz, 2H), 7.65
(dt, J = 5.8, 0.8 Hz, 2H), 7.48 (d, J = 6.0 Hz, 2H), 6.99 (s, 2H), 6.94 (ddd, J = 7.9, 1.6, 0.8 Hz,
2H), 6.28 (dd, J = 1.8, 0.9 Hz, 2H), 4.23 (s, 6H), 2.10 (s, 6H). ^{13}C NMR (101 MHz, Acetone)
δ 206.28, 169.20, 168.88, 166.10, 166.08, 164.90, 152.74, 149.97, 146.92, 142.42, 140.63,
597 140.62, 139.21, 134.43, 125.33, 125.15, 123.86, 123.68, 121.54, 120.38, 120.36, 107.45,
57.78, 30.43, 30.24, 30.05, 29.86, 29.66, 29.47, 29.28, 21.97. HRMS $[\text{C}_{38}\text{H}_{32}\text{O}_2\text{RhN}_4]^+$
calculated: 679.1575, found: 679.1541. Anal. ($\text{C}_{38}\text{H}_{32}\text{RhN}_4\text{O}_2\text{PF}_6$) C, H, N: calculated: 49.94,
3.53, 6.13; found: 49.47, 3.48, 6.07.

**Complex 1l.** Yield: 61%. ^1H NMR (400 MHz, Acetone- d_6) δ 8.67 (s, 2H), 8.45 (s, 4H), 8.31
(dt, J = 8.2, 1.0 Hz, 2H), 8.13 – 8.00 (m, 2H), 7.97 (d, J = 8.3 Hz, 2H), 7.78 (ddt, J = 5.8, 1.7,
0.9 Hz, 2H), 7.37 (dd, J = 8.3, 2.0 Hz, 2H), 7.09 (ddd, J = 7.3, 5.8, 1.4 Hz, 2H), 6.48 (dd, J =
2.0, 1.2 Hz, 2H). ^{13}C NMR (100 MHz, Acetone- d_6) δ 206.16, 167.21, 152.95, 152.25, 150.89,
605 148.32, 144.46, 140.15, 137.52, 134.76, 132.72, 132.59, 129.18, 127.68, 126.95, 125.76,
125.10, 121.40, 30.42, 30.23, 30.04, 29.85, 29.65, 29.46, 29.27. HRMS $[\text{C}_{34}\text{H}_{20}\text{RhN}_4\text{Br}_4]^+$
calculated: 906.7436, found: 906.7450. Anal. ($\text{C}_{34}\text{H}_{20}\text{Br}_4\text{RhN}_4\text{PF}_6$) C, H, N: calculated: 38.82,

1.92, 5.33; found: 38.97, 2.05, 5.34.

**Complex 1m.** Yield: 60%. ^1H NMR (400 MHz, Acetone- d_6) δ 8.40 (s, 2H), 8.22 (dd, $J = 6.0$,
0.5 Hz, 2H), 8.16 (dt, $J = 8.2$, 1.1 Hz, 2H), 7.93 (ddd, $J = 8.2$, 7.4, 1.6 Hz, 2H), 7.85 (d, $J =$
7.9 Hz, 2H), 7.65 (ddd, $J = 4.9$, 1.6, 0.8 Hz, 2H), 7.48 (d, $J = 6.0$ Hz, 2H), 6.99 (ddd, $J = 7.3$,
5.7, 1.4 Hz, 2H), 6.94 (ddd, $J = 7.9$, 1.6, 0.8 Hz, 2H), 6.28 (dd, $J = 1.8$, 0.9 Hz, 2H), 4.23 (s,
5H), 2.10 (s, 5H). ^{13}C NMR (100 MHz, Acetone- d_6) δ 205.50, 170.00, 169.68, 164.22, 163.23,
614 163.21, 151.75, 147.56, 146.24, 146.09, 142.37, 133.16, 133.14, 132.86, 129.07, 129.06,
615 128.64, 123.24, 123.02, 122.67, 120.89, 116.99, 106.73, 57.11, 29.76, 29.57, 29.37, 29.32,
29.18, 29.12, 28.99, 28.94, 28.80, 28.60. HRMS [$\text{C}_{46}\text{H}_{30}\text{RhN}_4\text{Br}_2$] $^+$ calculated: 900.9872,
found: 900.0836. Anal. ($\text{C}_{46}\text{H}_{30}\text{Br}_2\text{RhN}_4\text{PF}_6$) C, H, N: calculated: 52.80, 2.89, 5.35; found:
52.90, 2.95, 5.38.

**Complex 1n.** Yield: 64%. ^1H NMR (400 MHz, Acetone- d_6) δ 8.95 (dd, $J = 8.3$, 1.5 Hz, 2H),
8.57 (ddd, $J = 4.9$, 1.5, 0.5 Hz, 2H), 8.41 (s, 2H), 8.32 (dt, $J = 8.2$, 1.1 Hz, 2H), 8.12 – 7.96 (m,
6H), 7.67 (ddt, $J = 5.8$, 1.6, 0.9 Hz, 2H), 7.37 (dd, $J = 8.3$, 2.0 Hz, 2H), 7.11 (ddd, $J = 7.3$, 5.7,
1.4 Hz, 2H), 6.52 (dd, $J = 2.0$, 1.1 Hz, 2H). ^{13}C NMR (100 MHz, Acetone- d_6) δ 205.50,
623 205.28, 169.07, 164.10, 151.19, 149.75, 145.58, 143.39, 139.43, 139.39, 135.02, 131.45,
624 128.41, 127.06, 127.01, 126.64, 124.66, 124.64, 124.44, 120.84, 116.99, 29.75, 29.55, 29.36,
29.23, 29.17, 28.98, 28.79, 28.59. HRMS [$\text{C}_{34}\text{H}_{22}\text{RhN}_4\text{Br}_2$] $^+$ calculated: 748.9246, found:
748.7485. Anal. ($\text{C}_{34}\text{H}_{22}\text{Br}_2\text{RhN}_4\text{PF}_6$) C, H, N: calculated: 45.67, 2.48, 6.27; found: 45.47,

2.45, 6.21.

[revised manuscript text omitted]

Reviewers' Comments:

Reviewer #1:

Remarks to the Author:

The authors have done a reasonable job at revising their manuscript. They provide new DSF on VH298, that are more closely in line with the thermal shifts that are observed of this known compound. They ascribe the inconsistency with the previous report to potential inactivation with the previous stock of the compound, albeit no evidence is provided to support this claim. Nonetheless, the data now appear convincing, and the authors report to have obtained the new data using VH298 compound from three different commercial sources.

The authors are also now more scholarly acknowledging the prior work developing VH298, and in acknowledging that their compound is 100-fold less potent than VH298 - so more scholarly and accurately place their report in the context of the state-of-the-art.

Based on this, I think the manuscript now is in a format that may be considered suitable for publication in the journal, and ready to reach the wider scientific community.

Our Responses to the Comments of the Reviewer #1

Reviewer #1 (Remarks to the Author):

Comments:

1) The authors have done a reasonable job at revising their manuscript. They provide new DSF on VH298, that are more closely in line with the thermal shifts that are observed of this known compound. They ascribe the inconsistency with the previous report to potential inactivation with the previous stock of the compound, albeit no evidence is provided to support this claim. Nonetheless, the data now appear convincing, and the authors report to have obtained the new data using VH298 compound from three different commercial sources.

Our response: We sincerely thank the reviewer for the valuable comments.

2) The authors are also now more scholarly acknowledging the prior work developing VH298, and in acknowledging that their compound is 100-fold less potent than VH298 - so more scholarly and accurately place their report in the context of the state-of-the-art.

Our response: We are very grateful to the reviewer for the constructive suggestions, which have significantly improved the quality of this manuscript.

3) Based on this, I think the manuscript now is in a format that may be considered suitable for publication in the journal, and ready to reach the wider scientific community.

Our response: We thank the reviewer for the valuable comment.